# MoDeGPT: Modular Decomposition for Large Language Model Compression

**Chi-Heng Lin**[*]
Samsung Research America

**Shangqian Gao**
Florida State University

**James Seale Smith**
Samsung Research America

**Abhishek Patel**
Samsung Research America

**Shikhar Tuli**
Samsung Research America

**Yilin Shen**
Samsung Research America

**Hongxia Jin**
Samsung Research America

**Yen-Chang Hsu**
Samsung Research America

## ABSTRACT

Large Language Models (LLMs) have significantly advanced AI with their exceptional performance across a wide range of tasks. However, their extensive computational requirements restrict their use on devices with limited resources. While recent compression methods based on low-rank matrices show potential solutions, they often suffer from significant loss of accuracy or introduce substantial overhead in parameters and inference time. In this paper, we introduce Modular Decomposition (MoDeGPT), a new, efficient, and structured compression framework that overcomes these limitations. MoDeGPT jointly decomposes pairs of consecutive subcomponents within Transformer blocks, reduces hidden dimensions through output reconstruction on a larger structural scale than conventional low-rank methods, and repurposes three classical matrix decomposition algorithms—Nyström approximation, CR decomposition, and SVD—to ensure bounded errors in our novel decomposition approach. Our experiments show that MoDeGPT, without relying on backward propagation, consistently matches or surpasses the performance of prior techniques that depend on gradient information, while achieving a 98% reduction in compute costs when compressing a 13B-parameter model. On LLaMA-2/3 and OPT models, MoDeGPT retains 90-95% of zero-shot performance with compression rates of 25-30%. The compression process can be completed on a single GPU in a few hours, boosting inference throughput by up to 46%.

## 1 INTRODUCTION

Recent advancements in Large Language Models (LLMs) (Thoppilan et al., 2022; OpenAI, 2023; Touvron et al., 2023; Zhang et al., 2022; AI@Meta, 2024) have led to remarkable breakthroughs in the understanding and generation of natural language. Despite their significant capabilities, these models are computationally and memory-intensive, posing deployment challenges on resource-limited devices. To mitigate these challenges, model compression (Gupta & Agrawal, 2022; Zhu et al., 2023) has emerged as a popular post-training solution, reducing model size and complexity.

Predominant compression techniques encompass model distillation (Sun et al., 2019; 2020; Pan et al., 2020), pruning (LeCun et al., 1989; Hassibi et al., 1993; Suzuki et al., 2018; Wang et al., 2019b; Zafrir et al., 2021; Xia et al., 2022; Kurtic et al., 2022; Ma et al., 2023; van der Ouderaa et al., 2023), matrix decomposition (Hsu et al., 2022; Noach & Goldberg, 2020; Golub & Reinsch, 1971), and quantization (Gholami et al., 2022; Bai et al., 2020; Frantar et al., 2022; Wang et al., 2023). This study focuses on matrix decomposition techniques that require minimal computing resources and do not involve backward propagation as seen in recovery fine-tuning (RFT) or Fisher matrix calculations from Taylor expansion (Ma et al., 2023; van der Ouderaa et al., 2023). Conven-

---

[*]chiheng.lin@samsung.com

tional matrix decomposition such as SVD typically splits each matrix $\boldsymbol{W} \in \mathbb{R}^{d \times d}$ into two low-rank matrices $\boldsymbol{W} = \boldsymbol{AB}$, requiring the rank less than $d/2$ to achieve true compression, as shown in Figure 1(b). This stringent requirement often results in a significant drop in accuracy, necessitating the use of RFT (Hsu et al., 2022). A novel decomposition approach, SliceGPT (Ashkboos et al., 2024), multiplies the original matrix by an orthogonal matrix, effectively projecting inputs into a lower-dimensional subspace and reducing the matrix's overall dimensionality. However, this approach requires additional adapters to manage the reduced dimensions; as illustrated in Figure 1(c), adapters $\boldsymbol{Q}_i^\top \boldsymbol{Q}_j$ are added to the residual paths to facilitate this reduction. For a target sparsity $s$, this introduces additional $2(1-s)^2 d^2$ parameters per layer, which can add up to 10% of additional parameters, significantly offsetting the parameter savings. In summary, matrix decomposition approaches either (i) **discard a large portion of ranks**, or (ii) **introduce substantial parameter overheads**. These challenges significantly hinder the effective reduction of parameters without compromising accuracy.

In response to these challenges, we introduce MoDeGPT, which applies matrix decomposition to multiple matrices jointly, avoiding the dual-matrix structure and extra adapters used in prior methods. As depicted in Figure 1(d), MoDeGPT elevates the matrix decomposition approach to a modular level by grouping weight matrices into modules and then applying matrix decomposition jointly within each module. Unlike SliceGPT, MoDeGPT reduces the intermediate dimensions within each module rather than between blocks, as illustrated by the matrix shapes in Figure 1(c) and (d). This crucial difference eliminates the need for adapters while still enabling dimension reduction in the compressed matrix. Importantly, MoDeGPT establishes a comprehensive mathematical framework that maps each module's compression task to one of the three matrix approximation techniques: CR decomposition (Drineas et al., 2006), singular value decomposition (SVD) (Golub & Reinsch, 1971), and Nyström approximation (Gittens & Mahoney, 2013; Musco & Musco, 2017). These methods enable MoDeGPT to efficiently compress matrices. In summary, we make the following contributions:

- We introduce MoDeGPT, a training-free compression method that jointly decomposes multiple matrices within a module using closed-form expressions. To our knowledge, this is the first method to apply matrix decomposition at the module level for model compression.

- We extend the theoretical foundations of language model weight decomposition beyond SVD, introducing a systematic framework for categorizing approximation challenges in Transformer compression, complete with error guarantees.

- To our knowledge, this is the first demonstration in large language models where a purely matrix decomposition-based approach achieves state-of-the-art structured compression efficiency, rivaling the compression rates of semi-structured pruning methods—all without the need for recovery fine-tuning

- We present a thorough evaluation of MoDeGPT, comparing it against existing methods across key metrics, including perplexity, downstream accuracy, and real-world speed improvements. MoDeGPT preserves up to 90% of zero-shot performance with compression rates of up to 30% on LLaMA 2 and 3, significantly outperforming prior approache. Moreover, MoDeGPT delivers a 46% increase in inference throughput, further enhancing its practical value.

## 2 BACKGROUND AND RELATED WORK

In this section, we begin by reviewing the existing body of literature related to LLM compression, highlighting key contributions and methodologies in the field. Subsequently, we examine the standard components of a transformer decoder layer. Lastly, we delve into the three matrix approximations employed for our proposed compression across various components in a transformer layer.

### 2.1 RELATED WORKS

**Pruning** In early pruning methods like magnitude-based tuning, scalability is achieved but often at the cost of reduced effectiveness in large language models (LLMs) (Hagiwara, 1994; Han et al., 2015; Li et al., 2017; Frantar & Alistarh, 2023; van der Ouderaa et al., 2023). To improve

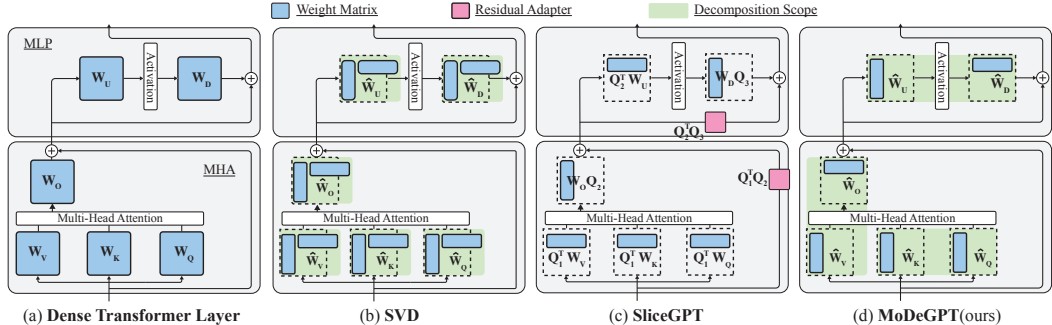

**Figure 1: Comparison of Matrix Decomposition-Based Methods for Transformer Compression. (a)** Original transformer layer. **(b)** SVD applied to each weight matrix separately, resulting in dual matrices. **(c)** SliceGPT multiplies each weight matrix by an orthogonal matrix $Q$, reducing dimensions and introducing additional adapters. **(d)** MoDeGPT organizes matrices into modules (highlighted by green boxes) and jointly decomposes them, producing reduced-size matrices without extra adapters.

performance while managing computational demands, frameworks such as Optimal Brain Damage (LeCun et al., 1989) and Surgeon (Hassibi et al., 1993; Yu et al., 2022; van der Ouderaa et al., 2023) incorporate second-order loss information, necessitating substantial resources for Hessian calculations. Recent adaptations like WoodFisher (Singh & Alistarh, 2020), Kronecker factorization (Wang et al., 2019a; van der Ouderaa et al., 2023), and layer-wise compression (Dong et al., 2017; Frantar & Alistarh, 2022) aim to streamline these intensive methods. Concurrently, learnable parameters for pruning in vision and language models have been investigated (Liu et al., 2017; Huang & Wang, 2018; Xia et al., 2022), although these techniques generally demand significant computational resources for intensive backward propagation. Other approaches, such as feature-mimic-based methods (An et al., 2024; Ji et al., 2024), have not matched the performance of gradient-based methods like LLM Surgeon (van der Ouderaa et al., 2023). Alternatives like SparseGPT (Frantar & Alistarh, 2023), Wanda (Sun et al., 2024), and ZeroPruner (Dong et al., 2024), exploring unstructured and semi-structured pruning, offer scalability but often compromise runtime speed. Additional research has utilized layer importance scores for layer pruning and sparsity distribution, as demonstrated by ShortGPT (Men et al., 2024), OWL (Yin et al., 2023), LaCo (Yang et al., 2024), and others (Chen et al., 2024). Recent advances in LLM compression have introduced innovative methods such as LLM-Pruner (Ma et al., 2023), LLM Surgeon (van der Ouderaa et al., 2023), and SliceGPT (Ashkboos et al., 2024), marking significant progress in the field by providing effective compression techniques for LLMs.

**Low-Rank Matrix Approximation** In related low-rank matrix techniques for compression, the traditional decomposition approach substitutes matrices with two low-rank matrices but retains the original dimensions, which can limit effectiveness (Noach & Goldberg, 2020; Hsu et al., 2022; Golub & Reinsch, 1971; Povey et al., 2018; Xu et al., 2023; Yuan et al., 2023; Wang et al., 2024; Yu & Wu, 2023; Chen et al., 2021). MoDeGPT improves upon this by applying low-rank approximation to matrix

**Table 1:** LLM Compression Comparisons.

| Method | No Backward Propagation | No Additional Parameters | Fully-Structured |
|---|---|---|---|
| LLM Pruner | ✗ | ✓ | ✓ |
| LLM Surgeon | ✗ | ✓ | ✓ |
| SliceGPT | ✓ | ✗ | ✓ |
| SparseGPT | ✓ | ✓ | semi- |
| **MoDeGPT (ours)** | ✓ | ✓ | ✓ |

pairs, reducing the size of individual matrices and merging the additional matrices from the decompositions. SliceGPT introduces a technique involving matrix multiplication with orthogonal matrices derived from PCA to compress weights, which reduces matrix sizes but adds additional parameters (Ashkboos et al., 2024). In contrast, MoDeGPT compresses without adding parameters by folding the decomposed matrices back to the original weights. A summary of MoDeGPT's comparison to other leading LLM compression methods is provided in Table 1.

## 2.2 TRANSFORMER ARCHITECTURE

The transformer architecture (Vaswani et al., 2017) consists of multiple decoder layers. A typical layer such as in LLAMA (Touvron et al., 2023; AI@Meta, 2024) includes two blocks: the Multi-Head Attention (MHA) and Multi-Layer Perceptron (MLP). Let $T$, $d_h$, $d_{\text{int}}$, and $H$ denote the sequence length, hidden dimension, intermediate dimension, and the number of attention heads, respectively,

the formulation of these blocks is as follows:

$$\text{(MLP block)} \quad f_{\text{MLP}}(\boldsymbol{X}) = \overset{\text{Type-I}}{\sigma_s(\boldsymbol{X}\boldsymbol{W}_U)}\boldsymbol{W}_D, \tag{1}$$

$$\text{(MHA block)} \quad f_{\text{MHA}}(\boldsymbol{X}) = \sum_{i=1}^{H} \text{Softmax}\left(\overset{\text{Type-II}}{\sigma_r(\boldsymbol{X}\boldsymbol{W}_{Q,i})\sigma_r^\top(\boldsymbol{X}\boldsymbol{W}_{K,i})}\right)\overset{\text{Type-III}}{\boldsymbol{X}\boldsymbol{W}_{V,i}\boldsymbol{W}_{O,i}}, \tag{2}$$

where $\boldsymbol{X} \in \mathbb{R}^{T \times d_h}$ is the input matrix, $\boldsymbol{W}_{Q,i}, \boldsymbol{W}_{K,i}, \boldsymbol{W}_{V,i} \in \mathbb{R}^{d_h \times \frac{d_h}{H}}, \boldsymbol{W}_{O,i} \in \mathbb{R}^{\frac{d_h}{H} \times d_h}$ are the head-specific query, key, value, and output matrices. The matrices $\boldsymbol{W}_U \in \mathbb{R}^{d_h \times d_{\text{int}}}$ and $\boldsymbol{W}_D \in \mathbb{R}^{d_{\text{int}} \times d_h}$ denote up and down matrices, respectively, with $\sigma_r$ and $\sigma_s$ denoting positional embedding and nonlinear activation functions. Note that our MLP formulation encompasses the gated MLP: the up matrix is defined by the concatenations of the gated and up matrix $\boldsymbol{W}_U = [\boldsymbol{W}_u^\top, \boldsymbol{W}_g^\top]^\top$, and the nonlinear function is defined by $\sigma_s(\boldsymbol{X}\boldsymbol{W}_U) := \boldsymbol{X}\boldsymbol{W}_u \odot \sigma_g(\boldsymbol{X}\boldsymbol{W}_g)$, where $\sigma_g$ is the gate function.

In the expressions of equation 1 and equation 2, the blocks can be divided into three types of functional modules, each associated with a pair of matrices:

$$f_{\text{Type-I}}(\boldsymbol{X}; \boldsymbol{W}_U, \boldsymbol{W}_D) = \sigma_s(\boldsymbol{X}\boldsymbol{W}_U)\boldsymbol{W}_D, \quad f_{\text{Type-II}}(\boldsymbol{X}; \boldsymbol{W}_K^i, \boldsymbol{W}_Q^i) = \sigma_r(\boldsymbol{X}\boldsymbol{W}_{Q,i})\sigma_r^\top(\boldsymbol{X}\boldsymbol{W}_{K,i}),$$

$$f_{\text{Type-III}}(\boldsymbol{X}; \boldsymbol{W}_V^i, \boldsymbol{W}_O^i) = \boldsymbol{X}\boldsymbol{W}_{V,i}\boldsymbol{W}_{O,i},$$

where $\boldsymbol{X}$ denotes the input and the variables after ";" denote the associated matrices. These three types are distinguished by varying levels of nonlinearity. We will employ different matrix decomposition methods for compression based on the optimization tractability of each type.

## 2.3 LOW-RANK MATRIX APPROXIMATION

The goal of a low-rank approximation method is to approximate a matrix $\boldsymbol{W} \in \mathbb{R}^{d_1 \times d_2}$ with two low-rank matrices $\boldsymbol{A} \in \mathbb{R}^{d_1 \times k}$ and $\boldsymbol{B} \in \mathbb{R}^{k \times d_2}$. For formalism, we make the following definition:

**Definition 1.** *For a low-rank approximation method $\mathcal{M}$ that decomposes a matrix $\boldsymbol{W}$ into $\boldsymbol{A}$ and $\boldsymbol{B}$, the approximation matrix is $\boldsymbol{W}_{\mathcal{M}} = \boldsymbol{A}\boldsymbol{B}$ and the error relative to $\boldsymbol{W}$ is $\mathcal{E}_{\mathcal{M}}(\boldsymbol{W}) = \|\boldsymbol{W} - \boldsymbol{W}_{\mathcal{M}}\|_F$.*

We review three approximation methods that facilitate our algorithms in the next section.

**I. Nyström approximation (Gittens & Mahoney, 2013)**   If $\boldsymbol{W}$ is a positive semidefinite matrix, let $\boldsymbol{S}_k$ be a $k$-column selection matrix where each column has a single non-zero element indicating the selected index, then the corresponding Nyström approximation of $\boldsymbol{W}$ is,

$$\boldsymbol{W}_{\text{Nys}} = \boldsymbol{A}\boldsymbol{B}, \quad \text{where} \quad \boldsymbol{A} = \boldsymbol{W}\boldsymbol{S}_k \quad \text{and} \quad \boldsymbol{B} = (\boldsymbol{S}_k^\top \boldsymbol{W}\boldsymbol{S}_k)^\dagger \boldsymbol{S}_k^\top \boldsymbol{W}. \tag{3}$$

**II. CR decomposition (Drineas et al., 2006)**   Assuming $\boldsymbol{W}$ can be factored as $\boldsymbol{W}_1\boldsymbol{W}_2$, let $\boldsymbol{S}_k$ be a $k$-column selection matrix, the corresponding CR approximation of $\boldsymbol{W}$ is

$$\boldsymbol{W}_{\text{CR}} = \boldsymbol{A}\boldsymbol{B}, \quad \text{where} \quad \boldsymbol{A} = \boldsymbol{W}_1\boldsymbol{S}_k \quad \text{and} \quad \boldsymbol{B} = \boldsymbol{S}_k^\top \boldsymbol{W}_2. \tag{4}$$

**III. Singular value decomposition (Golub & Reinsch, 1971)**   SVD is renowned for yielding the minimum approximation error when measured in the Frobenius norm. It decomposes $\boldsymbol{W}$ into:

$$\boldsymbol{W}_{\text{SVD}} = \boldsymbol{A}\boldsymbol{B}, \quad \text{where} \quad \boldsymbol{A} = \boldsymbol{U}_k \quad \text{and} \quad \boldsymbol{B} = \boldsymbol{\Sigma}_k \boldsymbol{V}_k^\top. \tag{5}$$

Here, $\boldsymbol{U}_k$ and $\boldsymbol{V}_k$ are matrices containing the top-$k$ left and right singular vectors, respectively, and $\boldsymbol{\Sigma}_k$ is the diagonal matrix consisting of the top-$k$ singular values of $\boldsymbol{W}$.

## 3 MoDeGPT

MoDeGPT introduces a module-level optimization that jointly compresses two matrices within each of our three defined functional modules, rather than compressing each matrix independently as in traditional low-rank approximation methods.

An illustration of MoDeGPT is presented in Figure 2, where different colors distinguish the various modules. For each module, we apply a tailored low-rank approximation to compress the matrix pair within it. The twill hatch pattern represents dimension reductions.

In the rest of this section, we first present the mathematical objective for our approach. Then, we detail our application of low-rank approximations for effective compression within each module. Finally, we introduce a method for assigning sparsity levels across different layers that requires only one forward pass of the model on the calibration data.

## 3.1 MODULAR RECONSTRUCTION OBJECTIVE

The objective of MoDeGPT is to jointly optimize two matrices within the module types described in Sec. 2.2, a process we term modular decomposition, to minimize the modular reconstruction error:

$$V^* \triangleq \min_{\hat{W}_1, \hat{W}_2} \sum_{i=1}^{N} \|f(X_i; W_1, W_2) - f(X_i; \hat{W}_1, \hat{W}_2)\|_F^2 \text{ such that } (\hat{W}_1, \hat{W}_2) \in \mathcal{C}, \quad (6)$$

where $X_i \in \mathbb{R}^{T \times d}$ are samples in the calibration set, and $\mathcal{C}$ represents the constrained search space for compressed matrices that mandates specific structures or dimensions.

A key motivation of our objective is that it expands the search space to include dimension-reduced matrices, thereby increasing optimization flexibility and enhancing inference speedup in the compressed model. This contrasts with independent optimization, where each matrix must adhere to the original dimensions.

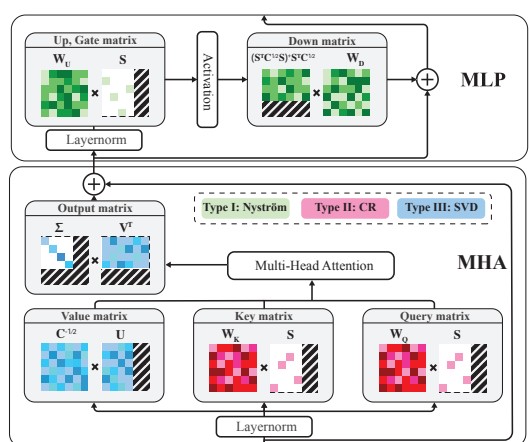

**Figure 2: The MoDeGPT Framework.** MoDeGPT divides a transformer layer into three distinct colored modules, each optimizing two matrices using a specific low-rank approximation method. A twill hatch pattern represents the dimension reduction.

## 3.2 ALGORITHMS

**From LLM compression to matrix decomposition**   The core technical contribution of this work is the establishment of a one-to-one mapping between a specific type of modular compression problem and a corresponding matrix decomposition problem. As outlined in Section 2.2, the modules in the transformer architecture can be categorized based on the number of nonlinear functions they contain: Type I, II, and III modules contain 1, 2, and 0 nonlinearties, respectively. For a weight matrix $W$ within a nonlinear function, we compress it into a structured form $\hat{W} = W S_k$, where $S_k$ is a $k$-column selection matrix to be optimized. This restrictive structural form is a cornerstone of our framework, as it ensures the *tractable* optimization of equation 6.

After characterizing the modules and the structure of the compressed matrices, our framework solves the modular decomposition problem in equation 6 for each module. Since each module contains a different number of nonlinear functions, the corresponding solutions vary. As we demonstrate in the subsequent sections, the solutions correspond to Nyström, CR, and SVD for Type I, II, and III modules, respectively. A summary of this roadmap is provided in Table 2. The detailed connections are formalized in the following subsections, with detailed proofs included in Appendix A.

| Module Type | I | II | III |
|---|---|---|---|
| Weight Matrices | up,down,gate | key,query | value,output |
| Associated Decomp. | Nyström | CR | SVD |
| # Nonlinearities | 1 | 2 | 0 |
| Compression Alg. | Alg. 1 | Alg. 2 | Alg. 3 |

**Table 2:** Module characteristics and their associated matrix decompositions.

**TYPE-I COMPRESSION**   First, we focus on the MLP module. As detailed in Section 2.2, the matrices $W_1$ and $W_2$ that require compression are $W_U$ and $W_D$. Since $W_U$ resides within a nonlinear function $\sigma_s$, we constrain its approximation to the form $W_U S_k$ for tractable optimization of equation 6, where $S_k$ is the $k$-column selection matrix. For $W_D$, we simply ensure that its dimensions are compatible with $W_U S_k$. Our first theorem suggests that when a single column selection matrix is used, the optimization in equation 6 is closely related to the Nyström approximation of the activation correlation matrix.

**Theorem 1** (MLP compression by Nyström approximation).  *Let $\hat{W}_U$ be searched over the matrix multiplication form $W_U S_k$, where $S_k$ is a $k$-column selection matrix, and let $\hat{W}_D$ be searched over $\mathbb{R}^{k \times d_h}$. The optimal $\hat{W}_D^*$ is then given by: $(S_k^\top C_\sigma S_k)^\dagger S_k^\top C_\sigma W_D$. Using $W_U S_k$ and $\hat{W}_D^*$ as the*

---

**Algorithm 1** Type-I compression for MLP by Nyström approximation.

---

1: **Input:** concatenated up and gated matrices $\boldsymbol{W}_U \in \mathbb{R}^{d_h \times d_{\text{int}}}$, down matrix $\boldsymbol{W}_D \in \mathbb{R}^{d_{\text{int}} \times d_h}$, activation correlation $\boldsymbol{C}_\sigma = \sum_{i=1}^N \sigma(\boldsymbol{X}_i \boldsymbol{W}_U)^\top \sigma(\boldsymbol{X}_i \boldsymbol{W}_U)$, rank $k = \lceil (1 - \text{sparsity}) d_{\text{int}} \rceil$, and ridge intensity $\lambda$

2: $s_i \leftarrow [\boldsymbol{C}_\sigma (\boldsymbol{C}_\sigma + \lambda \boldsymbol{I})^{-1}]_{ii}$, for $i = 1, \dots, d_{\text{int}}$          ▷ *Calculate the ridge leverage score*

3: Let $\boldsymbol{S}_k \in \mathbb{R}^{d_{\text{int}} \times k}$ be the matrix that selects the top $k$ columns based on $s_i$ scores

4: **return** $(\boldsymbol{W}_U, \boldsymbol{W}_D) \leftarrow (\boldsymbol{W}_U \boldsymbol{S}_k, \ (\boldsymbol{S}_k^\top \boldsymbol{C}_\sigma \boldsymbol{S}_k)^\dagger \boldsymbol{S}_k^\top \boldsymbol{C}_\sigma \boldsymbol{W}_D)$

---

*compressed matrices, the Type-I reconstruction error in equation 6 satisfies:*

$$V_I \leq \|\boldsymbol{W}_D\|_2^2 \|\boldsymbol{C}_\sigma^{-1}\|_2 \mathcal{E}_{Nys}^2(\boldsymbol{C}_\sigma), \tag{7}$$

*where $\mathcal{E}_{Nys}(\boldsymbol{C}_\sigma)$ denotes the Nyström approximation error, defined in Def. 1, relative to the activation correlation $\boldsymbol{C}_\sigma \triangleq \sum_{i=1}^N \sigma(\boldsymbol{X}_i \boldsymbol{W}_U)^\top \sigma(\boldsymbol{X}_i \boldsymbol{W}_U)$, using the same $\boldsymbol{S}_k$ in the compression of $\boldsymbol{W}_U$.*

Theorem 1 shows that effective Type-I compression can be achieved through a well-designed Nyström approximation of $\boldsymbol{C}_\sigma$. Thus, we propose Algorithm 1 to control the error as shown below.

**Proposition 1** (**MLP compression error**). *Suppose that the rank $k$ and the scores $s_i$ in Algorithm 1 are chosen such that there exists an error $\varepsilon > 0$ satisfying $\varepsilon \geq \sum_{i=k+1}^{d_{int}} s_i$, then the Type-I modular reconstruction error in equation 6 is bounded by $V_I \leq \|\boldsymbol{W}_D\|_2^2 \|\boldsymbol{C}_\sigma^{-1}\|_2 \frac{\varepsilon^2 d_{int}^2}{k^2 (1-\varepsilon)^2} \sum_{i=k+1}^{d_{int}} \sigma_i^2(\boldsymbol{C}_\sigma)$, where $d_{int}$ and $\sigma_i$ denote the intermediate dimension (i.e., the input dimension of $W_D$) and singular values, respectively.*

**TYPE-II COMPRESSION**    Next, we turn our attention to the Type-II module, which includes the key-query interactions within the multi-head attention mechanisms. We will apply compression to each head independently [1]. Given that both $\boldsymbol{W}_Q$ and $\boldsymbol{W}_K$ are embedded with nonlinear functions, for tractability in the optimization of equation 6, the matrices are compressed using a column selection matrix: $\hat{\boldsymbol{W}}_Q = \boldsymbol{W}_Q \boldsymbol{S}_k$ and $\hat{\boldsymbol{W}}_K = \boldsymbol{W}_K \boldsymbol{S}_k$, where $\boldsymbol{S}_k$ is a shared $k$-column selection matrix. When both two compressed matrices are multiplied by the column selection matrix, the modular reconstruction problem naturally connects to the CR decomposition of the product of key-query correlations, as elaborated in the following theorem.

**Theorem 2** (**Key-Query compression by CR approximation**). *Let the compressed $\hat{\boldsymbol{W}}_Q$, $\hat{\boldsymbol{W}}_K$ to be the form of $\boldsymbol{W}_Q \boldsymbol{S}_k, \boldsymbol{W}_K \boldsymbol{S}_k$, then Type-II reconstruction error in equation 6 has*

$$V_{II} \leq \mathcal{E}_{CR}^2(\boldsymbol{C}_K^{\frac{1}{2}} \boldsymbol{C}_Q^{\frac{1}{2}}), \tag{8}$$

*where $\mathcal{E}_{CR}$ denotes the CR approximation error, defined in Def. 1, relative to $\boldsymbol{C}_Q^{1/2} \boldsymbol{C}_K^{1/2}$, utilizing the same $\boldsymbol{S}_k$ in the compression. Here, the matrices $\boldsymbol{C}_Q \triangleq \sum_{i=1}^N \sigma(\boldsymbol{X}_i \boldsymbol{W}_Q)^\top \sigma(\boldsymbol{X}_i \boldsymbol{W}_Q)$ and $\boldsymbol{C}_K \triangleq \sum_{i=1}^N \sigma(\boldsymbol{X}_i \boldsymbol{W}_K)^\top \sigma(\boldsymbol{X}_i \boldsymbol{W}_K)$ denote the correlations of query and key states, respectively.*

The preceding theorem indicates that effective compression for the Type-II module can be achieved using a thoughtfully constructed CR approximation. In response, we present Algorithm 2, which offers the following guarantees for reconstruction:

**Proposition 2** (**Key-Query compression error**). *If we adopt Algorithm 2 then Type-II modular reconstruction error is bounded by $V_{II} \leq \left(\frac{d_h - k}{d_h}\right)^2 \left(\sum_{i=1}^{d_h} \sigma_i(\boldsymbol{C}_K)\right) \left(\sum_{i=1}^{d_h} \sigma_i(\boldsymbol{C}_Q)\right)$, where $\sigma_i$ denotes the singular values.*

**TYPE-III COMPRESSION**    Finally, we focus on the Type-III module, which involves the value-output matrices. For clarity and simplicity, we omit the head dependency. The module has no nonlinar function involved $f(\boldsymbol{X}) = \boldsymbol{X} \hat{\boldsymbol{W}}_V \hat{\boldsymbol{W}}_O$, so we seek general low-rank matrices for compressions: $\hat{\boldsymbol{W}}_V \in \mathbb{R}^{d_h \times k}$, $\hat{\boldsymbol{W}}_O \in \mathbb{R}^{k \times d_h}$ such that $\hat{\boldsymbol{W}}_V \hat{\boldsymbol{W}}_O \approx \boldsymbol{W}_V \boldsymbol{W}_O$. The subsequent theorem

---

[1]Dependency on the head is omitted in the equations for ease of notation.

---

**Algorithm 2** Type-II compression for key-query matrices by CR decomposition.

---

1: **Input:** head-specific query matrices $\boldsymbol{W}_{Q,j} \in \mathbb{R}^{d_h \times d_h/H}$, key matrices $\boldsymbol{W}_{K,j} \in \mathbb{R}^{d_h \times d_h/H}$, query state correlations $\boldsymbol{C}_{Q,j} = \sum_{i=1}^{N} \sigma_r(\boldsymbol{X}_i \boldsymbol{W}_{Q,j})^\top \sigma_r(\boldsymbol{X}_i \boldsymbol{W}_{Q,j})$, key state correlations $\boldsymbol{C}_{K,j} = \sum_{i=1}^{N} \sigma(\boldsymbol{X}_i \boldsymbol{W}_{K,j})^\top \sigma(\boldsymbol{X}_i \boldsymbol{W}_{K,j})$, for head $j = 1, \ldots, H$, and rank $k = \lceil (1 - \text{sparsity}) d_h / H \rceil$

2: **for** $j = 1, \ldots, H$ **do**                    ▷ *Apply compression to each head independently*

3:     $s_i \leftarrow \|\boldsymbol{C}_{Q,j}^{1/2}[:, i]\| \|\boldsymbol{C}_{K,j}^{1/2}[:, i]\|$                    ▷ *Calculate the norm score*

4:     Let $\boldsymbol{S}_k \in \mathbb{R}^{d_h \times k}$ be the matrix that selects the top $k$ columns based on $s_i$ scores

5:     $(\boldsymbol{W}_{Q,j}, \boldsymbol{W}_{K,j}) \leftarrow (\boldsymbol{W}_{Q,j} \boldsymbol{S}_k, \boldsymbol{W}_{K,j} \boldsymbol{S}_k)$

6: **return** $(\boldsymbol{W}_Q, \boldsymbol{W}_K) \leftarrow ([\boldsymbol{W}_{Q,1}, \ldots, \boldsymbol{W}_{Q,H}], [\boldsymbol{W}_{K,1}, \ldots, \boldsymbol{W}_{K,H}])$                    ▷ *Concatenate the heads*

---

**Algorithm 3** Type-III compression for value-output matrices by SVD.

---

1: **Input:** head-specific value matrices $\boldsymbol{W}_{V,j} \in \mathbb{R}^{d_h \times d_h/H}$, output matrices $\boldsymbol{W}_{O,j} \in \mathbb{R}^{d_h/H \times d_h}$ for head $j = 1, \ldots, H$, input correlation $\boldsymbol{C} = \sum_{i=1}^{N} \boldsymbol{X}_i^\top \boldsymbol{X}_i$, and rank $k = \lceil (1 - \text{sparsity}) d_h / H \rceil$

2: **for** $j = 1, \ldots, H$ **do**                    ▷ *Apply compression to each head independently*

3:     $(\boldsymbol{U}, \boldsymbol{\Sigma}, \boldsymbol{V}^\top) \leftarrow SVD(\boldsymbol{C}^{1/2} \boldsymbol{W}_{V,j})$                    ▷ *Efficient SVD of $\boldsymbol{C}^{1/2} \boldsymbol{W}_{V,j} \boldsymbol{W}_{O,j}$ (1/2)*

4:     $(\boldsymbol{U}', \boldsymbol{\Sigma}', \boldsymbol{V}'^\top) \leftarrow SVD(\boldsymbol{\Sigma} \boldsymbol{V}^\top \boldsymbol{W}_{O,j})$                    ▷ *Efficient SVD of $\boldsymbol{C}^{1/2} \boldsymbol{W}_{V,j} \boldsymbol{W}_{O,j}$ (2/2)*

5:     $(\boldsymbol{W}_{V,j}, \boldsymbol{W}_{O,j}) \leftarrow (\boldsymbol{C}^{-1/2} \boldsymbol{U} \boldsymbol{U}'[:, :k], \boldsymbol{\Sigma}'[:k, :k] \boldsymbol{V}'[:, :k]^\top)$

6: **return** $(\boldsymbol{W}_V, \boldsymbol{W}_O) \leftarrow ([\boldsymbol{W}_{V,1}, \ldots, \boldsymbol{W}_{V,H}], [\boldsymbol{W}_{O,1}, \ldots, \boldsymbol{W}_{O,H}])$                    ▷ *Concatenate the heads*

---

reveals that the reconstruction can be solved optimally by applying the well-known Singular Value Decomposition.

**Theorem 3** (**Value-Output compression by SVD**). *If we search $\hat{\boldsymbol{W}}_V$ and $\hat{\boldsymbol{W}}_O$ over $\mathbb{R}^{d_h \times k}$ and $\mathbb{R}^{k \times d_h}$, respectively, the optimum in equation 6 is $\hat{\boldsymbol{W}}_V = \boldsymbol{C}^{-1/2} \boldsymbol{U}_k$ and $\hat{\boldsymbol{W}}_O = \boldsymbol{\Sigma} \boldsymbol{V}^\top$. Here, $\boldsymbol{U} \boldsymbol{\Sigma} \boldsymbol{V}^\top$ and $\boldsymbol{C} \triangleq \sum_{i=1}^{N} \boldsymbol{X}_i^\top \boldsymbol{X}_i$ are the SVD of $\boldsymbol{C}^{1/2} \boldsymbol{W}_V \boldsymbol{W}_O$ and input correlation, respectively. The corresponding Type-III reconstruction error in equation 6 is exactly the SVD approximation error, defined in Def. 1, relative to $\boldsymbol{C}^{\frac{1}{2}} \boldsymbol{W}_V \boldsymbol{W}_O$:*

$$V_{III} = \mathcal{E}_{SVD}^2 (\boldsymbol{C}^{\frac{1}{2}} \boldsymbol{W}_V \boldsymbol{W}_O). \tag{9}$$

Building on the established equivalence to SVD via Theorem 3, we introduce Algorithm 3. This algorithm guarantees the following:

**Proposition 3** (**Value-Output compression error**). *Denote $\sigma_i$ as the singular values, Algorithm 3 yields the optimal Type-III modular reconstruction error $V_{III} = \sum_{i=k+1}^{d} \sigma_i^2 (\boldsymbol{C}^{\frac{1}{2}} \boldsymbol{W}_V \boldsymbol{W}_O)$.*

### 3.3 GLOBAL SPARSITY ALLOCATION

While MoDeGPT modules are optimized locally, we propose a global optimization strategy that translates layer importance scores into sparsity allocations across layers. This strategy seeks to maximize the sum of importance scores, weighted by the parameters retained in each layer. To avoid the negative effects of excessive sparsity (Yin et al., 2023), we incorporate entropic regularization for smoothing. The formulation of this constrained optimization problem is as follows:

$$\max_{\phi_{1:L}} \sum_{i=1}^{L} s_i (1 - \phi_i) + \varepsilon H(\phi_i) \quad \text{such that} \quad \frac{1}{L} \sum_{i=1}^{L} \phi_i = \phi_{\text{avg}}, \quad 0 \le \phi_i \le 1, \tag{10}$$

where $\phi_i$ and $s_i$ represent the sparsity and importance score of layer $i$, respectively, and $\phi_{\text{avg}}$ denotes the overall target sparsity. For sufficiently large $\varepsilon$, the following theorem demonstrates that the optimal layer sparsity distribution can be easily computed as:

$$\phi = L \phi_{\text{avg}} \times \text{Softmax}(-\mathbf{s}/\varepsilon). \tag{11}$$

**Theorem 4.** *For sufficient large $\varepsilon$, (11) is the optimal sparsity allocation in the equation 10.*

In our implementations, we adopt the Block Influence (BI) score in Men et al. (2024), which is the negative correlation between a layer's input and output defined by: $s = 1 - \mathbb{E} \mathbf{x}_{\text{in}}^\top \mathbf{x}_{\text{out}} / \|\mathbf{x}_{\text{in}}\|_2 \|\mathbf{x}_{\text{out}}\|_2$.

## 4 EXPERIMENTS

### 4.1 SETUPS

**Models** We evaluated MoDeGPT on several models that employ a sequential transformer block structure: OPT (Zhang et al., 2022) across multiple scales (125M, 1.3B, 2.7B, 6.7B), LLAMA-1 at 7B, LLAMA-2 (Touvron et al., 2023) at 7B, 13B, 70B, and LLAMA-3 (AI@Meta, 2024) at 8B.

**Implementations and environments** We implemented our models using Hugging Face Transformers (Wolf et al., 2019), with correlation computations in FP64. Model compression and performance testing were conducted on a single NVIDIA A100 80GB GPU, except for the 70B model, which we used 8 A100 GPUs. Additional details are in Appendix B.2.

**Datasets** Following calibration setups similar to prior studies (Frantar et al., 2022; Ashkboos et al., 2024; Dettmers et al., 2023), we employed the WikiText-2 (Merity et al., 2016) and Alpaca datasets (Taori et al., 2023), each comprising 128 samples of 2048 characters. Zero-shot performance was evaluated using the LM Evaluation Harness (Gao et al., 2021), with task details provided in Appendix B.2.

**Baseline comparisons** We benchmarked our approach against several baselines. For non-gradient-based large language model pruning, we compared it with Uniform Pruning, Magnitude Pruning, SVD, SliceGPT (Ashkboos et al., 2024), ShortGPT (Men et al., 2024), SLEB (Song et al., 2024) and Optimal Brain Damage (LeCun et al., 1989). For methods involving backward propagation, our comparisons included LLM-Pruner (Ma et al., 2023) and LLM Surgeon (van der Ouderaa et al., 2023). Additionally, in Appendices B.4 and B.5, we evaluated our methods against feature-mimic compression techniques and SVD-based methods, respectively.

### 4.2 GENERATION PERFORMANCE

**Table 3:** Perplexity comparisons of structured pruning methods for LLAMA-2 7B and 13B on WikiText-2, calibrated with 128 sequences of 2048 tokens.

| Method | No Gradient | 7B (ppl: 5.12 ↓) | | | | | LLAMA-2 13B (ppl: 4.57 ↓) | | | | |
|---|---|---|---|---|---|---|---|---|---|---|---|
| | | 10% | 20% | 30% | 40% | 50% | 10% | 20% | 30% | 40% | 50% |
| K-OBD (LeCun et al., 1989) | ✗ | 5.48 | 9.14 | 15.43 | 28.03 | 46.64 | 4.91 | 6.29 | 10.08 | 13.06 | 16.06 |
| LLM-Pruner (Ma et al., 2023) | ✗ | 7.11 | 9.29 | 13.56 | 17.90 | 31.05 | 5.57 | 6.67 | 12.19 | 19.56 | 32.20 |
| LLM surgeon (van der Ouderaa et al., 2023) | ✗ | **5.25** | 6.18 | 7.83 | 10.39 | 15.38 | **4.69** | 5.29 | 6.21 | 7.25 | 9.43 |
| Uniform | ✓ | 19.09 | 27.13 | 46.25 | 176.24 | 327.99 | 13.78 | 18.18 | 29.05 | 45.44 | 82.60 |
| Magnitude | ✓ | 861.76 | 821.34 | 9623 | Overflow | Overflow | 22.41 | 320.39 | 723.60 | 2105 | 3004 |
| SVD | ✓ | Overflow | Overflow | 52719 | 51229 | Overflow | 7655 | 9919 | 21248 | 53672 | 39521 |
| ShortGPT (Men et al., 2024) | ✓ | 6.98 | 14.31 | 33.21 | 71.04 | 268.11 | 5.40 | 7.69 | 30.48 | 48.83 | 187.23 |
| SLEB (Song et al., 2024) | ✓ | 6.05 | 7.64 | 11.23 | 29.10 | 103.38 | 5.23 | 6.31 | 8.24 | 11.76 | 27.67 |
| SliceGPT (Ashkboos et al., 2024) | ✓ | 6.46 | 7.68 | 10.47 | 15.19 | 24.82 | 5.67 | 6.68 | 8.68 | 12.56 | 20.57 |
| MoDeGPT (ours) | ✓ | 5.48 | **6.16** | **7.51** | **8.41** | **11.88** | 4.83 | **5.29** | **6.10** | **6.95** | **8.95** |

We evaluated the generation performance of compressed LLAMA-2 models (7B and 13B) using the WikiText-2 test split in Table 3, 19 and B.3. Results for OPT and LLAMA-3 8B are included in Appendices B.1 and B.3. The table distinguishes between compression methods using gradients (top rows) and those without (bottom rows). Among non-gradient methods, the traditional matrix decomposition approach using SVD performed the worst. In sharp contrast, MoDeGPT outperformed all other baselines at various compression rates by jointly applying decomposition to multiple matrices within a module; it only increased the perplexity by 20% for 20% compression of the 7B model, which is substantially better than the next best alternative that saw a 50% increase. In comparison to gradient-based methods, MoDeGPT surpassed other structured compression techniques except for a low compression rate (20%). This demonstrates that using local reconstruction as a proxy for true loss can achieve state-of-the-art compression.

### 4.3 ZERO-SHOT PERFORMANCE

We evaluated our method on zero-shot tasks, comparing it to leading baselines in Table 4. Our method showed superior performance at higher compression rates. The bottom rows indicate that calibrating with the Alpaca dataset (instead of WikiText-2) significantly improved performance, with a 30% compression resulting in only a 10% accuracy drop. This effect was more pronounced for LLAMA-13B, as shown in Table 13 in Appendix B.3. We also tested the newer LLAMA-3 8B model, adapting our algorithm for grouped query attention head dependency as detailed in Appendix

**Table 4:** Zero-shot task performance of compressed LLAMA-2 7B and LLAMA-3 8B.

| Model | Compress. | Method | ARC-e | ARC-c | PIQA | WinoG. | HellaS. | Average |
|---|---|---|---|---|---|---|---|---|
| **LLAMA-2 7B** | 0% | Dense | 74.58 | 46.25 | 79.11 | 69.06 | 75.99 | 69.00 |
| | 30% | ShortGPT (Men et al., 2024) | 48.65 | 32.85 | 64.31 | 64.33 | 56.13 | 53.25 |
| | | SliceGPT (Ashkboos et al., 2024) | 58.88 | 33.36 | 68.55 | 58.01 | 49.86 | 53.73 |
| | | LLM surgeon (van der Ouderaa et al., 2023) | 63.09 | 36.69 | 73.56 | 61.09 | 60.72 | 59.03 |
| | | MoDeGPT (ours) | 63.26 | 38.73 | 70.40 | 67.32 | 63.26 | 60.78 |
| | | MoDeGPT-Alpaca (ours) | 65.49 | 39.16 | 73.34 | 66.22 | 65.90 | 62.02 |
| | 40% | ShortGPT (Men et al., 2024) | 41.16 | 29.94 | 60.12 | 60.46 | 43.67 | 47.07 |
| | | SliceGPT (Ashkboos et al., 2024) | 36.49 | 24.57 | 54.90 | 53.43 | 34.80 | 40.84 |
| | | LLM surgeon (van der Ouderaa et al., 2023) | 52.31 | 30.29 | 69.26 | 54.38 | 48.04 | 50.86 |
| | | MoDeGPT (ours) | 49.45 | 30.03 | 64.96 | 61.96 | 53.01 | 51.88 |
| | | MoDeGPT-Alpaca (ours) | 59.76 | 34.73 | 70.35 | 64.40 | 58.63 | 57.58 |
| **LLAMA-3 8B** | 0% | Dense | 77.69 | 53.58 | 80.63 | 72.69 | 79.16 | 72.75 |
| | 25% | ShortGPT-Alpaca (Men et al., 2024) | 38.13 | 31.40 | 60.94 | 54.22 | 31.52 | 43.24 |
| | | SliceGPT-Alpaca (Ashkboos et al., 2024) | 44.44 | 29.27 | 57.56 | 58.48 | 41.08 | 46.17 |
| | | MoDeGPT-Alpaca (ours) | 67.05 | 41.13 | 75.52 | 69.61 | 66.49 | 63.96 |

**Table 5:** Comparisons of 30% compression on LLAMA-2 70B using 128 wikitext-2 samples for calibration.

| Method | ↓WikiText-2 | ↑ARC-e | ARC-c | PIQA | WinoG. | HellaS. | BoolQ | OBQA | MathQA | MMLU-ml | COPA | Lamb. | ↑Average. |
|---|---|---|---|---|---|---|---|---|---|---|---|---|---|
| Dense LLAMA-2 70B | 3.12 | 80.98 | 57.25 | 82.75 | 77.90 | 83.83 | 83.79 | 48.80 | 38.42 | 42.86 | 94.00 | 79.60 | 70.02 |
| SliceGPT (Ashkboos et al., 2024) | 5.76 | 67.05 | 42.06 | 67.52 | 71.11 | 55.57 | 41.56 | 40.20 | 27.87 | 32.14 | 82.00 | 52.03 | 52.65 |
| ShortGPT (Men et al., 2024) | 66.33 | 60.65 | 34.47 | 72.74 | 64.01 | 63.80 | 66.88 | 34.40 | 23.05 | 31.25 | 75.00 | 27.01 | 48.06 |
| SLEB (Song et al., 2024) | 5.54 | 71.97 | 44.20 | 77.74 | 69.38 | 73.54 | 67.25 | 41.80 | 27.47 | 32.15 | 88.00 | 64.22 | 59.79 |
| MoDeGPT + OWL sparsity | 4.67 | 76.01 | 50.34 | 74.70 | 72.85 | 72.43 | 69.88 | 44.20 | 32.26 | 44.64 | 87.00 | 69.61 | 63.08 |
| MoDeGPT + our sparsity | 4.89 | 77.69 | 50.94 | 77.53 | 76.87 | 78.16 | 74.71 | 45.60 | 35.04 | 42.86 | 89.00 | 72.17 | 65.51 |
| MoDeGPT + our sparsity + Alpaca | 5.73 | 78.57 | 51.54 | 80.85 | 77.19 | 79.60 | 82.81 | 46.40 | 32.83 | 40.18 | 94.00 | 70.72 | 66.79 |

B.1. The performance gap between our method and baselines was notably larger with this model, aligning with quantization challenges observed in (Huang et al., 2024).

Finally, we compared our method against decomposition and layer-pruning baselines in a large-scale experiment on the LLAMA-2 70B, as shown in Table 5. Our method demonstrates improved performance in larger models, with minimal drops of 4.51% and 3.23% in zero-shot task performance at 30% compression. This is achieved using only 128 samples from WikiText-2 and Alpaca, respectively, for calibration, without requiring recovery fine-tuning. This result highlights the scalability and effectiveness of our approach in large models. On the middle two rows, we compared our sparsity allocation strategy with the recent state-of-the-art OWL method (Yin et al., 2023). While our method shows a slightly higher perplexity, it consistently achieves superior zero-shot performances. Appendix B.10 provides additional analysis on the comparisons with OWL.

**Table 6:** Compute time.

| Model | MoDeGPT (ours) | | LLM surgeon | |
|---|---|---|---|---|
| | Time | GPUs | Time | GPUs |
| LLAMA-2 7B | 4h09m | 1xA100 | 17h08m | 4xH100 |
| LLAMA-2 13B | 8h26m | 1xA100 | 1d9h26m | 8xH100 |

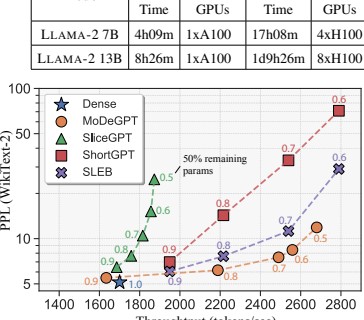

**Figure 3:** PPL vs. throughput.

### 4.4 COMPUTATION AND THROUGHPUT

Table 6 compares the compression costs of MoDeGPT with LLM surgeon, the best-performing prior work. Given that the hourly rate for H100 is about twice that of A100 (Lambda, 2024), MoDeGPT offers significant cost savings—97% for the 7B model and 98.5% for the 13B model. Next, we analyze the trade-off between model performance, measured in perplexity (on WikiText-2), and throughput (tokens/sec), as illustrated in Figure 3. For this experiment, we set the sequence length to 256 and measured the average generation time of LLAMA-2 7B on a single A100 GPU with a batch size of 256. As illustrated, MoDeGPT consistently outperforms other models in both speedup and accuracy across various compression sizes, with size ratios annotated alongside each point. Remarkably, at 50% compression, MoDeGPT increases throughput by 58% while maintaining per-

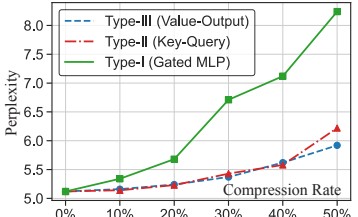

| Type | I | II | III |
|---|---|---|---|
| **Parameters** | MLP | K-Q | V-O |
| **Method** | Nyström | CR | SVD |
| **Size Ratio** | 66.84% | 16.58% | 16.58% |
| **Complexity** | $O(d_{int}^3)$ | $O(d_h^3/H^2)$ | $O(Hd_h^3)$ |
| **Effective $r$** | 0.094 | 0.121 | 0.095 |
| **Time** | 1h13m | 0h36m | 2h26m |

| Block | LLAMA-7B (model size: 13.81 GiB) | |
|---|---|---|
| | **Peak Memory** (GiB) | **GPU** hours |
| **MHA** | 15.54 (+11.5%) | 2h52m |
| **MLP** | 23.33 (+68.9%) | 1h13m |

**Figure 4:** Module-wise compression. **Table 7:** Module breakdown statistics. **Table 8:** Memory utilizations.

plexity on par with the top competitor at 30% compression. Further details on speedup, including different batch sizes and hardware setups, are in Appendix B.16."

### 4.5 ABLATION STUDY

We first analyzed the impact of compression on each module type within the LLAMA-2 7B model using a single A100 GPU. As shown in Figure 4 , the majority of perplexity degradation occurs when the MLP module is compressed. However, after normalizing by parameter size, i.e., the effective ratio $r$ in Table 7, it becomes evident that the Type-II module is the most sensitive to compression. This observation aligns with our theoretical analysis, which demonstrates that Type-II has the weakest error bounds, as it is constrained by the complete spectrum rather than just the residuals (see Propositions 1, 2, 3). In the middle, Table 7 provides a detailed breakdown of the module-wise compression statistics. Notably, the SVD method dominates the compression time for the value-output components, suggesting that techniques such as SVD approximation (Yuan et al., 2023) have the potential to reduce overall compression time. Meanwhile, Table 8 reports memory usage, showing that MLP compression requires the most memory, as it has the largest correlation dimension among the modules. Despite this, all compression tasks only consumed up to 23 GiB of memory, which is approximately double the model size. A similar memory consumption pattern for the 13B model is discussed in Appendix B.9.

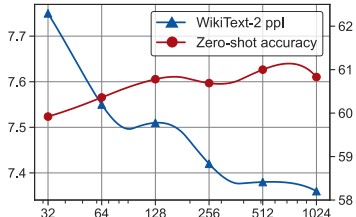

**Figure 5:** Impact of calibration size.

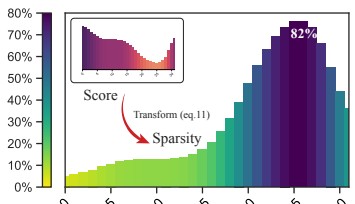

**Figure 6:** Score-to-sparsity conversion.

**Table 9:** Sparsity allocation.

| | Perplexity↓ | Zero-Shot Acc.↑ |
|---|---|---|
| Dense | 5.12 | 69.00% |
| Uniform | 9.06 | 53.47% |
| Sparsity Alloc. | 7.51 | 60.78% |

Second, we explored the effects of calibration size on a 30% compressed LLAMA-2 7B model. As shown in Figure 5, increasing the calibration size initially boosts performance; however, the gains in zero-shot performance diminish for sizes larger than 128.

Lastly, we evaluated sparsity allocation effects on the same model. Figure 6 shows the score-to-sparsity mapping from Section 3.3 with $\varepsilon = 0.1$, highlighting areas of higher sparsity in darker shades. Our findings indicate that certain layers, such as layer 26, can forgo up to 82% of parameters with minimal accuracy loss. Table 9 demonstrates that our global sparsity allocation significantly surpasses a uniform approach, affirming the efficacy of our decomposition method with a simple scoring function for sparsity distribution.

## 5 CONCLUSION

In this work, we introduced **MoDeGPT**, a novel structured compression method that generalizes matrix decomposition to the modular level, achieving state-of-the-art results for structured model compression via low-rank decomposition. Our approach has a strong theoretical grounding, offering bounds on the reconstruction errors for the components in the Transformers. Furthermore, MoDeGPT stands out by relying solely on forward propagation to achieve comparable or better compression performance to methods that use the gradients from backward propagation. We believe our novel methods and findings will inspire more theoretical and algorithmic innovations for training-free model compression.

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

<p align="center">SUPPLEMENTARY MATERIAL</p>

CONTENTS

## A    PROOFS

This section provides proofs for the theorems and propositions in the main text, along with definitions and assumptions for formalism.

First, we define the following notation.

**Definition 2** (Column Selection Matrix)**.** *A $k$-column selection matrix $\boldsymbol{S}_k$ is a matrix with $k$ columns where each column has a single non-zero element indicating the selected index. For example, $\boldsymbol{S}_3 = [[0, 0, 1, 0]^\top, [0, 1, 0, 0]^\top, [0, 0, 0, 1]^\top]$ is a 3-column selection matrix selecting the third, second, and fourth columns. An important property is that any matrix right-multiplied by a column selection matrix will result in a matrix consisting of the selected columns.*

Next, we make an assumption regarding the nonlinear functions used in all modules, which is crucial for validating our algorithms.

**Assumption 1.** *Any column selection matrix $\boldsymbol{S}$ is commutative with the nonlinear functions under consideration. Specifically, the function $\sigma$ satisfies the property that $\sigma(\boldsymbol{X})\boldsymbol{S} = \sigma(\boldsymbol{X}\boldsymbol{S})$ for any $\boldsymbol{X}$ and any column selection matrix $\boldsymbol{S}$.*

Importantly, Assumption 1 is met by any activation function that operates element-wise on the inputs, as well as by widely used embedding functions, such as the rotary positional embedding (Su et al., 2024).

### A.1    PROOF OF THEOREM 1 AND PROPOSITION 1: MLP COMPRESSION WITH NYSTRÖM APPROXIMATION

**Theorem 1** (**MLP compression can be solved by Nyström approximation**)**.** *Let $\hat{\boldsymbol{W}}_U$ be searched over the matrix multiplication form $\boldsymbol{W}_U \boldsymbol{S}_k$, where $\boldsymbol{S}_k$ is a $k$-column selection matrix, and let $\hat{\boldsymbol{W}}_D$ be searched over $\mathbb{R}^{k \times d_h}$. The optimal $\hat{\boldsymbol{W}}_D^*$ can then be expressed as: $(\boldsymbol{S}_k^\top \boldsymbol{C}_\sigma \boldsymbol{S}_k)^\dagger \boldsymbol{S}_k^\top \boldsymbol{C}_\sigma \boldsymbol{W}_D$. Using $\boldsymbol{W}_U \boldsymbol{S}_k$ and $\hat{\boldsymbol{W}}_D^*$ as the compressed matrices, the Type-I reconstruction error in equation 6 satisfies:*

$$V_I \le \|\boldsymbol{W}_D\|_2^2 \, \mathcal{E}_{Nys}^2(\boldsymbol{C}_\sigma^{\frac{1}{2}}), \tag{12}$$

*where $\mathcal{E}_{Nys}(\boldsymbol{C}_\sigma^{\frac{1}{2}})$ denotes the Nyström approximation error, defined in Def. 1, relative to the activation correlation matrix $\boldsymbol{C}_\sigma \triangleq \sum_{i=1}^N \sigma(\boldsymbol{X}_i \boldsymbol{W}_U)^\top \sigma(\boldsymbol{X}_i \boldsymbol{W}_U)$, using the same $\boldsymbol{S}_k$ in the compression of $\boldsymbol{W}_U$.*

*Proof.* Ideally, we want to seek low-rank matrices $\boldsymbol{W}_U$, $\boldsymbol{W}_D$ without any constraint; however, the nonlinearity between the two matrices makes the optimal solution intractable. To overcome the nonlinearity between the two matrices, we instead restrict the compressed up matrix $\hat{\boldsymbol{W}}_U$ to be of the form $\boldsymbol{W}_U \boldsymbol{S}_k$, $\boldsymbol{S}_k \in \mathbb{R}^{d \times k}$ is a $k$-column selection matrix and $\hat{\boldsymbol{W}}_D$ is a general $\mathbb{R}^{k \times d}$ matrix. Plug in this form, we can simplify equation 6 as

$$\min_{\boldsymbol{S}_k, \hat{\boldsymbol{W}}_D} \sum_{i=1}^N \|f(\boldsymbol{X}_i) - \sigma(\boldsymbol{X}_i \boldsymbol{W}_U \boldsymbol{S}_k) \hat{\boldsymbol{W}}_D\|_F^2$$

$$\stackrel{(a)}{=} \min_{\boldsymbol{S}_k, \hat{\boldsymbol{W}}_D} \sum_{i=1}^N \|\sigma(\boldsymbol{X}_i \boldsymbol{W}_U) \boldsymbol{W}_D - \sigma(\boldsymbol{X}_i \boldsymbol{W}_U) \boldsymbol{S}_k \hat{\boldsymbol{W}}_D\|_F^2$$

$$= \min_{\boldsymbol{S}_k, \hat{\boldsymbol{W}}_D} \mathrm{Tr}\left( \sum_{i=1}^N \sigma(\boldsymbol{X}_i \boldsymbol{W}_U)^\top \sigma(\boldsymbol{X}_i \boldsymbol{W}_U) \left( \boldsymbol{W}_D - \boldsymbol{S}_k \hat{\boldsymbol{W}}_D \right) \left( \boldsymbol{W}_D - \boldsymbol{S}_k \hat{\boldsymbol{W}}_D \right)^\top \right)$$

$$= \min_{\boldsymbol{S}_k, \hat{\boldsymbol{W}}_D} \| \boldsymbol{C}_\sigma^{\frac{1}{2}} \left( \boldsymbol{W}_D - \boldsymbol{S}_k \hat{\boldsymbol{W}}_D \right) \|_F^2, \tag{13}$$

where $\boldsymbol{C}_\sigma$ is the empirical correlation matrix of latent features $\boldsymbol{C}_\sigma = \sum_{i=1}^N \sigma(\boldsymbol{X}_i \boldsymbol{W}_U)^\top \sigma(\boldsymbol{X}_i \boldsymbol{W}_U)$, and (a) follows from Assumption 1. Setting the gradient of the last expression with respect to $\hat{\boldsymbol{W}}_D$

to zero, we obtain the optimal down matrix $\hat{\boldsymbol{W}}_D^* = \left(\boldsymbol{S}_k^\top \boldsymbol{C}_\sigma \boldsymbol{S}_k\right)^\dagger \boldsymbol{S}_k^\top \boldsymbol{C}_\sigma \boldsymbol{W}_D$. After Plugging this back to the objective, we can further simplify the objective into,

$$\min_{\boldsymbol{S}_k} \left\| \left( \boldsymbol{C}_\sigma^{\frac{1}{2}} - \boldsymbol{C}_\sigma^{\frac{1}{2}} \boldsymbol{S}_k \left( \boldsymbol{S}_k^\top \boldsymbol{C}_\sigma \boldsymbol{S}_k \right)^\dagger \boldsymbol{S}_k^\top \boldsymbol{C}_\sigma \right) \boldsymbol{W}_D \right\|_F^2$$

$$\leq \|\boldsymbol{W}_D\|_2^2 \|\boldsymbol{C}_\sigma^{-\frac{1}{2}}\|_2^2 \min_{\boldsymbol{S}_k} \left\| \boldsymbol{C}_\sigma - \boldsymbol{C}_\sigma \boldsymbol{S}_k \left( \boldsymbol{S}_k^\top \boldsymbol{C}_\sigma \boldsymbol{S}_k \right)^\dagger \boldsymbol{S}_k^\top \boldsymbol{C}_\sigma \right\|_F^2 = \|\boldsymbol{W}_D\|_2^2 \|\boldsymbol{C}_\sigma^{-1}\|_2 \mathcal{E}_{\text{Nys}}^2(\boldsymbol{C}_\sigma). \tag{14}$$

Now, observe that the error on the right side of equation 14 is proportional to the Nyström matrix approximation to the matrix $\boldsymbol{C}_\sigma$ in Definition 1. Hence, the variable $\boldsymbol{S}_k$ can be optimized with any Nyström approximation algorithm (Gittens & Mahoney, 2013). In this work, we adapt a deterministic Nyström algorithm, Algorithm 1, that has the theoretical guarantee proved in the next proposition. $\qquad \square$

**Proposition 1.** *Suppose that the rank $k$ and the scores $s_i$ in Algorithm 1 are chosen such that there exists an error $\varepsilon > 0$ satisfying $\varepsilon \geq \sum_{i=k+1}^{d_{int}} s_i$, then the Type-I modular reconstruction error in equation 6 is bounded by $V_I \leq \|\boldsymbol{W}_D\|_2^2 \|\boldsymbol{C}_\sigma^{-1}\|_2 \frac{\varepsilon^2 d_{int}^2}{k^2(1-\varepsilon)^2} \sum_{i=k+1}^{d_{int}} \sigma_i^2(\boldsymbol{C}_\sigma)$, where $d_{int}$ and $\sigma_i$ denote the intermediate dimension (i.e., the input dimension of $W_D$) and singular values, respectively.*

*Proof.* Since our column selection is equivalent to applying the deterministic ridge leverage score sampling (DRLS) to $\boldsymbol{C}_\sigma^{\frac{1}{2}}$ (McCurdy, 2018), Theorem 1 in McCurdy (2018) implies that

$$(1-\varepsilon)\boldsymbol{C}_\sigma - \frac{\epsilon}{k} \left\| (\boldsymbol{C}_\sigma^{\frac{1}{2}})_{\backslash k} \right\|_F^2 \boldsymbol{I} \preceq \boldsymbol{C}_\sigma^{\frac{1}{2}} \boldsymbol{S}_k \boldsymbol{S}_k^\top \boldsymbol{C}_\sigma^{\frac{1}{2}} \preceq \boldsymbol{C}_\sigma \tag{15}$$

$$\Rightarrow \boldsymbol{C}_\sigma \preceq \frac{\epsilon}{k(1-\varepsilon)} \left\| (\boldsymbol{C}_\sigma^{\frac{1}{2}})_{\backslash k} \right\|_F^2 \boldsymbol{I} + \frac{1}{1-\varepsilon} \boldsymbol{C}_\sigma^{\frac{1}{2}} \boldsymbol{S}_k \boldsymbol{S}_k^\top \boldsymbol{C}_\sigma^{\frac{1}{2}}. \tag{16}$$

Next, we define $\boldsymbol{P} = \boldsymbol{C}_\sigma^{\frac{1}{2}} \boldsymbol{S}_k (\boldsymbol{S}_k^\top \boldsymbol{C}_\sigma \boldsymbol{S}_k)^\dagger \boldsymbol{S}_k^\top \boldsymbol{C}_\sigma^{\frac{1}{2}}$. We note that $\boldsymbol{P}$ is the projection matrix of the column space of $\boldsymbol{C}_\sigma^{\frac{1}{2}} \boldsymbol{S}$. Now, we multiply $\boldsymbol{I} - \boldsymbol{P}$ to both sides in the previous inequality to get

$$(\boldsymbol{I} - \boldsymbol{P})\boldsymbol{C}_\sigma(\boldsymbol{I} - \boldsymbol{P}) \tag{17}$$

$$\preceq \frac{\epsilon}{k(1-\varepsilon)} \left\| (\boldsymbol{C}_\sigma^{\frac{1}{2}})_{\backslash k} \right\|_F^2 (\boldsymbol{I} - \boldsymbol{P}) + \frac{1}{1-\varepsilon}(\boldsymbol{I} - P)\boldsymbol{C}_\sigma^{\frac{1}{2}} \boldsymbol{S}_k \boldsymbol{S}_k^\top \boldsymbol{C}_\sigma^{\frac{1}{2}}(\boldsymbol{I} - \boldsymbol{P}) \tag{18}$$

$$\preceq \frac{\epsilon}{k(1-\varepsilon)} \left\| (\boldsymbol{C}_\sigma^{\frac{1}{2}})_{\backslash k} \right\|_F^2 \boldsymbol{I}, \tag{19}$$

where in the last inequality we use the fact that $\boldsymbol{I} - \boldsymbol{P} \preceq \boldsymbol{I}$ and that $\boldsymbol{I} - \boldsymbol{P}$ is the orthogonal projection to the orthogonal complement of the column space $\boldsymbol{C}_\sigma^{\frac{1}{2}} \boldsymbol{S}$ so that $\boldsymbol{S}^\top \boldsymbol{C}_\sigma^{\frac{1}{2}}(\boldsymbol{I} - \boldsymbol{P}) = \boldsymbol{0}$. Now, we have

$$\|(\boldsymbol{I} - \boldsymbol{P})\boldsymbol{C}_\sigma^{\frac{1}{2}} \boldsymbol{C}_\sigma^{\frac{1}{2}}(\boldsymbol{I} - \boldsymbol{P})\|_2 \leq \frac{\varepsilon}{k(1-\varepsilon)} \left\| (\boldsymbol{C}_\sigma^{\frac{1}{2}})_{\backslash k} \right\|_F^2 = \frac{\varepsilon}{k(1-\varepsilon)} \sum_{i=k+1}^{d_{\text{int}}} \sigma_i(\boldsymbol{C}_\sigma) \tag{20}$$

$$\Rightarrow \|\boldsymbol{C}_\sigma^{\frac{1}{2}}(\boldsymbol{I} - \boldsymbol{P})^2 \boldsymbol{C}_\sigma^{\frac{1}{2}}\|_2 = \|\boldsymbol{C}_\sigma^{\frac{1}{2}}(\boldsymbol{I} - \boldsymbol{P})\boldsymbol{C}_\sigma^{\frac{1}{2}}\|_2 \leq \frac{\varepsilon}{k(1-\varepsilon)} \sum_{i=k+1}^{d_{\text{int}}} \sigma_i(\boldsymbol{C}_\sigma). \tag{21}$$

Since $\boldsymbol{C}_\sigma^{\frac{1}{2}} \boldsymbol{P} \boldsymbol{C}_\sigma^{\frac{1}{2}} = \boldsymbol{C}_\sigma \boldsymbol{S}_k \left( \boldsymbol{S}_k^\top \boldsymbol{C}_\sigma \boldsymbol{S}_k \right)^\dagger \boldsymbol{S}_k^\top \boldsymbol{C}_\sigma$, the inequality is equivalent to

$$\|\boldsymbol{C}_\sigma - \boldsymbol{C}_\sigma \boldsymbol{S} \left( \boldsymbol{S}^\top \boldsymbol{C}_\sigma \boldsymbol{S} \right)^\dagger \boldsymbol{S}^\top \boldsymbol{C}_\sigma\|_2 \leq \frac{\varepsilon}{k(1-\varepsilon)} \sum_{i=k+1}^{d_{\text{int}}} \sigma_i(\boldsymbol{C}_\sigma). \tag{22}$$

Finally, we complete the proof by,

$$
\begin{aligned}
\mathcal{E}_{\text{Nys}}^2(\boldsymbol{C}_\sigma) &\overset{(a)}{\leq} d_{\text{int}}\|\boldsymbol{C}_\sigma - \boldsymbol{C}_\sigma \boldsymbol{S}\left(\boldsymbol{S}^\top \boldsymbol{C}_\sigma \boldsymbol{S}\right)^\dagger \boldsymbol{S}^\top \boldsymbol{C}_\sigma\|_2^2 \\
&\overset{(b)}{\leq} \frac{\varepsilon^2 d_{int}}{k^2(1-\varepsilon)^2}\left(\sum_{i=k+1}^{d_{int}} \sigma_i(\boldsymbol{C}_\sigma)\right)^2 \\
&\overset{(c)}{\leq} \frac{\varepsilon^2 d_{int}^2}{k^2(1-\varepsilon)^2}\sum_{i=k+1}^{d_{int}} \sigma_i^2(\boldsymbol{C}_\sigma),
\end{aligned}
\tag{23}
$$

where (a) follows from that$\|\boldsymbol{A}\|_F \leq \sqrt{d}\|\boldsymbol{A}\|_2$ for any matrix $\boldsymbol{A} \in \mathbb{R}^{d\times d}$, (b) from equation 22, and (c) from Cauchy inequality that $\left(\sum_{i=1}^n x_i\right)^2 \leq n\sum_{i=1}^n x_i^2$ for any sequence $\{x_i\}_i$. $\qquad\square$

### A.2 Proof of Theorem 2 and Proposition 2: Key-Query Compression with CR Approximation

**Theorem 2** (**Key-Query compression can be solved by CR approximation**). *Let the compressed $\hat{\boldsymbol{W}}_Q, \hat{\boldsymbol{W}}_K$ to be the form of $\boldsymbol{W}_Q \boldsymbol{S}_k, \boldsymbol{W}_K \boldsymbol{S}_k$, then Type-II reconstruction error in equation 6 has*

$$
V_{II} \leq \mathcal{E}_{CR}^2(\boldsymbol{C}_K^{\frac{1}{2}}\boldsymbol{C}_Q^{\frac{1}{2}}),
\tag{24}
$$

*where $\mathcal{E}_{CR}$ denotes the CR approximation error, defined in Def. 1, relative to $\boldsymbol{C}_Q^{1/2}\boldsymbol{C}_K^{1/2}$, utilizing the same $\boldsymbol{S}_k$ in the compression. Here, the matrices $\boldsymbol{C}_Q \triangleq \sum_{i=1}^N \sigma(\boldsymbol{X}_i\boldsymbol{W}_Q)^\top \sigma(\boldsymbol{X}_i\boldsymbol{W}_Q)$ and $\boldsymbol{C}_K \triangleq \sum_{i=1}^N \sigma(\boldsymbol{X}_i\boldsymbol{W}_K)^\top \sigma(\boldsymbol{X}_i\boldsymbol{W}_K)$ denote the correlation matrices of query and key states, respectively.*

*Proof.* Regarding two nonlinear functions satisfying Assumption 1, we propose to optimize the reconstruction error with compressed key query matrices of the form $\boldsymbol{W}_K \boldsymbol{S}_k, \boldsymbol{W}_Q \boldsymbol{S}_k$, where $\boldsymbol{S}_k$ is some column selection matrix. Now the reconstruction error of this module is

$$
\begin{aligned}
&\sum_{i=1}^N \|f(\boldsymbol{X}_i) - \sigma_r(\boldsymbol{X}_i\boldsymbol{W}_Q\boldsymbol{S}_k)\sigma_r^\top(\boldsymbol{X}_i\boldsymbol{W}_K\boldsymbol{S}_k)\|_F^2 \\
&\overset{(a)}{=} \sum_{i=1}^N \|\sigma_r(\boldsymbol{X}_i\boldsymbol{W}_Q)\left(\boldsymbol{I} - \boldsymbol{S}_k\boldsymbol{S}_k^\top\right)\sigma_r^\top(\boldsymbol{X}_i\boldsymbol{W}_K)\|_F^2 \\
&= \sum_{i=1}^N \text{Tr}\left(\left(\boldsymbol{I} - \boldsymbol{S}_k\boldsymbol{S}_k^\top\right)\sigma_r(\boldsymbol{X}_i\boldsymbol{W}_Q)^\top \sigma_r(\boldsymbol{X}_i\boldsymbol{W}_Q)\left(\boldsymbol{I} - \boldsymbol{S}_k\boldsymbol{S}_k^\top\right)\sigma_r(\boldsymbol{X}_i\boldsymbol{W}_K)^\top \sigma_r(\boldsymbol{X}_i\boldsymbol{W}_K)\right) \\
&\overset{(b)}{\leq} \text{Tr}\left(\sum_{i=1}^N\left(\boldsymbol{I} - \boldsymbol{S}_k\boldsymbol{S}_k^\top\right)\sigma_r(\boldsymbol{X}_i\boldsymbol{W}_Q)^\top \sigma_r(\boldsymbol{X}_i\boldsymbol{W}_Q)\left(\boldsymbol{I} - \boldsymbol{S}_k\boldsymbol{S}_k^\top\right)\sum_{j=1}^N \sigma_r(\boldsymbol{X}_j\boldsymbol{W}_K)^\top \sigma_r(\boldsymbol{X}_j\boldsymbol{W}_K)\right) \\
&\overset{(c)}{\leq} \text{Tr}\left(\boldsymbol{C}_K\left(\boldsymbol{I} - \boldsymbol{S}_k\boldsymbol{S}_k^\top\right)\boldsymbol{C}_Q\right) = \|\boldsymbol{C}_K^{\frac{1}{2}}\boldsymbol{C}_Q^{\frac{1}{2}} - \boldsymbol{C}_K^{\frac{1}{2}}\boldsymbol{S}_k\boldsymbol{S}_k^\top\boldsymbol{C}_Q^{\frac{1}{2}}\|_F^2 = \mathcal{E}_{CR}^2(\boldsymbol{C}_K^{\frac{1}{2}}\boldsymbol{C}_Q^{\frac{1}{2}}),
\end{aligned}
\tag{25}
$$

where $\boldsymbol{C}_K = \sum_{i=1}^N \sigma(\boldsymbol{X}_i\boldsymbol{W}_Q\boldsymbol{S}_k)^\top \sigma(\boldsymbol{X}_i\boldsymbol{W}_Q\boldsymbol{S}_k)$, $\boldsymbol{C}_Q = \sum_{i=1}^N \sigma(\boldsymbol{X}_i\boldsymbol{W}_K\boldsymbol{S}_k)^\top \sigma(\boldsymbol{X}_i\boldsymbol{W}_K\boldsymbol{S}_k)$ are the correlation matrices associated with the outputs of $\boldsymbol{W}_Q$ and $\boldsymbol{W}_K$, respectively. Here, (a) follows from Assumption 1, (b) follows from that $\left(\boldsymbol{I} - \boldsymbol{S}_k\boldsymbol{S}_k^\top\right)\sigma_r(\boldsymbol{X}_i\boldsymbol{W}_Q)^\top \sigma_r(\boldsymbol{X}_i\boldsymbol{W}_Q)\left(\boldsymbol{I} - \boldsymbol{S}_k\boldsymbol{S}_k^\top\right)$ and $\sigma_r(\boldsymbol{X}_j\boldsymbol{W}_K)^\top \sigma_r(\boldsymbol{X}_j\boldsymbol{W}_K)$ are positive semidefinite, and (c) follows from that $\boldsymbol{I} - \boldsymbol{S}_k\boldsymbol{S}_k^\top \preceq \boldsymbol{I}$. From the last expression, we observe that the reconstruction is bounded by the CR approximation (Drineas et al., 2006) to the matrix-product $\boldsymbol{C}_K^{\frac{1}{2}}\boldsymbol{C}_Q^{\frac{1}{2}}$. $\qquad\square$

**Proposition 2.** *If we adopt Algorithm 2 then Type-II modular reconstruction error is bounded by $V_{II} \leq \left(\frac{d_h-k}{d_h}\right)^2\left(\sum_{i=1}^{d_h}\sigma_i(\boldsymbol{C}_K)\right)\left(\sum_{i=1}^{d_h}\sigma_i(\boldsymbol{C}_Q)\right)$, where $\sigma_i$ denotes the singular values.*

*Proof.* Our Algorithm 2 is a deterministic variant of Drineas et al. (2006). Recall that

$$\mathcal{E}_{\mathrm{CR}}(C_K^{\frac{1}{2}}C_Q^{\frac{1}{2}}) = \|C_k^{\frac{1}{2}}C_Q^{\frac{1}{2}} - C_K^{\frac{1}{2}}S_k S_k^\top C_Q^{\frac{1}{2}}\|_F = \|\sum_{i=k+1}^{d} k_i q_i^\top\|_F, \tag{26}$$

where $k_i$ and $q_i$ are the $i$-th column and $i$-th row of $C_k^{\frac{1}{2}}$ and $C_q^{\frac{1}{2}}$, respectively. Then,

$$\|\sum_{i=k+1}^{d} k_i q_i^\top\|_F \le \sum_{i=k+1}^{d} \|k_i\|_2 \|q_i\|_2 \overset{(a)}{\le} \sqrt{\left(\sum_{i=k+1}^{d} \|k_i\|_2^2\right)\left(\sum_{i=k+1}^{d} \|q_i\|_2^2\right)} \tag{27}$$

$$\overset{(b)}{\le} \frac{d-k}{d}\sqrt{\left(\sum_{i=1}^{d} \|k_i\|_2^2\right)\left(\sum_{i=1}^{d} \|q_i\|_2^2\right)} = \frac{d-k}{d}\|C_K^{\frac{1}{2}}\|_F \|C_Q^{\frac{1}{2}}\|_F \tag{28}$$

$$= \frac{d-k}{d}\sqrt{\left(\sum_{i=1}^{d} \sigma_i(C_K)\right)\left(\sum_{i=1}^{d} \sigma_i(C_Q)\right)}, \tag{29}$$

where in (a) we use Cauchy-Schwartz inequality and in (b) we use the fact that the column selection is based on the norm product in Algorithm 2. $\square$

### A.3 PROOF OF THEOREM 3 AND PROPOSITION 3: VALUE-OUTPUT COMPRESSION WITH SVD

**Theorem 3** (**Value-Output compression can be solved by SVD**). *If we search $\hat{W}_V$ and $\hat{W}_O$ over $\mathbb{R}^{d_h \times k}$ and $\mathbb{R}^{k \times d_h}$, respectively, the optimum in equation 6 is $\hat{W}_V = C^{-\frac{1}{2}}U_k$ and $\hat{W}_O = \Sigma V^\top$. Here, $U\Sigma V^\top$ and $C \triangleq \sum_{i=1}^{N} X_i^\top X_i$ are the SVD of $C^{\frac{1}{2}}W_V W_O$ and input correlation matrix, respectively. The corresponding Type-III reconstruction error in equation 6 is the SVD approximation error, defined in Def. 1, relative to $C^{\frac{1}{2}}W_V W_O$:*

$$V_{III} = \mathcal{E}_{SVD}^2(C^{\frac{1}{2}}W_V W_O). \tag{30}$$

*Proof.* $\hat{W}_V \in \mathbb{R}^{d \times k}$ and $\hat{W}_O \in \mathbb{R}^{k \times d}$. Plug in $\hat{f}(X) = X\hat{W}_V \hat{W}_O$ into equation 6 and simplify yields the objective

$$\min_{\hat{W}_V, \hat{W}_O} \sum_{i=1}^{N} \mathrm{Tr}\left(X_i^\top X_i (W_V W_O - \hat{W}_V \hat{W}_O)(W_V W_O - \hat{W}_V \hat{W}_O)^\top\right)$$

$$= \min_{\hat{W}_V, \hat{W}_O} \|C^{\frac{1}{2}}W_V W_O - C^{\frac{1}{2}}\hat{W}_V \hat{W}_O\|_F^2 = \mathcal{E}_{SVD}^2(C^{\frac{1}{2}}W_V W_O), \tag{31}$$

where $C = \sum_{i=1}^{N} X_i^\top X_i$ is the input correlation matrix. $\square$

**Proposition 3.** *Denote $\sigma_i$ as the singular values, Algorithm 3 yields the optimal Type-III modular reconstruction error $V_{III} = \sum_{i=k+1}^{d} \sigma_i^2(C^{\frac{1}{2}}W_V W_O)$.*

*Proof.* As $C^{\frac{1}{2}}\hat{W}_V \hat{W}_O$ has low rank $k$, this reconstruction error is upper bounded by the residue of the spectrum of the matrix $C^{\frac{1}{2}}W_V W_O$, i.e., $\mathcal{E}_{SVD} \le \sqrt{\sum_{i=k+1}^{d} \sigma_i^2(C^{\frac{1}{2}}W_V W_O)}$. In fact, the upper bound is achievable by Algorithm 2 since $C^{\frac{1}{2}}\hat{W}_V \hat{W}_O = U_k \Sigma_k V_k^\top$, which is the optimal rank $k$ approximation to the matrix $C^{\frac{1}{2}}W_V W_O$. $\square$

### A.4 PROOF OF THEOREM 4: GLOBAL SPARSITY ALLOCATION

**Theorem 4.** *For sufficient large $\varepsilon$, (11) is the optimal sparsity allocation in the equation 10.*

*Proof.* Consider the relaxed optimization problem

$$\max_{\phi_{1:L}} \sum_{i=1}^{L} s_i(1 - \phi_i) + \varepsilon H(\phi_i) \quad \text{s.t.} \quad \frac{1}{L} \sum_{i=1}^{L} \phi_i = \phi_{\text{avg}}. \tag{32}$$

Its associated Lagrangian is

$$\mathcal{L}(\phi_{1:L}, \lambda) = \sum_{i=1}^{L} s_i(1 - \phi_i) + \varepsilon H(\phi_i) + \lambda \left( \frac{1}{L} \sum_{i=1}^{L} \phi_i - \phi_{\text{avg}} \right). \tag{33}$$

To find the optimum, we set the gradient of the Lagrangian to zero, which yields

$$0 = \nabla_{\phi} \mathcal{L}(\phi_{1:L}, \lambda) = \nabla_{\phi} \left( \sum_{i=1}^{L} s_i(1 - \phi_i) - \varepsilon H(\phi_i) i \right) + \lambda \nabla_{\phi} \left( \frac{1}{L} \sum_{i=1}^{L} \phi_i - \phi_{\text{avg}} \right) \tag{34}$$

$$= \nabla_{\phi} \left( \sum_{i=1}^{L} s_i(1 - \phi_i) - \varepsilon \sum_{i=1}^{L} \phi_i \log \phi_i \right) + \lambda \nabla_{\phi} \left( \frac{1}{L} \sum_{i=1}^{L} \phi_i - \phi_{\text{avg}} \right). \tag{35}$$

This is equivalent to that, for any $j = 1, \ldots, L$,

$$0 = \partial_{\phi_j} \left( \sum_{i=1}^{L} s_i(1 - \phi_i) - \varepsilon \sum_{i=1}^{L} \phi_i \log \phi_i \right) + \lambda \partial_{\phi_j} \left( \frac{1}{L} \sum_{i=1}^{L} \phi_i - \phi_{\text{avg}} \right) \tag{36}$$

$$= -s_j - \varepsilon \log \phi_j - \varepsilon + \lambda \frac{1}{L}. \tag{37}$$

After rearrangement, we have $\phi_j = C \exp(-s_j/\varepsilon)$ for some constant $C$. On the other hand, $\phi_j$ satisfies the constraint of $\frac{1}{L} \sum_{i=1}^{L} \phi_i = \phi_{\text{avg}}$, which implies

$$\sum_{j=1}^{L} C \exp(-s_j/\varepsilon) = L\phi_{\text{avg}} \tag{38}$$

$$\Rightarrow \quad C = L\phi_{\text{avg}} / \sum_{j=1}^{L} \exp(-s_j/\varepsilon) \tag{39}$$

$$\Rightarrow \quad \phi_i = L\phi_{\text{avg}} \exp(-s_i/\varepsilon) / \sum_{j=1}^{L} \exp(-s_j/\varepsilon). \tag{40}$$

Finally, we must verify that for any $i = 1, \ldots, L$, the above expression of $\phi_i$ is a valid sparsity allocation, satisfying $\phi_i \leq 1$ for sufficiently large $\varepsilon$, to ensure it is also the optimum solution to the original optimization problem in equation 10. Since $\phi_i = L\phi_{\text{avg}} \exp(-s_i/\varepsilon) / \sum_{j=1}^{L} \exp(-s_j/\varepsilon)$ is a continuous function of $\varepsilon$ and $\lim_{\varepsilon \to \infty} \phi_i = \phi_{\text{avg}} < 1$, there must exist some constant $N_i$ such that when $\varepsilon \geq N_i$, $\phi_i$ is less than 1. Hence, the sparsity allocation is a valid optimal solution to equation 10 if $\varepsilon > \max(N_1, \ldots, N_L)$, completing the proof. $\square$

## B    ADDITIONAL EXPERIMENTS

### B.1    MODIFIED ALGORITHMS FOR GROUPED QUERY ATTENTION

Some modern LLMs such as Gemma (Team et al., 2024), Llama 3 (AI@Meta, 2024) utilize a shared key-query strategy to improve inference efficiency. This design adopts the grouped-query attention (GQA) mechanism (Ainslie et al., 2023) that couples multiple queries and output matrices with shared keys and values, so compressed key and value matrices also have the constraints of sharing across different heads. Its mechanism is illustrated as follows.

$$\text{(GQA)} \quad \sum_{i=1}^{H/G} \sum_{j \in G_i} \text{Softmax}(\underbrace{\sigma_r(\boldsymbol{X}\boldsymbol{W}_Q^j)\sigma_r^\top(\boldsymbol{X}\boldsymbol{W}_K^i)}_{\text{Type-II}}) \underbrace{\boldsymbol{X}\boldsymbol{W}_V^i \boldsymbol{W}_O^j}_{\text{Type-III}}, \tag{41}$$

where $G_i$ denotes the set of each group and $G = |G_i|$ is each of its size. We see that our compressed $\boldsymbol{W}_V^i$, $\boldsymbol{W}_K^j$ must be jointly optimized within each group. To address it, we modify Algorithm 2, 3 by using projection strategies. In line 3 of Algorithm 2 for Type-II module, we calculate the group score equal to the square root of sum of the scores of each head within a group, i.e., $s_i = \sqrt{\sum_{h \in G} \|\boldsymbol{C}_{h,Q}^{\frac{1}{2}}[:,i]\|^2 \|\boldsymbol{C}_{h,K}^{\frac{1}{2}}[:,i]\|^2}$, where $h$ indicates the head. By doing in this way, we ensure that the column selection matrix for compressions remains equal within the group. For grouped Type-III modification, in line 3 of Algorithm 3, we calculate the SVD of $\boldsymbol{C}\boldsymbol{W}_V = \boldsymbol{U}\boldsymbol{\Sigma}\boldsymbol{V}^\top$ and skip the calculation of $\boldsymbol{W}_O'$ and the second SVD and then outputs $\hat{\boldsymbol{W}}_V = \boldsymbol{W}_V\boldsymbol{U}_k$, $\hat{\boldsymbol{W}}_{O,j} = \boldsymbol{U}_k^\top\boldsymbol{W}_{O,j}, \forall j \in G_i$. Since $\boldsymbol{W}_V$ is shared within a group, this ensures that the compressed $\hat{\boldsymbol{W}}_V$ is also shared. In Table 14, we apply this modification to a Llama-3 8B compression.

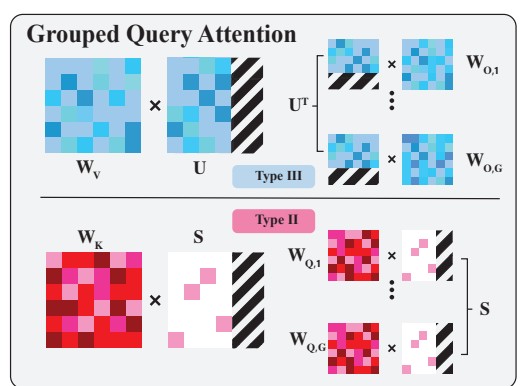

Figure 7: **Illustration of Type-II and Type-III modifications in GQA.** In Type-I, the index selection matrix $\boldsymbol{S}$ is shared among different key projection matrices in the same group. Similarly, in Type-II, the eigenmatrix $\boldsymbol{U}$ is shared among different output matrices within the same group.

### B.2    IMPLEMENTATION DETAILS

**Setup**    We utilize the HuggingFace generation library (Wolf et al., 2019) to implement our LLM models and adapt the SliceGPT (Ashkboos et al., 2024) GitHub repository for correlation matrix estimations. All compression experiments were conducted on a single NVIDIA A100 80GB GPU, except for the 70B model compressions, which utilized 8 A100 GPUs. The models use the FP16 data format. Unless otherwise specified, the calibration set consists of a random sample of 128 sequences, each of length 2048, from WikiText-2, following the common practice in the literature Ashkboos et al. (2024); van der Ouderaa et al. (2023).

**Datasets**    We consider multiple tasks in LM Evaluation Harness (Gao et al., 2021), including ARC-e, ARC-c (Clark et al., 2018), PIQA (Bisk et al., 2020), WinoGrande (Sakaguchi et al., 2021), and HellaSwag (Zellers et al., 2019), OpenBookQA (Mihaylov et al., 2018), MathQA (Amini et al., 2019), BoolQ (Clark et al., 2019), COPA (Roemmele et al., 2011), MMLU (Hendrycks et al., 2020), and LAMBADA (Paperno et al., 2016).

**Conversion of Layernorm to RMSNorm**    We adapt SliceGPT (Ashkboos et al., 2024) official code to implement our compression. As shown in the work, this conversion is an invariant transformation that preserves the model output. SliceGPT uses this transformation and the orthogonal invariance property of RMSNorm to slice the weight matrices. On the other hand, MoDeGPT does not reply on the invariance property. We use the transformation simply for easy adaptation from SliceGPT by avoiding building all from the scratch. A side benefit is that our work is compatible

with SliceGPT where a slicing and our compression can be applied independently. Although our experiments on OPT and LLAMA do not find clear improvement when incorporating the two (see Table 22), it might be beneficial for some other LLMs.

**Correlation Matrix Estimations**  Our algorithms utilize various input correlation matrices as detailed in Algorithms 1, 2, and 3. Following the approach used in SliceGPT (Ashkboos et al., 2024), we employ the Catcher function to gather empirical data from the calibration set. For matrix decomposition, we upcast correlation matrices from FP16 to FP64 and then downcast the decomposed weights back to FP16. Our process sequentially compresses weights across all layers, mirroring SliceGPT's method. Additionally, our approach is adaptable to parallel structural models like Phi-2, showcasing flexibility similar to that demonstrated by SliceGPT.

**Matrix Operations**  We utilize `torch.svd` and `torch.pinv` in PyTorch for performing Singular Value Decomposition (SVD) and computing the Moore-Penrose inverse on tensors of dtype FP64.

**MLP Module**  Algorithm 1 requires a ridge leverage score parameter $\lambda$. We find that the results are largely insensitive to this parameter; therefore, we simply use $\lambda = 1$ across all experiments.

**Key-Query Module**  MoDeGPT reduces the feature dimension in the key-query module. In our current setup, we store head-dependent index selections, which specify the rows of cosine and sine matrices in the rotary embedding, using only $O(d_h)$ INT8 numbers, together with the reduced dimension matrices. We've observed that this method may slow down generation speed at compression rates below 10%. A feasible modification is zeroing out columns associated with pruned indices; however, this increases the memory footprint because the matrices stored do not undergo dimension reduction. We think there is potential for improvements through enhanced engineering efforts that could better optimize the balance between memory savings and generation speed.

**Value-Output Module**  Lines 3-5 in Algorithm 3 provide a more computationally efficient implementation of the SVD of $C^{\frac{1}{2}}\boldsymbol{W}_{V,j}\boldsymbol{W}_{O,j}$. Since $\boldsymbol{W}_{V,j}$ and $\boldsymbol{W}_{O,j}$ are thin matrices, applying SVD directly on their product incurs $O(d_h^3)$ complexity, while applying SVD to them sequentially incurs only $O(d_h \times (d_h/H)^2)$ computations.

**Global Sparsity Allocation**  To allocate global sparsity, we first calculate the BI scores with a single forward pass on the calibration set. We then set the sparsity according to a chosen temperature $\varepsilon$, as detailed in Section 3.3. A high $\varepsilon$ leads to a very uniform allocation, while a low value introduces excessive sparsity in some layers. Empirically, we find that a simple rule of thumb is to choose a temperature $\varepsilon$ that results in maximal layer sparsity around 80%.

**Throughput Benchmark**  We use the official SliceGPT (Ashkboos et al., 2024) codebase to benchmark the throughput of all methods, with both sequence length and batch size set to 256 and utilizing KVCache.

## B.3  ADDITIONAL GENERATION AND ZERO-SHOT EXPERIMENTS

**Generation Performance**  In Table 10, we compare the perplexity of compressed OPT and LLAMA-2 7B models on WikiText-2 with other baselines that do not use gradient information. The rightmost column indicates their computational complexity per layer. We observe that MoDeGPT performs the best among all structured compression methods, and the 30-40% compressed MoDeGPT models outperform the 2:4 SparseGPT. Notably, our method shows better performance in LLAMA than in OPT models. We suspect this is due to the higher nonlinearity, such as RoPE and Gated MLP, adopted in LLAMA, and that our method favors more nonlinear structures. This table also shows that our compression is effective on small language models.

**Zero-Shot Task Performance**  In Table 11, we report the zero-shot performance of LLAMA-2 7B, calibrated with WikiText-2 and the Alpaca dataset, across various compression rates. We observe that MoDeGPT outperforms LLM Surgeon as the compression rate increases, and the benefits of using the Alpaca dataset also grow with higher compression rates. Notably, while ShortGPT performs

**Table 10:** Perplexities of none gradient-based structured compression methods on WikiText-2.

| Method | Compression | OPT | | | | LLAMA-2 | Complexity |
| | | 125M | 1.3B | 2.7B | 6.7B | 7B | |
|---|---|---|---|---|---|---|---|
| Dense | 0% | 27.65 | 14.62 | 12.47 | 10.86 | 5.12 | - |
| Sparse GPT 2:4 (Frantar & Alistarh, 2023) | 50% | 45.07 | 29.61 | 14.90 | 13.00 | 8.69 | $O(d_{\text{hidden}}^3)$ |
| Magnitude | 10% | 767.2 | 894.4 | 1229 | 3464 | 861.76 | $O(d_{\text{hidden}}^2)$ |
| | 20% | 4685 | 1278 | 2788 | 16747 | 821.34 | |
| | 30% | 17970 | 3098 | 9255 | 17312 | 9623 | |
| SVD | 10% | 36.29 | 68.36 | 20.82 | 357.61 | n/a | $O(d_{\text{hidden}}^3)$ |
| | 20% | 55.48 | 1023.49 | 50.01 | 2387.39 | n/a | |
| | 30% | 173.77 | 8851.45 | 707.17 | 9448.12 | 52719 | |
| OBD (LeCun et al., 1989) | 10% | 33.3 | 20.76 | 17.69 | 27.2 | 14259 | $O(Td_{\text{hidden}}^3)$ |
| | 20% | 94.14 | 1392 | 3236 | 7570 | 15630 | |
| | 30% | 545.6 | 2147 | 7233 | 7628 | 21386 | |
| SliceGPT (Ashkboos et al., 2024) | 10% | 34.48 | 16.58 | 13.86 | 11.6 | 6.46 | $O(d_{\text{hidden}}^3)$ |
| | 20% | 42.87 | 19.15 | 15.86 | 12.62 | 7.68 | |
| | 30% | 59.87 | 23.87 | 19.91 | 14.19 | 10.47 | |
| | 40% | 102.41 | 36.2 | 30.77 | 17.99 | 15.19 | |
| | 50% | 185.52 | 66.12 | 56.99 | 26.72 | 24.82 | |
| MoDeGPT (ours) | 10% | 28.06 | 15.03 | 12.78 | 11.17 | 5.48 | $O(d_{\text{hidden}}^3)$ |
| | 20% | 29.62 | 15.98 | 13.56 | 11.79 | 6.16 | |
| | 30% | 33.27 | 17.91 | 14.71 | 12.67 | 7.51 | |
| | 40% | 38.37 | 21.92 | 17.43 | 14.79 | 8.41 | |
| | 50% | 51.81 | 32.67 | 24.75 | 20.39 | 11.88 | |

**Table 11:** Downstream zero-shot task performance of LLAMA-2 7B calibrated with 128 samples from Wiki-Text2.

| Compression | Method | ARC-e | ARC-c | PIQA | WinoGrande | HellaSwag | Average |
|---|---|---|---|---|---|---|---|
| 0% | Dense | 74.58% | 46.25% | 79.11% | 69.06% | 75.99% | 69.00% |
| 20% | ShortGPT (Men et al., 2024) | 58.33% | 38.05% | 72.58% | 65.51% | 65.27% | 59.95% |
| | SliceGPT (Ashkboos et al., 2024) | 51.47% | 31.06% | 64.25% | 62.74% | 49.78% | 51.86% |
| | LLM surgeon (van der Ouderaa et al., 2023) | 71.36% | 41.89% | 77.09% | 66.30% | 71.30% | 65.59% |
| | MoDeGPT (ours) | 69.07% | 42.06% | 74.05% | 68.03% | 69.05% | 64.46% |
| | MoDeGPT-Alpaca (ours) | 71.71% | 41.89% | 76.22% | 68.19% | 69.59% | 65.52% |
| 30% | ShortGPT (Men et al., 2024) | 48.65% | 32.85% | 64.31% | 64.33% | 56.13% | 53.25% |
| | SliceGPT (Ashkboos et al., 2024) | 44.44% | 29.27% | 57.56% | 58.48% | 41.08% | 46.17% |
| | LLM surgeon (van der Ouderaa et al., 2023) | 63.09% | 36.69% | 73.56% | 61.09% | 60.72% | 59.03% |
| | MoDeGPT (ours) | 63.26% | 38.73% | 70.40% | 67.32% | 63.26% | 60.78% |
| | MoDeGPT-Alpaca (ours) | 65.49% | 39.16% | 73.34% | 66.22% | 65.90% | 62.02% |
| 40% | ShortGPT (Men et al., 2024) | 41.16% | 29.94% | 60.12% | 60.46% | 43.67% | 47.07% |
| | SliceGPT (Ashkboos et al., 2024) | 36.49% | 24.57% | 54.90% | 53.43% | 34.80% | 40.84% |
| | LLM surgeon (van der Ouderaa et al., 2023) | 52.31% | 30.29% | 69.26% | 54.38% | 48.04% | 50.86% |
| | MoDeGPT (ours) | 49.45% | 30.03% | 64.96% | 61.96% | 53.01% | 51.88% |
| | MoDeGPT-Alpaca (ours) | 59.76% | 34.73% | 70.35% | 64.40% | 58.63% | 57.58% |

**Table 12:** Downstream zero-shot task performance of LLAMA-2 13B calibrated with 128 samples from Wiki-Text2.

| Method | Compression | ARC-e | ARC-c | PIQA | WinoGrande | HellaSwag | Average |
|---|---|---|---|---|---|---|---|
| Dense | 0% | 77.48% | 49.23% | 80.47% | 72.22% | 79.39% | 71.76% |
| SliceGPT (Ashkboos et al., 2024) | 20% | 55.81% | 35.84% | 65.83% | 67.17% | 53.58% | 55.65% |
| | 30% | 45.96% | 30.80% | 59.63% | 61.80% | 44.09% | 48.46% |
| | 40% | 38.59% | 27.05% | 55.98% | 56.51% | 37.15% | 43.06% |
| MoDeGPT (ours) | 20% | 74.07% | 46.16% | 74.53% | 70.32% | 68.96% | 66.81% |
| | 30% | 71.93% | 43.60% | 73.94% | 71.90% | 68.21% | 65.92% |
| | 40% | 62.88% | 38.40% | 69.10% | 67.72% | 58.27% | 59.27% |

**Table 13:** Downstream zero-shot task performance of LLAMA-2 13B calibrated with 128 samples from Alpaca.

| Method | Compression | ARC-e | ARC-c | PIQA | WinoGrande | HellaSwag | Average |
|---|---|---|---|---|---|---|---|
| Dense | 0% | 77.48% | 49.23% | 80.47% | 72.22% | 79.39% | 71.76% |
| SliceGPT (Ashkboos et al., 2024) | 20% | 69.36% | 40.70% | 74.97% | 65.67% | 61.01% | 62.34% |
| | 30% | 60.27% | 36.18% | 69.42% | 64.09% | 49.74% | 55.94% |
| | 40% | 48.99% | 32.51% | 63.17% | 56.75% | 39.85% | 48.25% |
| MoDeGPT (ours) | 20% | 74.24% | 45.90% | 78.24% | 72.53% | 75.78% | 69.34% |
| | 30% | 70.24% | 41.47% | 77.15% | 71.27% | 71.84% | 66.39% |
| | 40% | 63.72% | 38.82% | 71.87% | 66.30% | 62.10% | 60.56% |

poorly in generation tasks, it significantly outperforms SliceGPT in zero-shot tasks. Both LLM Surgeon and MoDeGPT maintain high performance in generation and zero-shot tasks, but our method requires only 3% of the computational resources compared to LLM Surgeon.

We also test the performance on LLAMA-2 13B using the WikiText-2 calibration set, as shown in Table 12. Similar to the 7B model, our method excels at higher compression rates (above 20%). However, at a 40% compression rate, we notice a performance drop in the HellaSwag task compared to the 30% compression, likely due to inherent biases in our method. Nevertheless, with calibration from the Alpaca dataset, as shown in Table 13, our method achieves high performance at 20% and 30% compression. Addressing these inherent biases and enhancing performance on the HellaSwag task is a promising area for future research.

**Table 14:** Downstream zero-shot task performance of LLAMA-3 8B calibrated with 128 samples from Alpaca.

| Method | Compression | Perplexity ↓ | ARC-e | ARC-c | PIQA | WinoGrande | HellaSwag | Average |
|---|---|---|---|---|---|---|---|---|
| Dense | 0% | 2.98 | 77.69% | 53.58% | 80.63% | 72.69% | 79.16% | 72.75% |
| ShortGPT (Men et al., 2024) | 25% | 282.56 | 38.13% | 31.40% | 60.94% | 54.22% | 31.52% | 43.24% |
| | 30% | 659.33 | 36.83% | 30.72% | 58.98% | 54.62% | 29.08% | 42.04% |
| SliceGPT (Ashkboos et al., 2024) | 25% | 3.87 | 58.88% | 33.36% | 68.55% | 58.01% | 49.86% | 53.73% |
| | 30% | 4.52 | 52.02% | 29.18% | 64.85% | 54.62% | 41.38% | 48.41% |
| MoDeGPT (ours) | 25% | 3.52 | 67.05% | 41.13% | 75.52% | 69.61% | 66.49% | 63.96% |
| | 30% | 3.80 | 62.75% | 38.65% | 73.61% | 67.25% | 62.10% | 60.87% |

In Table 14, we test our method on LLAMA-3 8B using our modified algorithm tailored for grouped query attention. As this is a relatively new model, we could only compare results with SliceGPT and ShortGPT, which already support this model. We observe that compression has a more significant impact on performance degradation compared to LLAMA-2 for all tested methods. We believe this is due to denser information encoding in each parameter, making the model more sensitive to weight changes. However, MoDeGPT maintains approximately 90% performance with 25% compression. Another interesting observation is that the performance order of ShortGPT and SliceGPT is reversed for LLAMA-3 compared to LLAMA-2, with ShortGPT's layer pruning strategy performing much worse. This suggests fundamental differences between the two models and indicates that LLAMA-3 may require more sophisticated compression techniques.

In Table 16, we evaluate MoDeGPT on a broader range of tasks to assess its generalizability. We compare MoDeGPT with two baseline methods: SliceGPT (a decomposition approach) and Short-GPT (a layer-pruning method), all using 30% compression of LLAMA-2 7B, with calibration performed on the Alpaca dataset. The top rows of the table show the raw accuracies, while the bottom rows display the relative degradation compared to the original dense model.

MoDeGPT demonstrates the least degradation across all tasks, with an average drop of only 7.19%, while the other methods experience drops exceeding 17%. Notably, the degradation is generally consistent across tasks, except for Lambada and MMLU, which show more significant drops. These tasks also exhibit the largest degradations in the baseline methods, suggesting they are more sensitive to compression.

Lambada, in particular, exhibits an extreme degradation in both SliceGPT and ShortGPT (over 40% for both), making it the most challenging task to maintain accuracy after compression. In contrast, MoDeGPT shows a relatively small degradation of just 16.8%, almost 25% lower than the other

**Table 15:** Generation samples of LLAMA-2 7B with 30% compression.

| | |
|---|---|
| | **Prompt:** The universe is the entirety of space, time, matter, and energy that exists. |
| **ShortGPT** (Men et al., 2024) | sierpę 2018. The universe is everything that ever did or ever will exist. It includes all the past, all the present, and all the future. What are the 5 most important things you learned about life? How do you describe a person in words? Can you be happy with nothing? Is there a word for everything? What is the meaning of universe? Who is the best person to talk to when you need advice? What is the meaning of the word universe? What is the meaning of universe according to Godungsseite. According to Wikipedia, the universe refers to everything that ever existed, continues existence, or could possibly exist. Thus it includes all the past, all the present, and all the future. |
| **SliceGPT** (Ashkboos et al., 2024) | The universe contains many galaxies, stars, planets, other objects such as asteroids, planetary systems. It has been estimated to contain 100 billion trillion stars. The Sun is a star in the Milky Way galaxy , which is one of about 400 billion galaxies in the Milky Way's vicinity . The Milky Way itself is part of the Milk Way Group , which includes several smaller groups of galaxies . |
| **MoDeGPT** (ours) | The universe is a vast, dynamic, and complex structure that contains many different types of objects, including stars, galaxies, planets, asteroids, and intergalactic gas clouds. It is also home to life-forms such as humans, animals, plants, microorganisms, and microscopic organisms. The universe has been studied by astronomers for centuries, and its size and complexity have been estimated using various methods. Astronomical observations have revealed that the universe is much larger than previously thought, with an estimated diameter of approximately 10 billion light years. This means that the universe is roughly 33 times larger than our solar system. The universe is composed of two main components: the cosmos and the cosmic void. The cosmos consists of galaxies, which are large collections of stars and other objects bound together by gravitational forces. The cosmic void, on the other hand, is a region of space between galaxies that is largely empty of matter. |

**Table 16:** Zero-shot task performance degradation of LLAMA-2 7B, calibrated with 128 samples from the Alpaca dataset, evaluated across a broader set of tasks.

| Method | BoolQ | PIQA | HellaS. | WinoG. | ARC-e | ARC-c | OBQA | COPA | Lamb. | MMLU-ml | Average |
|---|---|---|---|---|---|---|---|---|---|---|---|
| Dense | 77.68% | 79.05% | 76.00% | 68.98% | 74.58% | 46.33% | 44.22% | 87.00% | 73.86% | 39.29% | 66.70% |
| SliceGPT (Ashkboos et al., 2024) | 61.99% | 68.55% | 48.69% | 59.75% | 59.69% | 34.47% | 31.40% | 75.00% | 21.02% | 23.21% | 48.08% |
| ShortGPT (Men et al., 2024) | 62.17% | 64.48% | 56.15% | 64.33% | 48.70% | 32.59% | 32.80% | 79.00% | 29.03% | 24.11% | 49.34% |
| MoDeGPT (ours) | **69.76%** | **73.34%** | **65.90%** | **66.22%** | **65.49%** | **39.16%** | **39.00%** | **87.00%** | **57.07%** | **32.14%** | **59.51%** |
| Δ SliceGPT | -15.69% | -10.50% | -27.31% | -9.23% | -17.89% | -11.86% | -12.80% | -12.00% | -52.84% | -16.08% | -18.62% |
| Δ ShortGPT | -15.51% | -14.57% | -19.85% | -4.65% | -25.88% | -13.74% | -11.40% | -8.00% | -44.83% | -15.18% | -17.36% |
| Δ MoDeGPT (ours) | -7.92% | -5.71% | -10.10% | -2.76% | -9.09% | -7.17% | -5.20% | 0% | -16.79% | -7.15% | -7.19% |

methods. This hints that MoDeGPT is better at preserving important information, which is crucial for excelling on more difficult tasks like Lambada.

Finally, we compare the generation quality using samples from the three methods' generations for 30% compressed LLAMA-2 7B, as shown in Table 15. ShortGPT produces the lowest quality generation, while both SliceGPT and MoDeGPT generate high-quality responses, with MoDeGPT providing more detailed responses than SliceGPT.

## B.4 ADDITIONAL BASELINE COMPARISONS: FEATURE-MIMIC AND SVD APPROACHES

In Table 17 and 18, we compare our method against feature-mimic and SVD-based approaches, respectively. In the former case, we observe that alternative methods generally underperform compared to state-of-the-art gradient-based techniques like LLM Surgeon, while our approach achieves comparable or even superior results. In the latter comparison, our advantage is even more pronounced, which we attribute to our more refined decomposition algorithms, tailored specifically to different components of the transformer architecture based on their levels of nonlinearity, rather than relying solely on SVD-based decompositions.

## B.5 ADDITIONAL BASELINE COMPARISONS: UNSTRUCTURED/SEM-STRUCTURED COMPRESSION

In Table 19, We compare MoDeGPT to the state-of-the-art non-structured methods, Wanda, SparseGPT, and ZeroPruner. These methods generally outperform MoDeGPT at 50% compres-

**Table 17:** Comparisons of feature-mimic based methods for 30% compression of LLAMA-2 7B and 13B models.

| Model | Method | ARC-e | ARC-c | PIQA | WinoG. | HellaS. | BoolQ | OBQA | Average. |
|---|---|---|---|---|---|---|---|---|---|
| **LLAMA-2 7B** | Dense | 74.58 | 46.25 | 79.11 | 69.06 | 75.99 | 77.74 | 44.20 | 66.70 |
| | LLM Pruner (Ma et al., 2023) | 61.41 | 33.96 | 71.93 | 58.72 | 59.49 | 61.41 | 36.60 | 53.52 |
| | FLAP (An et al., 2024) | 60.65 | 34.47 | 72.74 | 64.01 | 63.80 | 66.88 | 36.40 | 56.99 |
| | Bolaco (5 × 4) (Ji et al., 2024) | 65.87 | 34.30 | 71.27 | 64.48 | 57.85 | 73.85 | 37.80 | 57.92 |
| | MoDeGPT (ours) | 65.49 | 39.16 | 73.34 | 66.22 | 65.90 | 69.76 | 39.00 | 59.83 |
| **LLAMA-2 13B** | Dense | 77.48 | 49.23 | 80.47 | 72.22 | 79.39 | 80.52 | 45.20 | 69.22 |
| | LLM Pruner (Ma et al., 2023) | 65.45 | 40.36 | 75.90 | 60.22 | 67.90 | 62.43 | 44.60 | 59.55 |
| | FLAP (An et al., 2024) | 67.38 | 38.23 | 74.81 | 67.48 | 70.29 | 65.54 | 40.00 | 60.53 |
| | Bolaco (5 × 4) (Ji et al., 2024) | 71.76 | 40.10 | 74.16 | 69.06 | 66.66 | 75.63 | 41.60 | 62.71 |
| | MoDeGPT (ours) | 70.24 | 41.47 | 77.15 | 71.27 | 71.84 | 73.7 | 41.00 | 63.81 |

**Table 18:** Comparisons with SVD-based methods in LLAMA-1 7B.

| Compress. Rate | Method | WikiText-2 ↓ | PTB ↓ | ARC-e | ARC-c | PIQA | WinoG. | HellaS. | MathQA | OBQA | Avg. |
|---|---|---|---|---|---|---|---|---|---|---|---|
| 0% | Dense | 5.68 | 8.35 | 73 | 42 | 79 | 70 | 50 | 27 | 34 | 54 |
| 20% | FWSVD (Hsu et al., 2022) | 1727 | 2152 | 31 | 23 | 56 | 50 | 26 | 21 | 15 | 32 |
| | ASVD (Yuan et al., 2023) | 11.14 | 16.55 | 53 | 27 | 68 | 64 | 41 | 24 | 25 | 43 |
| | SVD-LLM (Wang et al., 2024) | 7.94 | 16.22 | 58 | 29 | 69 | 58 | 43 | 24 | 22 | 44 |
| | MoDeGPT (ours) | 6.53 | 39.17 | 70 | 36 | 74 | 69 | 50 | 26 | 31 | 51 |
| 40% | FWSVD (Hsu et al., 2022) | 18156 | 20990 | 26 | 22 | 53 | 51 | 26 | 21 | 16 | 30 |
| | ASVD (Yuan et al., 2023) | 1407 | 3292 | 28 | 22 | 55 | 48 | 26 | 19 | 12 | 30 |
| | SVD-LLM (Wang et al., 2024) | 13.11 | 63.75 | 42 | 25 | 60 | 58 | 33 | 21 | 19 | 37 |
| | MoDeGPT (ours) | 9.39 | 60.55 | 58 | 30 | 65 | 64 | 40 | 23 | 22 | 43 |

sion rate. However, MoDeGPT with 40% compression achieves a significantly better perplexity (8.41 versus 10.17).

The observation suggests that with a small concession on compression rate, our structured compression can be on par with the semi-structured method that requires special GPU support for efficient inference.

### B.6 RECOVERY FINE-TUNING

**Table 19:** Comparisons with semi-structured pruning.

| Method | Structure | 40% | 50% |
|---|---|---|---|
| SparseGPT (2:4) | Semi-structured | - | **10.17** |
| Wanda (2:4) | Semi-structured | - | 11.02 |
| ZeroPruner (2:4) | Semi-structured | - | 10.52 |
| MoDeGPT (ours) | Structured | 8.41 | 11.88 |

**Table 20:** Compression and recovery fine-tuning for LLAMA-2 7B using Alpaca dataset

| Method | Compress. | ARC-e | ARC-c | PIQA | WinoGrande | HellaSwag | Average |
|---|---|---|---|---|---|---|---|
| Dense | 0% | 74.58% | 46.25% | 79.11% | 69.06% | 75.99% | 69.00% |
| MoDeGPT RCT-MLP | 20% | 69.78 % (↓ 1.93%) | 44.20% (↑ 2.31%) | 76.99% (↑ 0.77%) | 66.61% (↓ 1.58%) | 69.23% (↓ 0.36%) | 65.36% (↓ 0.16%) |
| | 30% | 64.94% (↓ 0.55%) | 42.15% (↑ 2.99%) | 73.83% (↑ 0.49%) | 66.54% (↑ 0.32%) | 67.08% (↓ 1.18%) | 62.91% (↓ 0.89%) |
| | 40% | 59.26% (↓ 0.50%) | 37.12% (↑ 2.39%) | 72.09% (↑ 1.74%) | 64.33% (↓ 0.07%) | 60.82% (↑ 2.19%) | 58.72% (↑ 1.14%) |
| MoDeGPT RCT-ALL | 20% | 70.45% (↓ 1.26%) | 42.92% (↑ 1.03%) | 77.20% (↑ 0.98%) | 66.30% (↓ 1.89%) | 68.07% (↓ 1.52%) | 64.99% (↓ 0.53%) |
| | 30% | 63.38% (↓ 2.11%) | 41.47% (↑ 2.31%) | 74.81% (↑ 1.47%) | 66.06% (↓ 0.16%) | 65.64% (↓ 0.58%) | 62.27% (↑ 0.25%) |
| | 40% | 58.42% (↓ 1.34%) | 38.23% (↑ 3.50%) | 72.03% (↑ 1.68%) | 63.61% (↓ 0.79%) | 59.55% (↑ 0.92%) | 58.34% (↑ 0.76%) |

While MoDeGPT does not require recovery fine-tuning, in this section, we explore how RFT can further enhance performance. In Table 20, we present recovery fine-tuning results for our method on LLAMA-2 7B, following the same tuning setting as SliceGPT (Ashkboos et al., 2024). We use a calibration set of 128 random samples, each 2048 in length, from the Alpaca dataset, and a recovery fine-tuning set of 8000 samples, each 1024 in length, employing LoRA (Hu et al., 2021). We use SliceGPT's hyperparameters for LoRA, except for the learning rate, which is set to $5 \times 10^{-5}$. The other primary hyperparameters used are $lora\_alpha = 10$, $lora\_r = 32$, $lora\_dropout = 0.05$, and $batch\_size = 3$. We evaluate two scenarios: 1) fine-tuning all linear matrices, and 2) tuning only the MLP.

The green and red indicators in the table denote performance increases or decreases relative to the compressed model before fine-tuning. Notably, tuning exclusively within the MLP consistently yields better performance than tuning all parameters. Since we followed the same tuning setting as

SliceGPT (Ashkboos et al., 2024) for a fair comparison, it is likely that better configurations exist for our method, potentially enhancing performance further. Another key observation is that despite fine-tuning using 40 times more data than calibration and employing backpropagation, MoDeGPT without RFT achieves very similar performance. The percentage difference is minimal, suggesting that using local reconstruction error as the objective is an effective and efficient method with our compression technique.

The table demonstrates that fine-tuning can slightly improve performance for higher compression rates, with the most significant increase observed in the ARC-c task. Evaluating the full benefits of fine-tuning remains a subject for future research.

## B.7 EXPERIMENTS WITH EQUAL COMPUTATIONAL BUDGETS

**Table 21:** Compression comparisons with approximately equal computational budgets.

| Method | Time (Compress / Fine-tune) | PPL | ARC-e | ARC-c | PIQA | WinoG. | HellaS. | Average. |
|---|---|---|---|---|---|---|---|---|
| SliceGPT | 26m / 4h05m | **2.59** (3.52) | 56.82 (56.69) | 38.48 (34.47) | 71.82 (68.55) | 59.83 (59.75) | 59.30 (48.69) | 57.26 (53.63) |
| SLEB | 9m / 4h50m | 2.67 (4.36) | 52.36 (52.36) | 34.04 (31.91) | 71.00 (69.58) | 59.98 (58.17) | 60.16 (58.28) | 55.51 (54.06) |
| MoDeGPT | 4h09m / 31m | 2.70 (**3.08**) | **67.42 (65.49)** | **40.96 (39.16)** | **74.10 (73.34)** | **65.98 (65.49)** | **66.57 (65.90)** | **63.01 (62.02)** |

We study the combined effect of compression and recovery fine-tuning for different approaches with equal computational cost, as shown in Table 21 . In this experiment, we compress LLAMA-2 7B with a 30% compression rate on a single A100 GPU. The model is first compressed using 128 samples from the Alpaca dataset for calibration, followed by fine-tuning with LoRA on 5k Alpaca samples. For fair comparisons, we fix the hyperparameters as $lora\_alpha = 10$, $lora\_r = 32$, and $lora\_dropout = 0.05$. We compare MoDeGPT against SliceGPT and SLEB, which serve as baselines for decomposition-based and layer-pruning-based approaches, respectively.

Since the methods vary in compression time, we adjust the fine-tuning epochs to equalize the total time spent across methods. The table reports the time spent in each phase for different methods. Notably, MoDeGPT has the longest compression time and is therefore fine-tuned for only one epoch. The table presents zero-shot accuracies both before and after fine-tuning (after/before).

MoDeGPT achieves the highest zero-shot performance across all tasks, excluding perplexity, both before and after fine-tuning, with its performance advantage primarily arising from the compression phase. The superior perplexity but lower zero-shot performance of SliceGPT compared to MoDeGPT underscores the pivotal role of the compression stage, suggesting that an excessive computational focus on fine-tuning may lead to overfitting.

Lastly, SLEB, despite having the longest fine-tuning time, exhibits smaller improvements than SliceGPT in zero-shot performances, further emphasizing the pivotal role of the compression phase in determining the final model's performance. Moreover, MoDeGPT outperforms the baselines even without fine-tuning, demonstrating its effectiveness during the compression stage.

## B.8 COMBINATION OF MODEGPT AND SLICEGPT

MoDeGPT is orthogonal to SliceGPT as it reduces dimensions from different sides of a weight matrix. Figures 1 (c) and (d) provide an illustrative comparison. Combining SliceGPT with MoDeGPT seems to be a natural extension. To demonstrate their compatibility, we experimented with various configurations as shown in Table 22. The numbers $x$-$y$-$z$ in the leftmost column indicate $x$% slicing rate of SliceGPT, and $y$% and $z$% compression rates of MoDeGPT in MLP and MHA modules, respectively. The two rightmost columns test the use of sparsity allocation in the MLP and/or MHA modules.

Notably, our tests show that applying sparsity allocation with SliceGPT barely improves performance, consistent with the findings in the SliceGPT paper (Ashkboos et al., 2024). Therefore, we do not use sparsity allocation for slicing. Compared to the results in Table 3, the combination of SliceGPT and MoDeGPT does not improve perplexity over pure MoDeGPT. We attribute this to two points: 1. the significant overhead induced by slicing: to achieve a target compression rate, the

**Table 22:** Perplexity performance of SliceGPT + MoDeGPT on LLAMA-2 7B

| Slice-MLP-MHA (%-%-%) | Compression Rate | WikiText2 Perplexity ↓ | MLP Sparsity Allocation | MHA Sparsity Allocation |
|---|---|---|---|---|
| Dense | 0% | 5.12 | - | - |
| 20-20-0 | 19.65% | 7.38 | ✓ | ✗ |
| 20-20-0 | 19.65% | 7.33 | ✗ | ✗ |
| 25-25-0 | 27.38% | 8.42 | ✓ | ✗ |
| 30-30-0 | 34.93% | 9.99 | ✓ | ✗ |
| 20-25-0 | 22.25% | 7.70 | ✗ | ✗ |
| 15-30-0 | 11.77% | 7.27 | ✗ | ✗ |
| 10-30-0 | 9.03% | 6.83 | ✗ | ✗ |
| 10-25-25 | 28.00% | 7.31 | ✗ | ✗ |
| 10-30-25 | 30.91% | 7.78 | ✗ | ✗ |
| 20-20-20 | 29.18% | 8.00 | ✗ | ✗ |

model must slice at a higher rate. 2. the slicing and compression ratio might not be optimal, and it might changes from layer to layer.

Although we did not make exhaustive search, we believe there is an efficient sparsity allocation for slicing, and better tuning of the slicing and compression ratios could enhance the performance of the combined method. We leave this as a topic for future research.

## B.9 COMPRESSION TIME AND MEMORY CONSUMPTION

**Table 23:** Compression computations for calibration set of size 128 in WikiText2.

| Model | MoDeGPT | | SliceGPT Ashkboos et al. (2024) | | LLM surgeon van der Ouderaa et al. (2023) | |
|---|---|---|---|---|---|---|
| | Time | GPUs | Time | GPUs | Time | GPUs |
| LLAMA-2 7B | 4h09m | 1xA100 80GB | 0h26m | 1xA100 80GB | 17h08m | 4xH100 80GB |
| LLAMA-2 13B | 8h26m | 1xA100 80GB | 0h45m | 1xA100 80GB | 1d9h26m | 8xH100 80GB |

**Table 24:** Memory consumption and compute time of 30% compression for blocks in transformer layers tested on a single A100 80GB GPU.

| Block | LLAMA-7B (13.81 GiB) | | LLAMA-13B (25.92 GiB) | |
|---|---|---|---|---|
| | Peak Memory (GiB) | GPU hours | Peak Memory (GiB) | GPU hours |
| MHA | 15.54 (+11.5%) | 2h52m | 28.60 (+9.4%) | 5h04m |
| MLP | 23.33 (+68.9%) | 1h13m | 41.40 (+54.1%) | 3h22m |

In Table 23, we compare the compression times of MoDeGPT, SliceGPT, and LLM Surgeon. Since MoDeGPT and SliceGPT do not leverage gradients, they can compress a model of size 13B using a single GPU. From previous tables, we observe that while our compute time is longer than

**Table 25:** Downstream zero-shot task performance of 30% MoDeGPT on LLAMA-2 7B for varying global rank temperature.

| Method | $\varepsilon$ | Perplexity ↓ | ARC-e | ARC-c | PIQA | WinoGrande | HellaSwag | Average |
|---|---|---|---|---|---|---|---|---|
| Dense | - | 5.12 | 74.58% | 46.25% | 79.11% | 69.06% | 75.99% | 69.00% |
| MoDeGPT (ours) | 0.075 | 7.44 | 59.72% | 37.29% | 68.50% | 65.90% | 61.55% | 58.59% |
| | 0.1 | 7.46 | 63.43% | 39.42% | 70.78% | 65.59% | 63.24% | 60.49% |
| | 0.5 | 7.03 | 56.14% | 32.34% | 67.68% | 64.88% | 58.01% | 55.81% |
| | 1 | 7.25 | 53.20% | 31.06% | 66.16% | 64.17% | 56.66% | 54.25% |
| | 2 | 7.35 | 53.62% | 31.06% | 65.83% | 63.14% | 55.98% | 53.93% |
| | $\infty$ | 9.06 | 52.36% | 30.80% | 65.18% | 63.69% | 55.31% | 53.47% |

SliceGPT, MoDeGPT achieves significantly better performance. Conversely, our computation time is considerably shorter than LLM Surgeon, yet we achieve comparable performance. Even when equating 1 H100 to 1 A100, our method can save up to 97% of computations. In Table 24, we report the peak GPU memory usage when compressing LLAMA-2 7B and 13B models on a single A100 GPU. The primary source of additional memory overhead, beyond the model itself, is the storage of intermediate activations required for correlation estimation in the MLP. The table shows that this overhead ranges from approximately 50% to 70%. However, for the 13B model, the peak memory usage remains under 50% of the total GPU memory capacity.

### B.10 GLOBAL SPARSITY ALLOCATION

In Table 25, we report the perplexity and zero-shot performance as we vary the temperature parameter in the global sparsity allocation. Initially, the uniform strategy, corresponding to an infinite temperature, performs significantly worse than our sparsity allocation strategy.

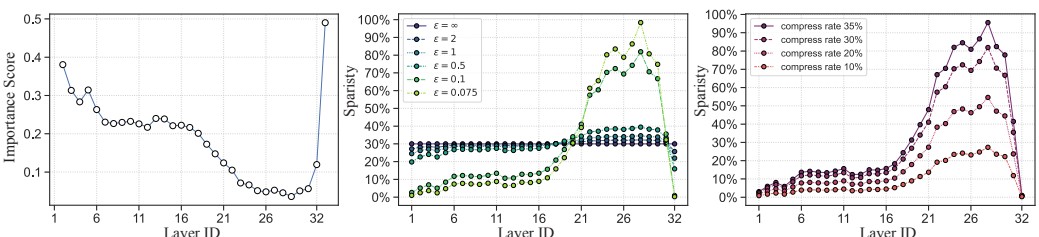

**Figure 8:** Dynamic sparsity allocation across layers for LLAMA-2 7B.

For $\varepsilon = 0.075$, this results in extreme sparsity, as shown in Figure 8, and performance begins to drop. For $\varepsilon \geq 1$, the allocation becomes too similar to the uniform strategy. In practice, we find that the $\varepsilon$ value that yields a minimum layer allocation around 20% performs exceptionally well. In Figure 8, left and middle panels, we also observe how the allocation shape for different $\varepsilon$ values corresponds to the importance score in the left figure. For LLMs, both OPT and LLAMA show that the first and last few layers have significantly higher importance and, therefore, should allocate less sparsity, as depicted in the figure. On the right of Figure 8, we also show the allocation for different compression levels. The shape remains similar across different levels using the same $\varepsilon$. For higher compression rates, the maximum layer sparsity also increases, suggesting that we should increase $\varepsilon$ to avoid extreme sparsity. We report the ranks of the QKV projection matrices across various layers of compressed LLAMA-2 7B and 70B models, as determined by the global sparsity allocation used in this study (equation 11), with their distributions visualized in Figure 10.

The ranks were computed using equation 11 with 128 samples from WikiText-2 and $\varepsilon$ values of 0.1 and 0.02 for the 7B and 70B models, respectively. These $\varepsilon$ values were selected to ensure the maximum layer sparsity remains around 70–80%, as a 90% sparsity level is often too extreme, based on our experience from experiments.

Interestingly, we found that the rank distributions exhibit similar shapes across the models, suggesting a deep connection between the allocation strategy and the LLAMA-2 family architectures.

**Table 26:** Layer ranks for various models.

| Model | Layer Rank |
|-------|------------|
| LLAMA-2 7B | 3989, 3886, 3813, 3889, 3750, 3616, 3598, 3612 |
| | 3625, 3593, 3546, 3660, 3654, 3568, 3575, 3544 |
| | 3453, 3241, 2997, 2703, 2413, 1741, 1620, 1217 |
| | 1129, 1254, 1054, 741, 1203, 1363, 2640, 4060 |
| LLAMA-2 70B | 8192, 8183, 8186, 8169, 8143, 8103, 8130, 8088 |
| | 8134, 7983, 7908, 7873, 7957, 8018, 7932, 7968 |
| | 7772, 8000, 7858, 7784, 7486, 7419, 7079, 7016 |
| | 7090, 7596, 7214, 6784, 6620, 6556, 6204, 6384 |
| | 6366, 6762, 6719, 6411, 6472, 6356, 6651, 6918 |
| | 7138, 6839, 6872, 6112, 6620, 5467, 5042, 5328 |
| | 4402, 3940, 3563, 3745, 3632, 3076, 2814, 3051 |
| | 2814, 2622, 3025, 2395, 2189, 2128, 2158, 2128 |
| | 2248, 2037, 2760, 2947, 2453, 3051, 3152, 3609 |
| | 3446, 3540, 4148, 4694, 5548, 5994, 7355, 8187 |

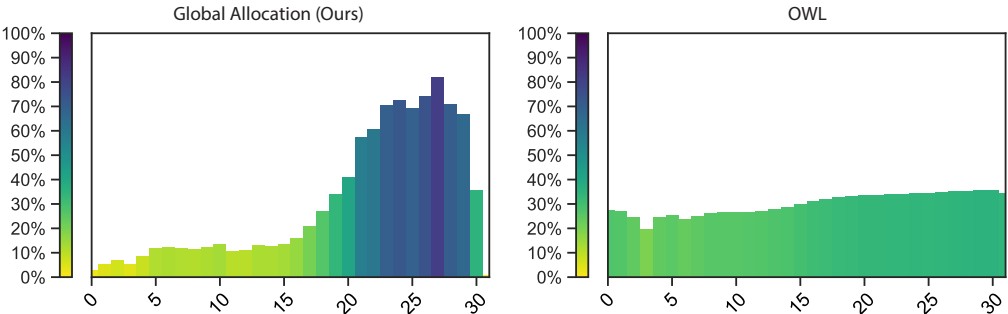

**Figure 9:** Layer sparsity distribution comparisons.

**Table 27:** Global sparsity allocation comparisons.

| Method | Sparsity Mean | Sparsity Std | Perplexity ↓ | PIQA | HellaS. | WinoG. | ARC-E | ARC-C | Average |
|--------|---------------|--------------|--------------|------|---------|--------|-------|-------|---------|
| Uniform Allocation | 30% | 0% | 9.06 | 65.18 | 55.31 | 63.69 | 52.36 | 30.80 | 53.47 |
| Global Sparsity Allocation (Ours) | 30% | 26.72% | 7.51 | **71.40** | **63.26** | **67.32** | **63.26** | **38.73** | **60.79** |
| OWL Yin et al. (2023) | 30% | 4.46% | **6.9** | 68.17 | 59.12 | 65.67 | 56.9 | 33.36 | 56.64 |

We also compare our global allocation strategy in equation 11 with a state-of-the-art alternative, OWL (Yin et al., 2023), as shown in Table 27. In this experiment, we compress LLAMA-2 7B using MoDeGPT with different global sparsity strategies. Despite our method having a higher perplexity, it consistently achieves better zero-shot performance across all reported tasks. Figure 9 visualizes the layer sparsity distributions of the two approaches. Unlike OWL, our distribution exhibits much greater heterogeneity across layers, showing less sparsity in the first and last layers. This figure suggests that heterogeneity may play a crucial role in structured compression.

### B.11 REFINED SPARSITY ALLOCATION: DIFFERENTIATING MLP AND MHA BLOCKS WITH FINER-GRAINED SCORES

In this subsection, we refined our global sparsity allocation strategy by introducing different sparsity levels for the MLP and MHA blocks within each transformer layer. A similar approach to layer pruning, which employs distinct sparsity levels for MLP and MHA, has been explored by Finercut (Zhang et al., 2024). This strategic refinement has significantly improved both compression accuracy and inference throughput, particularly enhancing inference speed. Notably, in our 30% compression experiments on LLAMA-2 7B, as illustrated in Table 28, we achieved the highest throughput among

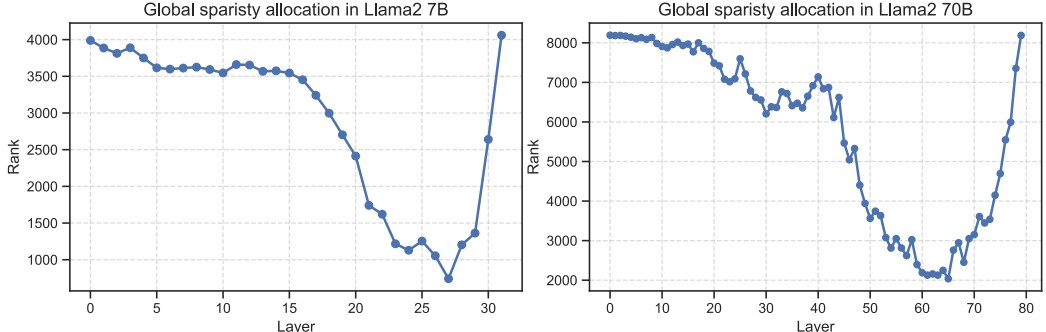

**Figure 10:** Layer Ranks of LLAMA-2 7B and 70B.

all baselines, surpassing even dedicated layer pruning strategies, while maintaining exceptional accuracy.

Instead of computing a single score per layer, we now calculate two distinct scores—one for MLP and another for MHA—applying the correlation methodology outlined in Section 3.3. This dual-score system enables a more nuanced sparsity allocation that aligns better with the unique computational and structural characteristics of each block type, thereby optimizing performance without incurring significant computational overhead. The updated global sparsity allocation in Equation equation 10 is as follows:

$$\max_{\phi_{1:L}} \sum_{i=1}^{L} \sum_{j \in \text{mlp,mha}} w_j \left( s_i^j (1 - \phi_i^j) + \varepsilon H(\phi_i) \right),$$

$$\text{such that} \quad \frac{1}{L(w_{\text{mlp}} + w_{\text{mha}})} \sum_{i=1}^{L} \sum_{j \in \text{mlp,mha}} w_j \phi_i^j = \phi_{\text{avg}}, \quad 0 \le \phi_i^j \le 1, \tag{42}$$

where $\phi_i^j$ and $s_i^j$ represent the sparsity and score for the $j$-th block in layer $i$, respectively, and the weights $w_{\text{mlp}} = 2, w_{\text{mha}} = 1$ are applied to preserve the average sparsity, consistent with the parameter size ratio in transformer blocks. The solution has a similar closed-form expression:

$$\phi = L(w_{\text{mlp}} + w_{\text{mha}})\phi_{\text{avg}} \times \text{Softmax}(-\mathbf{w} \odot \mathbf{s}/\varepsilon). \tag{43}$$

Importantly, these updates come with minimal computational overhead. Although we now calculate two scores per layer (instead of one), the computational cost is negligible as score calculation remains lightweight and does not increase compression time.

**Table 28:** Enhanced Compression through Nonuniform Sparsity in MLP and MHA Blocks.

| Method | MLP Sparsity Mean | MHA Sparsity Mean | ↑Throughput (token/s) | PIQA | HellaS. | WinoG. | ARC-e | ARC-c | ↑Average |
|---|---|---|---|---|---|---|---|---|---|
| SLEB Song et al. (2024) | 30.0% | 30.0% | 2539.39 (1.49×) | 69.58 | 58.28 | 58.17 | 52.36 | 31.91 | 54.06 |
| SliceGPT Ashkboos et al. (2024) | 30.0% | 30.0% | 1815.67 (1.07×) | 68.55 | 48.69 | 59.75 | 56.69 | 34.47 | 53.63 |
| MoDeGPT (uniform module sparsity) | 30.0% | 30.0% | 2490.15 (1.46×) | 73.34 | **65.90** | 66.22 | 65.49 | 39.16 | 62.02 |
| MoDeGPT (nonuniform module sparsity) | 26.8% | 36.4% | **2722.98 (1.60×)** | **73.78** | 65.14 | **68.03** | **66.79** | **38.40** | **62.43** |

## B.12 ABLATION STUDY ON COMPRESSION IN EACH MODULE

**Impact of Module-Wise Compression on Perplexity.** Table 29 presents the perplexity changes in the LLAMA-2 7B model when compressing each module individually. The rightmost column shows the normalized slope of perplexity change relative to the parameter size in each module. The results reveal that the MLP module has the most significant impact on overall performance, likely due to its containing 66% of the model's parameters. On the other hand, the slope indicates that the compression algorithms for Type I and Type III modules perform comparably, while Type II performs the worst. This finding aligns with our theoretical results, which suggest that the reconstruction bounds are weakest in Type III. From a decomposition perspective, the CR approximation is the most coarse-grained, leading to the least effective compression outcomes.

**Table 29:** Perplexity of compressed LLAMA-2 7B in each module.

| Module \ Compression Rate | 0% | 10% | 20% | 30% | 40% | 50% | Normalized Slope |
|---|---|---|---|---|---|---|---|
| Type I: MLP | 5.12 | 5.34 | 5.68 | 6.71 | 7.12 | 8.24 | **0.094** |
| Type II: Query, Key | 5.12 | **5.14** | **5.23** | 5.43 | **5.58** | 6.33 | 0.121 |
| Type III: Value, Output | 5.12 | 5.16 | 5.24 | **5.37** | 5.62 | **5.92** | 0.095 |

**Table 31:** Heterogeneous sparsity allocation in modules.

| Sparsity (MLP, MHA) | Perplexity ↓ | ARC-e | ARC-c | PIQA | WinoGrande | HellaSwag | Average |
|---|---|---|---|---|---|---|---|
| 30%, 30% | 7.51 | **65.49%** | **39.16%** | **73.34%** | **66.22%** | **65.90%** | **62.02%** |
| 35%, 20% | 7.79 | 60.52% | 38.48% | 68.82% | 65.98% | 61.34% | 59.03% |
| 25%, 40% | 7.14 | 57.03% | 35.15% | 70.89% | 65.27% | 61.63% | 57.99% |

**Impact of Module-Wise Compression on Throughput.** Table 30 presents the throughputs for the 30% compressed LLAMA-2 7B across different modules. The results indicate that the compression yields similar speedups for both Type-II and Type-III modules. This sharp difference in speedups highlights the potential for uniform compression across modules, which we leave as a direction for future research.

**Table 30:** Module-Wise Throughputs of 30% Compressed LLAMA-2 7B

| Module | Throughputs (tokens/s) |
|---|---|
| Type I: MLP | 1585 |
| Type II: Query, Key | **2136** |
| Type III: Value, Output | 2121 |

**Heterogeneous Sparsity Across Modules** In our work, we apply nonuniform sparsity across layers while maintaining uniform sparsity across modules within the same layer. To investigate whether heterogeneity in module sparsity can enhance performance, we conducted an experiment compressing LLAMA-2 7B by 30%, with varying sparsity levels in the MLP and MHA blocks. The sparsity levels were adjusted to ensure the average compression rate remained at 30%.

We tested three configurations: equal sparsity for MLP and MHA, higher sparsity in MLP, and higher sparsity in MHA. The results are presented in Table 31 . From the table, we observe that while lower sparsity in MLP yields the best perplexity, it results in the worst zero-shot performance among the three configurations. Conversely, uniform sparsity across modules outperforms the other configurations in all tasks, while high and low MLP sparsity each demonstrate strengths in specific tasks compared to one another. These findings underscore the sensitivity of compression performance to variations in module sparsity, suggesting that a more sophisticated allocation method may be necessary to surpass the performance of uniform allocation.

## B.13 SCALABILITY TO LARGER MODELS

While our work is tested on a single GPU, it can be extended to multi-GPU setups to compress larger models, such as those with 70B parameters. To apply our method to larger models, the model must fit within the GPU memory to perform the forward pass. As shown in Table 24 , memory utilization is less than twice the model size, so approximately double the model's size in GPU memory is expected for running our method. In our compression process, the most computationally intensive part is the compression of the value-output module, as highlighted in Table 7. Since the computational complexity of this module scales cubically with the hidden dimension (due to the SVD in the value-output compression) and is proportional to the number of layers being compressed, the time required to compress a 70B model using multi-GPUs can be estimated using the following

formula:

$$\text{Compute Time (70B)}$$
$$= \text{Compute Time (7B)} \times \left( \frac{\text{hidden dim(70B)}}{\text{hidden dim(7B)}} \right)^3 \times \frac{\text{layer num(70B)}}{\text{layer num(7B)}}$$
$$= 4 \text{ hours} \times (8192/4096)^3 \times (80/32) = 80 \text{ hours}$$

For a sanity check, we applied the same formula to estimate the compression time for a 13B model and obtained an estimate of 9 hours, which aligns closely with our empirical result of 8 hours and 26 minutes, as shown in Table 7.

### B.14  HIGH COMPRESSION RATE EXPERIMENTS

**Table 32:** Perplexity of LLAMA-2 7B Across 10% to 80% Compressions

| Compression Rate | 0% | 10% | 20% | 30% | 40% | 50% | 60% | 70% | 80% |
|---|---|---|---|---|---|---|---|---|---|
| Perplexity | 5.12 | 5.48 | 6.16 | 7.51 | 8.41 | 11.88 | 26.59 | 84.22 | 245.84 |

We analyzed the perplexity of LLAMA-2 7B at high compression rates, using 128 samples from WikiText2 for calibration. We observed a significant breakdown point at 50% compression, where the perplexity increased sharply from 41% to 123%. This indicates the compression limit of our method.

### B.15  SENSITIVITY ANALYSIS OF DIFFERENT CALIBRATION SETS

In Table 33, we evaluate in-domain and out-of-domain perplexity using different calibration sets: WikiText2, PTB (Marcus et al., 1993), and Alpaca. Our results indicate that perplexity is minimized when the model is calibrated with the same dataset as the test set. Notably, when calibrated with different datasets, the results on Alpaca demonstrate the most consistent performance with the least variance, while PTB shows the highest variance. Nevertheless, calibration with PTB provides the most robust results across all three datasets.

### B.16  ADDITIONAL SPEEDUP EXPERIMENTS

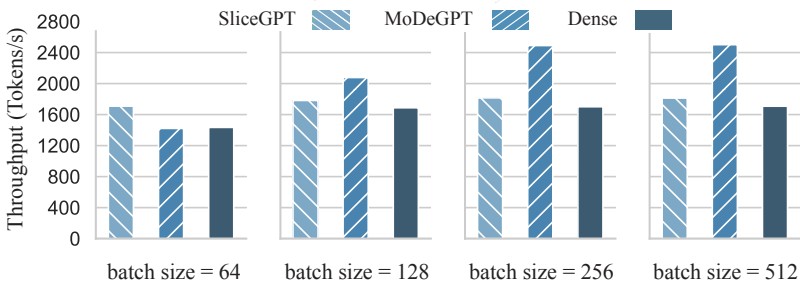

**Figure 11:** Throughput benchmarks of compressed LLAMA-2 7B on a single A100 80GB GPU.

Table 34 reports the throughputs and latency on fast and slow parallel computing environments using NVIDIA A100 and Intel(R) Xeon(R) CPU E5-2670 v2 @ 2.50 GHz with 20 cores. The results indicate that the reduction in computational complexity is proportional to the compression percentage. While the speedup for single-batch inference is comparable to that of the original model, we observe significant improvements in speedup for multi-batch inference on the GPU and on the CPU. Therefore, our method performs optimally when the parallel computing capabilities of the environment are fully utilized.

In Figure 11, we explored various batch sizes, comparing the throughput of 30% compressed MoDeGPT with 30% sliced SliceGPT (Ashkboos et al., 2024) and the dense model. We found that throughput surpassed that of the dense model for batch sizes over 64. Particularly, at batch

**Table 33:** Perplexity results under different calibration datasets.

| Test Set / Calibration Set | WikiText2 ↓ | PTB ↓ | Alpaca ↓ |
|---|---|---|---|
| WikiText2 | **6.16** | 27.69 (+22%) | 3.12 (+11%) |
| PTB | 6.99 (+13%) | **22.75** | 3.14 (+12%) |
| Alpaca | 7.64 (+24%) | 40.71 (+79%) | **2.80** |

**Table 34:** Inference speed and computational complexity of the pruned LLAMA-2 7B model.

| Method | # Parameter (B) | Memory (GiB) | Compute Complexity (GMACs) ↓ | Latency CPU (s/token) ↓ | Latency GPU (s/token) ↓ | 256-Batch Throughputs (tokens/s) ↑ |
|---|---|---|---|---|---|---|
| Dense | 6.74 | 12.92 | 425.12 (1.00×) | 32.41 (1.00×) | 0.035 (1.00×) | 1700 (1.00×) |
| 20% SliceGPT | 5.45 | 10.45 | **339.04 (0.80×)** | 26.46 (0.82×) | 0.037 (1.06×) | 1802 (1.06×) |
| 20% MoDeGPT | 5.44 | 10.43 | 339.34 (0.80×) | **22.66 (0.70×)** | **0.034 (0.97×)** | **2168 (1.28×)** |
| 30% SliceGPT | 4.73 | 9.07 | 298.36 (0.70×) | 25.28 (0.78×) | 0.037 (1.06×) | 1830 (1.08×) |
| 30% MoDeGPT | 4.79 | 9.07 | **297.91 (0.70×)** | **19.20 (0.59×)** | **0.034 (0.97×)** | **2521 (1.48×)** |
| 40% SliceGPT | 4.11 | 7.88 | 262.12 (0.62×) | 22.68 (0.70×) | 0.037 (1.06×) | 1839 (1.08×) |
| 40% MoDeGPT | 4.14 | 7.94 | **256.34 (0.60×)** | **18.57 (0.57×)** | **0.036 (1.03×)** | **2568 (1.51×)** |

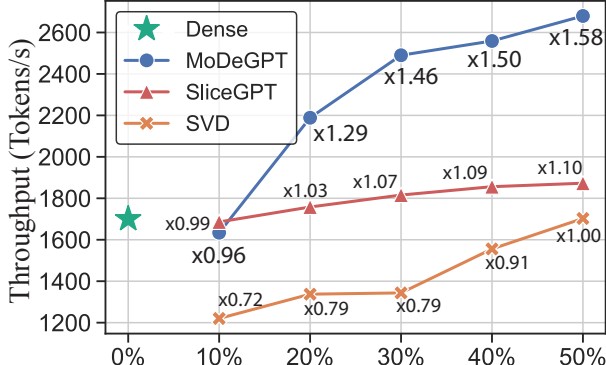

**Figure 12:** Speedup vs. compression.

sizes exceeding 256, MoDeGPT's throughput was 1.46 times greater than the dense model, while SliceGPT only achieved 1.07 times the throughput. MoDeGPT's increased throughput stems from its reduced matrix size and avoidance of extra adapters in residual paths.

Finally, we benchmark throughput. We set the sequence length to 256 and recorded the average generation time of LLAMA-2 7B on a single A100 GPU with batch size 256. In Figure 12, SVD exhibits lower throughput than the uncompressed model due to the doubled amount of matrices in its decomposed form which makes computation less parallelizable. SliceGPT, while achieving greater throughput, sees less than a 10% speedup, hindered by additional computations in residual paths. In contrast, MoDeGPT achieves non-trivial speedups that increase with compression rates; at 50% compression, it achieves a 58% increase in throughput, significantly surpassing both SVD and SliceGPT. However, at compression rates below 10%, throughput drops below that of the uncompressed model. This decrease is attributed to the implementation of the compressed Type-II module, which needs an optimized kernel to better parallelize the computation of the pruned attention heads. We leave the implementation of such an optimized computation kernel as future work to address the corner case.

## C LIMITATIONS AND BROADER IMPACTS

**Intrinsic Bias** Our experiments on zero-shot tasks show that MoDeGPT excels in certain zero-shot tasks while underperforming in others, indicating an intrinsic bias toward specific tasks. Our

current method does not offer a definitive solution to eliminate this bias. Addressing bias removal will be a critical area for future research.

**Overfitting the Reconstruction Loss**    While MoDeGPT excels in zero-shot tasks by minimizing local reconstruction error, we noted instances where compressed models, despite achieving lower perplexity, underperformed in zero-shot tasks. This discrepancy may stem from the models overfitting local reconstructions to calibration data. Addressing this overfitting remains a challenge for our method.

**Broader Impacts**    The introduction of **Mo**dular **De**composition (MoDeGPT) significantly impacts the ethical deployment and broader adoption of Large Language Models (LLMs). By minimizing computational demands, MoDeGPT enables effective deployment on resource-constrained devices, democratizing access to cutting-edge AI and potentially reducing the technological divide between large and small entities.

Additionally, MoDeGPT 's efficiency in using computational resources can decrease energy consumption during AI training and inference, promoting sustainable AI practices and reducing the environmental impact of large-scale computations. However, the potential for increased misuse of AI technologies, such as surveillance and disinformation, highlights the need for robust governance and ethical frameworks.

Ultimately, by maintaining high accuracy while reducing model size, MoDeGPT ensures the reliability of AI applications in critical domains such as healthcare. The development of MoDeGPT thus promises greater AI accessibility and sustainability, but it also introduces new challenges in governance and ethical technology use.

