# OpenReview forum: "MoDeGPT: Modular Decomposition for Large Language Model Compression"
_ICLR.cc/2025/Conference — ICLR 2025 Oral_

### Official Review · Reviewer_dBU8 · 2024-10-28

**Soundness:** 3
**Presentation:** 3
**Contribution:** 3
**Rating:** 8
**Confidence:** 3

**Summary:**

This paper introduces MoDeGPT, a novel training-free compression method for large language models.  It presents a systematic framework for categorizing approximation challenges in Transformer compression, complete with error guarantees. MoDeGPT demonstrates significant performance gains. This method outperforms prior approaches in compression, and achieves a 46% increase in inference throughput.

**Strengths:**

The paper has the following strengths：

(1) The paper presents a novel training-free compression method called MoDeGPT, applies matrix decomposition at the module level for the first time, and extends the theoretical foundation for weight decomposition in language models.

(2) The paper offers a comprehensive literature review and theoretical analysis, demonstrates significant performance improvements through experimental results, and provides error guarantees along with a theoretical framework.

(3) The method outperforms previous approaches in compression performance, achieves a 46% increase in inference throughput, and enhances the practical value of large language models.

**Weaknesses:**

The Weaknesses of the paper are listed as follows：
(1) MoDeGPT shows intrinsic bias, performing well on some zero-shot tasks but poorly on others, and currently lacks a solution for bias removal.
(2) Overfitting of the model to calibration data prevents the compression method from generalizing across most tasks.

**Questions:**

The specific questions and suggestions are listed below:

(1)Do you consider evaluating on more diverse tasks to verify the method's generalizability?

(2)In the specific experiments, could you provide the chosen rank size for the matrix decomposition or an analysis of the related experiments?

---

> ### Author Response · Authors · 2024-11-22
> **Response to Reviewer dBU8**
>
> **Dear Reviewer dBU828**, we appreciate your time and insightful feedback. We especially thank you for evaluating our work as "novel" and recognizing the literature review and analysis as "comprehensive". We greatly appreciate your positive comments!
>
> ---
>
> ### **W1 & Q1: Intrinsic bias and model generalizability on more diverse zero-shot tasks**
>
> We evaluated our model’s generalizability on additional zero-shot tasks, including **OpenBookQA**, **COPA**, **Lambada**, **MMLU**, and **BoolQ**, and compared it against decomposition and layer-pruning baselines SliceGPT [1] and ShortGPT [2].
>
> The comparisons were conducted on 30% compressed Llama-2 7B. From the table below, we observed that MoDeGPT consistently outperforms the baselines across a diverse range of tasks**, demonstrating its robustness and generalizability.
>
> Additionally, the relative performance degradation compared to the dense model on these tasks:
> - The degradation is most significant for **Lambada**, with around **16.7% drops** compared to the average **7.2% drop**.
> - However, similar task biases are observed for the baselines, with **over 40% degradation** on the task. This suggests that **Lambada** is intrinsically sensitive to model compression.
>
> Despite this sensitivity, **MoDeGPT exhibits significantly better resistance** to performance degradation, with reductions of only 33% to 50% of the baseline degradation levels. This highlights our method’s advantage on tasks sensitive to compression.
>
> Notably, on the **COPA** task, **MoDeGPT achieves zero degradation**, suggesting that it is particularly well-suited for this task. Overall, while our method shows some intrinsic bias, it demonstrates strong and consistent performance across diverse tasks and superior robustness on compression-sensitive tasks.
>
> ---
>
> | **Method**         | **BoolQ** | **PIQA**  | **HellaS.** | **WinoG.** | **ARC-e** | **ARC-c** | **OBQA** | **COPA** | **Lamb.**  | **MMLU-ml** | **Average**  |
> |---------------------|-----------|-----------|-------------|------------|-----------|-----------|----------|----------|------------|-------------|--------------|
> | Dense           | 77.68%    | 79.05%    | 76.00%      | 68.98%     | 74.58%    | 46.33%    | 44.22%   | 87.00%   | 73.86%     | 39.29%      | 66.70%       |
> | SliceGPT [1]    | 61.99%    | 68.55%    | 48.69%      | 59.75%     | 59.69%    | 34.47%    | 31.40%   | 75.00%   | 21.02%     | 23.21%      | 48.08%       |
> | ShortGPT [2]    | 62.17%    | 64.48%    | 56.15%      | 64.33%     | 48.70%    | 32.59%    | 32.80%   | 79.00%   | 29.03%     | 24.11%      | 49.34%       |
> | MoDeGPT (ours)  | **69.76%**| **73.34%**| **65.90%**  | **66.22%** | **65.49%**| **39.16%**| **39.00%**| **87.00%**| **57.07%** | **32.14%**  | **59.51%**   |
>
> ---
>
> ### **W2: Overfitting of the model to calibration data**
>
> While calibration with a specific dataset may risk overfitting, our new experiments on layer sparsity allocation comparisons revealed that our global sparsity allocation improves resistance to overfitting compared to baselines.
>
> In these experiments, we used MoDeGPT as the base compression method, combined with our global sparsity allocation, the state-of-the-art allocation strategy **OWL**, and uniform allocation. The following key observations were made:
>
> - While OWL achieves better perplexity, our sparsity allocation **outperforms OWL on every downstream task**. This indicates that OWL may overfit the calibration data, as its low PPL does not translate to better generalization in downstream tasks.
> - Additionally, our method outperforms uniform allocation, demonstrating that global sparsity allocation not only enhances task generalization but also mitigates overfitting compared to the baseline.
> - By inspecting the sparsity standard deviation (visualized in **Figure 9**, Appendix **B.9**), we observed that our sparsity distribution is more heterogeneous. This suggests that **heterogeneity** plays a critical role in improving **task generalization** and preventing **overfitting**.
>
> ---
>
> | **Method**                       | **Sparsity Mean** | **Sparsity Std** | **Perplexity ↓** | **PIQA ↑** | **HellaS. ↑** | **WinoG. ↑** | **ARC-E ↑** | **ARC-C ↑** | **Average ↑** |
> |----------------------------------|-------------------|------------------|------------------|------------|---------------|--------------|------------|------------|--------------|
> | Uniform Allocation           | 30%              | 0%               | 9.06             | 65.18      | 55.31         | 63.69        | 52.36      | 30.80      | 53.47        |
> | Global Sparsity Allocation (Ours) | 30%              | 26.72%           | 7.51             | **71.40**  | **63.26**     | **67.32**    | **63.26**  | **38.73**  | **60.79**     |
> | OWL [3]                      | 30%              | 4.46%            | **6.9**          | 68.17      | 59.12         | 65.67        | 56.9       | 33.36      | 56.64        |
>
> ---

---

> ### Author Response · Authors · 2024-11-22
>
> ### **Q2: Rank of matrices in the experiments**
>
> Throughout our experiments, we selected **ε** values that maximize sparsity levels around **20–30%**. This approach avoids extreme sparsity while maintaining a certain level of heterogeneity, which proved to be very effective when compared against other state-of-the-art global allocation strategies, as shown in **Appendix B.9** and the **general response**.
>
> In **Appendix B.9**, we present the resultant ranks for 30% compression of the LLaMA-2 7B and 70B models, as shown in **Table 26** and **Figure 10**. These results were obtained using our global sparsity allocation strategy with **ε = 0.1** and **ε = 0.02** for the 7B and 70B models, respectively.
>
> A general trend observed across different model sizes includes:
> - **Ranks peak in the very first and last layers.**
> - **Ranks are minimal across approximately 75% of the model's depth.**
> - Rank distributions are remarkably similar for both models, suggesting a **deep connection between the allocation strategy and the LLaMA-2 architecture.**
>
>
> As a demonstration, the table below shows the ranks of the key, query, and value projection matrices of the Llama-2 7B model in every layer, the ranks for 70B model can be found in **Table 26 in Appendix B.9**:
>
> | **Model**             | **Layer Rank**                                                                                   |
> |------------------------|--------------------------------------------------------------------------------------------------|
> | Llama-2 7B        | 3989, 3886, 3813, 3889, 3750, 3616, 3598, 3612                                                  |
> |                        | 3625, 3593, 3546, 3660, 3654, 3568, 3575, 3544                                                  |
> |                        | 3453, 3241, 2997, 2703, 2413, 1741, 1620, 1217                                                  |
> |                        | 1129, 1254, 1054, 741, 1203, 1363, 2640, 4060                                                   |
>
>
>
> ---
>
> ### **References**
>
> [1] Ashkboos, S., et al. *"SliceGPT: Efficient fine-tuning of large language models by slicing and pruning,"* 2024.
> [2] Men, H., et al. *"ShortGPT: Compressed language models for faster inference and reduced memory footprint,"* 2024.
> [3] Yin, X., et al. *"Outlier-aware layer sparsification for efficient neural networks,"* 2023.

---

### Official Review · Reviewer_XvJF · 2024-11-01

**Soundness:** 2
**Presentation:** 3
**Contribution:** 2
**Rating:** 8
**Confidence:** 4

**Summary:**

This paper proposes MoDeGPT, an accurate structured pruning algorithm for LLMs.
The main idea of MoDeGPT is to define "modules", a novel pruning structure, and apply tailored decomposition algorithms for three different types of modules.
The main strengths of this paper are (1) introducing decomposition algorithms that are not previously used in this domain, (2) proposing a new global sparsity allocation algorithm, and (3) exhaustive experiments and theoretical analysis in Appendix.
However, I have concerns regarding the following: (1) overclaiming regarding the efficiency of MoDeGPT,  (2) lack of experiments regarding large models, e.g., Llama 3 70B, and (3) too simplified proof of Theorem 4.
Therefore, I summarized my concerns in "Weaknesses" and "Questions" and I need to discuss them with the authors.
The score can be increased according to the author's response.

**Strengths:**

This paper has diverse strengths and I summarize them as follows:

### Method
1. The authors introduce Nystrom approximation, CR decomposition, and SVD to pruning row-column pairs in LLMs. To the best of my knowledge, this is the first work to use Nystrom approximation and CR decomposition to prune LLMs. The authors carefully use them to prune different types of modules.

2. The authors propose a novel global sparsity allocation algorithm with entropic regularization. If this algorithm contributes a lot to improving the accuracy of the pruned models, then this algorithm can be broadly used in pruning.

### Experiments
3. The authors conduct exhaustive experiments to show the superiority of MoDeGPT. Their experiments not only covers accuracies, but also inference speed and pruning cost.

4. The authors analyze the effect of MoDeGPT in a detailed way. They also analyze the sparsity patterns.

### Writing

5. The contents are well-organized and easy to read. Specifically, the authors assign unique colors for each module type and consistently use them. This was very helpful to understand this paper.

**Weaknesses:**

### Method

1. In the caption of Figure 1, the authors insist that their new pruning structure avoids the need for extra adapters. However, SliceGPT's adapters are introduced to improve accuracy and can be removed for inferencing without (dimensional) errors. Therefore, that statement should be modified.

2. The main contribution of this paper is introducing diverse decomposition algorithms and applying them to the proper modules. However, there are lack of explanations of the characteristics of these decomposition algorithms and justification for using them for each type of module.

3. The proof of Theorem 4 is too simplified and hard to understand. There are lack of explanations to get Equation 33. The authors impose a strong assumption that epsilon becomes infinity which indicates the uniformness of phis.

### Experiment

4. The authors emphasize that MoDeGPT is an efficient pruning algorithm, for example, in Lines 475-477, but MoDeGPT requires expensive pruning costs more than 8 hours for pruning Llama-2 13B models. According to SELB [1], most of pruning algorithms requires less than 16 minutes for pruning Llama-2 13B models. Therefore, it is an overclaiming to insist that MoDeGPT is an efficient algorithm.

5. There are lack of competitors. The authors should compare their results with state-of-the-art pruning algorithms, especially layer (or block) pruning algorithms, such as SLEB [1]. Layer pruning algorithms provide significant inference speedup and should be included in Figure 3.

### Writing

6. The second paragraph of the Introduction is too detailed and hard to find the main point. It is hard to capture "these challenges" in the third paragraph after reading.

7. The criteria of Table 1 are ambiguous. (1) "No backward propagation" seems like an indirect criteria of pruning efficiency, but MoDeGPT is slow without requiring backpropagation. (2) What is the threshold of maintaining accuracy? (3) SparseGPT supports 2:4 pruning which is treated as a (semi-)structured pruning algorithm.

**Questions:**

1. Can MoDeGPT outperform "efficient" competitors, such as SliceGPT [2], SLEB, if the competitors perform fine-tuning on the sample dataset to have the same pruning cost as MoDeGPT?

2. Could you elaborate on the detailed explanation of the proof for Theorem 4? Is it permissible to assume that epsilon is large enough to simplify the problem?

3. Does the proposed Global Sparsity Allocation outperform OWL [3]'s strategy?

4.  Does MoDeGPT outperform competitors when pruning gigantic models, e.g., Llama-3 70B?

5. What are the characteristics of Nystrom approximation, CR decomposition, and SVD, and why do we have to use them as proposed in this paper?

### References

[1] Song, Jiwon, et al. "SLEB: Streamlining LLMs through Redundancy Verification and Elimination of Transformer Blocks." arXiv preprint arXiv:2402.09025 (2024).

[2] Ashkboos, Saleh, et al. "Slicegpt: Compress large language models by deleting rows and columns." arXiv preprint arXiv:2401.15024 (2024).

[3] Yin, Lu, et al. "Outlier weighed layerwise sparsity (owl): A missing secret sauce for pruning llms to high sparsity." arXiv preprint arXiv:2310.05175 (2023).

---

> ### Author Response · Authors · 2024-11-22
> **Response to Reviewer XvJF**
>
> **Dear reviewer XvJF31**, we appreciate your time and thoughtful feedback. Especially, thank you for evaluating our work as “novel”, having “exhaustive experiments”, and recognizing our writing as “well-organized”. We are sincerely encouraged by your thoughtful comments!
>
> ---
>
> ### **W1: SliceGPT's adapter overhead can be eliminated**
>
> While removing the adapters does not lead to dimensional errors, it significantly **reduces performance**. To substantiate this, we evaluated the zero-shot performance of SliceGPT with and without adapters, as detailed in the table below. The experiments were conducted using Llama-2 7B with 30% compression, fine-tuned on the Alpaca dataset.
>
> The results demonstrate that the **adapters are indispensable** for maintaining model performance.
>
> ---
>
> | **Method**           | **BoolQ**  | **PIQA**   | **HellaS.** | **WinoG.** | **ARC-e**  | **ARC-c**  | **OBQA**   | **COPA**   | **Lamb.**  | **MMLU-ml** | **Average**  |
> |-----------------------|------------|------------|-------------|------------|------------|------------|------------|------------|------------|-------------|--------------|
> | Dense            | 77.68%     | 79.05%     | 76.00%      | 68.98%     | 74.58%     | 46.33%     | 44.22%     | 87.00%     | 73.86%     | 39.29%      | 66.70%       |
> | SliceGPT w/ adapters [1] | 61.99%     | 68.55%     | 48.69%      | 59.75%     | 59.69%     | 34.47%     | 31.40%     | 75.00%     | 21.02%     | 23.21%      | 48.08%       |
> | SliceGPT w/o adapters [1] | 50.37%     | 50.16%     | 26.14%      | 52.17%     | 25.29%     | 27.56%     | 25.60%     | 66.00%     | 0.00%       | 31.25%      | 35.45%       |
> | MoDeGPT (ours)   | **69.76%** | **73.34%** | **65.90%**  | **66.22%** | **65.49%** | **39.16%** | **39.00%** | **87.00%** | **57.07%** | **32.14%**  | **59.51%**   |
>
> ---
> ### **W2 & Q5: Characterizations of module and justifications for the decompositions**
>
> We have revised **Section 3.2** to provide clearer intuition and justification for our approach. Below is a brief summary of the key rationale:
>
> - The main reason for selecting different decompositions is the **number of nonlinear functions** in each module, which varies across modules:
>   - **Type-1** module: **1 nonlinear function**.
>   - **Type-2** module: **2 nonlinear functions**.
>   - **Type-3** module: **0 nonlinear functions**.
>
> - This variation leads to differing levels of complexity when solving the proposed modular decomposition problem (Equation 6). Specifically:
>   - For matrices embedded within nonlinear functions, directly solving Equation 6 without any structural constraints on the compressed matrix is **intractable**.
>   - To address this, we constrain the compressed matrix to the form of a multiplication by a column selection matrix (Section 3.2).
>
> - However, this constraint introduces additional challenges:
>   - The column selection matrix is highly **structured**, with only one non-zero element per column.
>   - Consequently, standard methods such as SVD are not suitable, as they generally produce dense matrices, which conflict with the desired structure.
>
> - Our **technical contribution** lies in deriving **closed-form optimization solutions** for these cases:
> - Depending on the number of matrices involved in column selection matrix multiplication, the optimal solutions correspond to different decompositions:
>   - **Type-3 (0 nonlinear functions):**
>     The compressed matrices are without constraints, so we can simply use **SVD** to obtain the optimal solution.
>   - **Type-1 (1 nonlinear function):**
>     Only one matrix is multiplied by a column selection matrix, and solving this selection matrix is equivalent to finding the optimal landmarks, as in the **Nystrom approximation**.
>   - **Type-2 (2 nonlinear functions):**
>     Two matrices are multiplied by a shared column selection matrix. As the key and query are multiplied together, the selection matrix can be solved by finding the optimal landmarks as in the **CR decomposition**.
> These connections are formalized in **Theorems 1, 2, and 3**.
>
> ---
> ### **W3 & Q2: lack of details in the proof of Theorem 4 and the assumption is too strong**
>
> We have revised the proof to include all the necessary details, as provided in **Appedix A.4**.
>
> Regarding **$\varepsilon$**, we would like to emphasize that we do **not assume it to be infinite**. Instead, we take the limit simply to demonstrate the **existence of a sufficiently large number** $N$, such that when $\varepsilon > N$, the proposed solution in Equation 11 is optimal.
>
> This explanation has been addressed with mathematical rigor in the updated **Appendix A.4**.
>
> ---

---

> ### Author Response · Authors · 2024-11-22
>
> ### **W4: Justification for efficiency**
> Although MoDeGPT requires a longer compression time compared to methods like SliceGPT and layer-pruning approaches such as ShortGPT and SLEB, it demonstrates superior efficiency in terms of both **accuracy** and **cost**.
> To substantiate MoDeGPT's cost and accuracy efficiency despite its longer compression time, we conducted evaluations under identical **computational budgets** (accounting for both compression and recovery fine-tuning using LoRA) and compared its accuracy performance against baseline methods.
>
> Specifically, we:
> 1. Conducted experiments on 30% compressed Llama-2 7B using the Alpaca dataset for calibration.
> 2. Adjusted the fine-tuning epochs for each method to equalize total computational budgets, accounting for differences in compression times.
> 3. Fixed the LoRA parameters to be consistent across all methods (lora_alpha=10, lora_r=32).
>
> The table below presents zero-shot accuracies before and after fine-tuning (shown as **after/before**).
>
> ### **Key Insights**:
> 1. **MoDeGPT outperforms baselines**:
>    - MoDeGPT achieves the **highest zero-shot accuracy** across all tasks (excluding perplexity), both **before and after fine-tuning**.
>    - Its superior performance is primarily attributed to the **compression phase**.
> 2. **Importance of compression**:
>    - The better perplexity but worse zero-shot performance of SliceGPT compared to MoDeGPT highlights the **critical importance of the compression phase**. Excessive focus on fine-tuning can exacerbate overfitting and underperform compared to a well-compressed model.
>
> 3. **SLEB's limited gains**:
>    - Despite its long fine-tuning time, SLEB achieves smaller improvements than SliceGPT in zero-shot performance, further emphasizing the pivotal role of compression in determining final performance.
>
> 4. **Effectiveness without fine-tuning**:
>    - MoDeGPT outperforms baselines **even without fine-tuning**, showcasing its effectiveness during the compression phase.
>
> In conclusion, while MoDeGPT has a longer compression time compared with some other baselines, it achieves the best performance under the same computation budget, which justifies that our method is also cost-efficient.
>
> ---
>
> | **Method**           | **Time (Compress / Fine-tune)** | **PPL (Alpaca)**           | **ARC-e**        | **ARC-c**        | **PIQA**         | **WinoG.**       | **HellaS.**      | **Average**       |
> |-----------------------|--------------------------------|--------------------|------------------|------------------|------------------|------------------|------------------|------------------|
> | SliceGPT [1]      | 26m / 4h05m                   | **2.59** (3.52)   | 56.82 (56.69)    | 38.48 (34.47)    | 71.82 (68.55)    | 59.83 (59.75)    | 59.30 (48.69)    | 57.26 (53.63)    |
> | SLEB [3]          | 9m / 4h50m                    | 2.67 (4.36)       | 52.36 (52.36)    | 34.04 (31.91)    | 71.00 (69.58)    | 59.98 (58.17)    | 60.16 (58.28)    | 55.51 (54.06)    |
> | MoDeGPT           | 4h09m / 31m                   | 2.70 (**3.08**)   | **67.42 (65.49)**| **40.96 (39.16)**| **74.10 (73.34)**| **65.98 (65.49)**| **66.57 (65.90)**| **63.01 (62.02)**|
>
> ---
> ### **W5: Comparisons with layer pruning methods**
> We include comparisons of perplexity (lower the better) with layer pruning strategies **SLEB** [3] and **ShortGPT** [2] for 7B and 13B models below (see the table in the reply to Q4 for 70B comparisons), showing that **MoDeGPT** outperforms in all cases.
>
> Additionally, **Figure 3** in the main text demonstrates that **MoDeGPT** achieves a superior trade-off between perplexity and throughput.
>
> | **Method**                     | **7B**                |      |      |      |      | **13B**              |      |      |      |      |
> |--------------------------------|-----------------------|------|------|------|------|-----------------------|------|------|------|------|
> |                                | **10%**               | **20%** | **30%** | **40%** | **50%** | **10%**               | **20%** | **30%** | **40%** | **50%** |
> | **ShortGPT**    [2]               | 6.98                  | 14.31 | 33.21 | 71.04 | 268.11 | 5.40                  | 7.69  | 30.48 | 48.83 | 187.23 |
> | **SLEB**              [3]         | 6.05                  | 7.64  | 11.23 | 29.10 | 103.38 | 5.23                  | 6.31  | 8.24  | 11.76 | 27.67 |
> | **MoDeGPT (Ours)**             | **5.48**              | **6.16** | **7.51** | **8.41** | **11.88** | **4.83**              | **5.29** | **6.10** | **6.95** | **8.95** |
>
> ---

---

> ### Author Response · Authors · 2024-11-22
>
> ### **W6: Unclear presentation of challenges in introduction.**
>
> We have revised the second paragraph of the introduction to clearly summarize the challenges by adding the lines
> "In summary, matrix decomposition approaches either (i) *discard a large portion of ranks*, or (ii) *introduce substantial parameter overheads*. These challenges significantly hinder the effective reduction of parameters without compromising accuracy."
>
> ---
>
> ### **W7: Ambiguous criteria in Table 1**
>
> 1. **Backward Propagation and Memory Efficiency**:
>    Although not relying on backward propagation does not necessarily result in a faster algorithm, it is usually more **memory efficient**. Backward propagation often consumes many times the memory of the model size, making its avoidance desirable for limited-resource environments. For instance, our algorithm can run on a **single GPU** for a 13B model, whereas methods relying on backward propagation, such as **LLM Pruner [5]**, require at least **two GPUs** and consume over **100GB of memory** in our experiments.
>
>
> 2. **Maintaining accuracy without fine-tuning**:
>    We agree that this criterion could lead to ambiguities. Based on your feedback, we have **removed this criterion** from Table 1. Thank you for the suggestion.
>
> 3. **Structured vs. semi-structured**:
>    We have revised Table 1 to better emphasize **fully-structured methods** and explicitly denote SparseGPT as semi-structured. We believe this distinction is significant, as semi-structured methods require special GPU support to achieve real-time speedup, creating a gap in practical applicability.
>
> ---
>
> ### **Q1: Can MoDeGPT outperforms others with the same pruning cost?**
>
> Please refer to the response to W4 above.
>
> ---
> ### **Q2: More details on the proof of Theorem 4.**
>
> Please refer to the response to W3 above.
>
> ---
> ### **Q3: Does the proposed Global Sparsity Allocation outperform OWL [4]'s strategy?**
> We updated **Appendix B.9** to include experiments comparing our method with the state-of-the-art allocation approach **OWL [4]** and uniform allocation** as baselines on 30% Llama2-7B compression in the first table below  (**Table 27 in Main**).
>
> In these experiments, we used MoDeGPT as the base compression method combined with our global sparsity allocation, OWL, and uniform allocation. Key observations from the results are as follows:
>
> - While OWL achieves better perplexity, our sparsity allocation **outperforms OWL on every downstream task**. This suggests that our method might be more **generalizable**.
> -By inspecting the sparsity standard deviation (a visualization of the distribution difference is also provided in **Figure 9** in **Appendix B.9**), we found that our distribution is more heterogeneous. This observation suggests that **heterogeneity** could play an important role in enhancing **task generalizability**.
> - The results are consistent with the findings from the Llama2-70B experiments, as shown in the second table below (**Table 6 in Main**).
>
> ---
>
> | **Method**                       | **Sparsity Mean** | **Sparsity Std** | **Perplexity ↓** | **PIQA ↑** | **HellaS. ↑** | **WinoG. ↑** | **ARC-E ↑** | **ARC-C ↑** | **Average ↑** |
> |----------------------------------|-------------------|------------------|------------------|------------|---------------|--------------|------------|------------|--------------|
> | Uniform Allocation           | 30%              | 0%               | 9.06             | 65.18      | 55.31         | 63.69        | 52.36      | 30.80      | 53.47        |
> | Global Sparsity Allocation (Ours) | 30%              | 26.72%           | 7.51             | **71.40**  | **63.26**     | **67.32**    | **63.26**  | **38.73**  | **60.79**     |
> | OWL [4]                      | 30%              | 4.46%            | **6.9**          | 68.17      | 59.12         | 65.67        | 56.9       | 33.36      | 56.64        |
>
> ---
>
> | **Method**                          | **WikitText-2 ↓** | **ARC-e ↑** | **ARC-c ↑** | **PIQA ↑** | **WinoG. ↑** | **HellaS. ↑** | **BoolQ ↑** | **OBQA ↑** | **MathQA ↑** | **MMLU-ml ↑** | **COPA ↑** | **Lamb. ↑** | **Average ↑** |
> |-------------------------------------|------------------|-------------|-------------|------------|--------------|---------------|-------------|-------------|--------------|---------------|-------------|-------------|---------------|
> | MoDeGPT + OWL Sparsity [4]  | **4.67**        | 76.01     | 50.34     | 74.70      | 72.85      | 72.43         | 69.88     | 44.20     | 32.26      | **44.64**     | 87.00       | 69.61     | 63.08       |
> | MoDeGPT + Our Sparsity      | 4.89          | **77.69**   | **50.94**   | **77.53**    | **76.87**    | **78.16**     | **74.71**   | **45.60**   | **35.04**    | *42.86*       | **89.00**   | **72.17**   | **65.51**     |
>
>
> ---

---

> ### Author Response · Authors · 2024-11-22
>
> ### **Q4: Does MoDeGPT outperforms competitors in large models (70B)?**
>
> - We conducted new experiments on the **Llama2-70B** model, yielding even more promising results than smaller models. Notably, we achieved:
>   - **4.5% and 3.2% drops in performance with 30% compression**, using only 128 calibration samples from WikiText-2 and Alpaca, respectively, without recovery fine-tuning.
>   - These results outperform decomposition and layer pruning baselines, including SliceGPT [1], ShortGPT [2], and SLEB [3].
>
> | **Method**                       |  **WikitText-2 ↓** | **ARC-e ↑** | **ARC-c ↑** | **PIQA ↑** | **WinoG. ↑** | **HellaS. ↑** | **BoolQ ↑** | **OBQA ↑** | **MathQA ↑** | **MMLU-ml ↑** | **COPA ↑** | **Lamb. ↑** | **Average ↑** |
> |-----------------------------------|------------------|-------------|-----------|----------|------------|-------------|-----------|-----------|------------|-------------|----------|----------|---------------|
> | Dense Llama-2 70B            | 3.12             | 80.98       | 57.25     | 82.75    | 77.90      | 83.83       | 83.79     | 48.80     | 38.42      | 42.86       | 94.00    | 79.60    | 70.02         |
> | SliceGPT                      | 5.76             | 67.05       | 42.06     | 67.52    | 71.11      | 55.57       | 41.56     | 40.20     | 27.87      | 32.14       | 82.00    | 52.03    | 52.65         |
> | ShortGPT                      | 66.33            | 60.65       | 34.47     | 72.74    | 64.01      | 63.80       | 66.88     | 34.40     | 23.05      | 31.25       | 75.00    | 27.01    | 48.06         |
> | SLEB                          | 5.54             | 71.97       | 44.20     | *77.74*  | 69.38      | *73.54*     | *67.25*   | 41.80     | 27.47      | *32.15*     | *88.00*  | 64.22    | 59.79         |
> | MoDeGPT + OWL Sparsity        | **4.67**         | 76.01       | 50.34     | 74.70    | 72.85      | 72.43       | 69.88     | 44.20     | 32.26      | **44.64**   | 87.00    | 69.61    | 63.08         |
> | MoDeGPT + Our Sparsity        | *4.89*           | *77.69*     | *50.94*   | 77.53    | *76.87*    | *78.16*     | *74.71*   | *45.60*   | **35.04**  | *42.86*     | *89.00*  | **72.17**| *65.51*       |
> | MoDeGPT + Our Sparsity + Alpaca| 5.73            | **78.57**   | **51.54** | **80.85**| **77.19**  | **79.60**   | **82.81** | **46.40** | *32.83*    | 40.18       | **94.00**| *70.72*  | **66.79**     |
>
>
> ---
>
> ### **Q5: What are the characteristics of the proposed method, and why to use them?**
>
> Please refer to the response to W2 above.
>
> ---
>
>
> **References**:
> [1] Ashkboos, S., et al. *"SliceGPT: Efficient fine-tuning of large language models by slicing and pruning,"* 2024.
> [2] Men, H., et al. *"ShortGPT: Compressed language models for faster inference and reduced memory footprint,"* 2024.
> [3] Song, Y., et al. *"SLEB: Structured Layer-wise Efficient BERT Pruning for Large-Scale Pre-trained Models,"* 2024.
> [4] Yin, X., et al. *"Outlier-aware layer sparsification for efficient neural networks,"* 2023.
> [5] Yuan, X., et al. *"LLM-Pruner: On the Structural Pruning of Large Language Models,"* *arXiv preprint* arXiv:2305.11627, 2023.

---

> > ### Comment · Reviewer_XvJF · 2024-11-24
> >
> > Thank you for your detailed response!
> > All of my concerns are resolved and I changed my score from 5 to 8.
> > Good luck!

---

> > > ### Author Response · Authors · 2024-11-24
> > >
> > > Thank you for the insightful review and suggestions for improvement. We are deeply encouraged that our revisions have addressed your questions, and we sincerely appreciate your thoughtful feedback!

---

### Official Review · Reviewer_Z3HT · 2024-11-04

**Soundness:** 3
**Presentation:** 3
**Contribution:** 3
**Rating:** 8
**Confidence:** 2

**Summary:**

This paper proposes MoDeGPT, which compresses transformers by applying structure decompositions on operations that span *two* weight matrices. The parameter subgroups targeted are the MLP weights, key and query projections, and value and attention output projections. Experimental results show that MoDeGPT is the best no-gradient structured method, and also comparable to the best structured and gradient-based method.

**Strengths:**

To the best of my knowledge, the method of structured approximations across multiple matrices is novel and the results are strong. For the most part, the paper is well-written.

**Weaknesses:**

One weakness is the lack of justification for the approximation methods for each weight group. Could you give more intuition behind why each method was chosen? For example, the sentence "Since $W_U$ is inside a nonlinear function $\sigma_s$, we constrain the search space for its approximation to a matrix multiplication $W_U S_k$ for tractability, where $S_k$ is the $k$-column selection matrix" (line 244) only describes the approximation, whereas a justification would explain why Nystrom is a better fit for this problem than other methods.

Another weakness is the relative lack of analysis on the global sparsity allocation. However, this is orthogonal to the main contribution of structured multi-weight approximations.

**Questions:**

1. In Table 3, is the main claim that although semi-structured methods may outperform MoDeGPT, they are held back by custom GPU support which hinders research velocity?
2. It would be nice to see a throughput versus perplexity graph as well, as opposed to just sparsity vs ppl/throughput, e.g. merge tables 2 and 3.

---

> ### Author Response · Authors · 2024-11-22
> **Response to Reviewer Z3HT**
>
> **Dear reviewer Z3HT03**, we appreciate your time and insightful feedback. Especially, thank you for evaluating our work as “novel”, “well-written” and for acknowledging that the "results are strong." We greatly appreciate your positive comments!
> Please see below of our response to your concerns.
>
> ---
> ### **W1: Justification for choice of decomposition for different modules**
>
>
> We have revised **Section 3.2** to provide clearer intuition and justification for our approach. Below is a brief summary of the key rationale:
>
> - The main reason for selecting different decompositions is the **number of nonlinear functions** in each module, which varies across modules:
>   - **Type-1** module: **1 nonlinear function**.
>   - **Type-2** module: **2 nonlinear functions**.
>   - **Type-3** module: **0 nonlinear functions**.
>
> - This variation leads to differing levels of complexity when solving the proposed modular decomposition problem (Equation 6). Specifically:
>   - For matrices embedded within nonlinear functions, directly solving Equation 6 without any structural constraints on the compressed matrix is **intractable**.
>   - To address this, we constrain the compressed matrix to the form of a multiplication by a column selection matrix (Section 3.2).
>
> - However, this constraint introduces additional challenges:
>   - The column selection matrix is highly **structured**, with only one non-zero element per column.
>   - Consequently, standard methods such as SVD are not suitable, as they generally produce dense matrices, which conflict with the desired structure.
>
> - Our **technical contribution** lies in deriving **closed-form optimization solutions** for these cases:
> - Depending on the number of matrices involved in column selection matrix multiplication, the optimal solutions correspond to different decompositions:
>   - **Type-3 (0 nonlinear functions):**
>     The compressed matrices are without constraints, so we can simply use **SVD** to obtain the optimal solution.
>   - **Type-1 (1 nonlinear function):**
>     Only one matrix is multiplied by a column selection matrix, and solving this selection matrix is equivalent to finding the optimal landmarks, as in the **Nystrom approximation**.
>   - **Type-2 (2 nonlinear functions):**
>     Two matrices are multiplied by a shared column selection matrix. As the key and query are multiplied together, the selection matrix can be solved by finding the optimal landmarks as in the **CR decomposition**.
> These connections are formalized in **Theorems 1, 2, and 3**.
>
> ---

---

> ### Author Response · Authors · 2024-11-22
>
> ### **W2: Lack of Analysis on Global Sparsity Allocation**
>
> We updated **Appendix B.9** to include experiments comparing our method with the state-of-the-art allocation approach **OWL [1]** and uniform allocation as baselines on 30% Llama2-7B compression in the first table below.
>
> In these experiments, we used MoDeGPT as the base compression method combined with our global sparsity allocation, OWL, and uniform allocation. Key observations from the results are as follows:
>
> - While OWL achieves better perplexity, our sparsity allocation **outperforms OWL on every downstream task**. This suggests that our method might be more **generalizable**.
> -By inspecting the sparsity standard deviation (a visualization of the distribution difference is also provided in **Figure 9** in **Appendix B.9**), we found that our distribution is more heterogeneous. This observation suggests that heterogeneity could play an important role in enhancing task generalizability.
> - The results are consistent with the findings from the Llama2-70B experiments, as shown in the second table.
>
> ---
>
> | **Method**                       | **Sparsity Mean** | **Sparsity Std** | **Perplexity ↓** | **PIQA ↑** | **HellaS. ↑** | **WinoG. ↑** | **ARC-E ↑** | **ARC-C ↑** | **Average ↑** |
> |----------------------------------|-------------------|------------------|------------------|------------|---------------|--------------|------------|------------|--------------|
> | Uniform Allocation           | 30%              | 0%               | 9.06             | 65.18      | 55.31         | 63.69        | 52.36      | 30.80      | 53.47        |
> | Global Sparsity Allocation (Ours) | 30%              | 26.72%           | 7.51             | **71.40**  | **63.26**     | **67.32**    | **63.26**  | **38.73**  | **60.79**     |
> | OWL [1]                      | 30%              | 4.46%            | **6.9**          | 68.17      | 59.12         | 65.67        | 56.9       | 33.36      | 56.64        |
>
> ---
>
> | **Method**                          | **WikitText-2 ↓** | **ARC-e ↑** | **ARC-c ↑** | **PIQA ↑** | **WinoG. ↑** | **HellaS. ↑** | **BoolQ ↑** | **OBQA ↑** | **MathQA ↑** | **MMLU-ml ↑** | **COPA ↑** | **Lamb. ↑** | **Average ↑** |
> |-------------------------------------|------------------|-------------|-------------|------------|--------------|---------------|-------------|-------------|--------------|---------------|-------------|-------------|---------------|
> | MoDeGPT + OWL Sparsity [1]  | **4.67**        | *76.01*     | *50.34*     | 74.70      | *72.85*      | 72.43         | *69.88*     | *44.20*     | *32.26*      | **44.64**     | 87.00       | *69.61*     | *63.08*       |
> | MoDeGPT + Our Sparsity      | *4.89*          | **77.69**   | **50.94**   | **77.53**    | **76.87**    | **78.16**     | **74.71**   | **45.60**   | **35.04**    | *42.86*       | **89.00**   | **72.17**   | **65.51**     |
>
>
> ---
> ### **Q1: In Table 3, is the main claim that although semi-structured methods may outperform MoDeGPT, they are held back by custom GPU support which hinders research velocity?**
>
> Indeed, the main claim is that while semi-structured methods may achieve better performance in specific scenarios, their reliance on custom GPU support significantly limits their practicality and adaptability. For instance, mobile device chips typically lack the necessary hardware support, making these methods inefficient or infeasible in such environments.
>
> ---
> ### **Q2: The Plot of Perplexity Versus Throughput**
>
> We have improved **Figure 3** in the main text to better illustrate the trade-off between **perplexity** and **throughput**. The relative model sizes are now annotated in the figure for added clarity.
> As shown, MoDeGPT achieves the best perplexity-throughput trade-off, with its line positioned in the bottom-right corner of the plot. This demonstrates the effectiveness of our method compared to other approaches.
>
> ---
> ### **References**
>
> [1] Yin, et al. "Outlier-aware layer sparsification for efficient neural networks," 2023.

---

### Official Review · Reviewer_ZuW6 · 2024-11-04

**Soundness:** 3
**Presentation:** 3
**Contribution:** 3
**Rating:** 8
**Confidence:** 3

**Summary:**

This paper proposes a novel model compression method by applying three different matrix decomposition algorithms to three distinct types of computations within Transformers. Compared to previous model compression algorithms, this approach achieves a significant improvement in performance.

**Strengths:**

1. The authors propose the interesting idea of using three different matrix decomposition algorithms to compress computations in both MLP and Attention.
2. Experimental results demonstrate that the proposed method offers advantages in terms of both performance and efficiency compared to prior pruning and matrix decomposition algorithms.
3. The Appendix includes additional methods and experiments related to group-query attention.

**Weaknesses:**

1. The authors suggest using three different types of matrix decompositions for three different types of computations within Transformers, but they do not provide motivation for this choice. For example, why is CR decomposition more suitable for Type-2 computation?

**Questions:**

1. Why does Table 3 include only 50% compression results for models like SparseGPT but lack results for 40% compression? Why is a 40% compression result of MoDeGPT compared to a 50% compression result of SparseGPT?
2. I am curious why magnitude-based and SVD-based compression methods seem to cause model collapse in Table 1, performing worse than random compression (Uniform).
3. The authors applied different compression rates to different layers, but are the compression rates for the three types of computations identical? Based on the analysis in Figure 4, it might be better to allocate a higher compression rate for Attention computations.
4. Why is MoDeGPT more efficient than the baseline at the same compression rate (Figure 3)?

---

> ### Author Response · Authors · 2024-11-22
> **Response to Reviewer ZuW6**
>
> **Dear Reviewer ZuW603**, Thank you for your time and thoughtful feedback. We particularly appreciate your recognition of the GQA-modified algorithm presented in the appendix and your positive comments. Below, we address your concern regarding the justification of different decompositions in our approach
>
> ---
>
> ### **W1: Justification for choice of decomposition for different modules**
>
>
> We have revised **Section 3.2** to provide clearer intuition and justification for our approach. Below is a brief summary of the key rationale:
>
> - The primary reason for selecting different decompositions lies in the **number of nonlinear functions** present in each module, which varies across the modules:
>   - **Type-1** module contains **1** nonlinear function.
>   - **Type-2** module contains **2** nonlinear functions.
>   - **Type-3** module contains **0** nonlinear functions.
>
> - This variation leads to differing levels of complexity when solving the proposed modular decomposition problem (**Equation 6**). Specifically:
>   - For matrices embedded within nonlinear functions, directly solving Equation 6 without any structural constraints on the compressed matrix is **intractable**.
>   - To address this, we constrain the compressed matrix to the form of a multiplication by a column selection matrix (Section 3.2).
>
> - However, this constraint introduces additional challenges:
>   - The column selection matrix is highly **structured**, with only one non-zero element per column.
>   - Consequently, standard methods such as SVD are not suitable, as they generally produce dense matrices, which conflict with the desired structure.
>
> - Our technical contribution lies in deriving closed-form optimization solutions by reformulating the problem into a form solvable using existing matrix decomposition techniques. Depending on the number of nonlinear functions in the module, the optimal solutions correspond to different decompositions:
>   - **Type-3 (0 nonlinear functions):**
>     The compressed matrices are without constraints, so we can simply use **SVD** to obtain the optimal solution.
>   - **Type-1 (1 nonlinear function):**
>     Only one matrix is multiplied by a column selection matrix, and solving this selection matrix is equivalent to finding the optimal landmarks, as in the **Nystrom approximation**.
>   - **Type-2 (2 nonlinear functions):**
>     Two matrices are multiplied by a shared column selection matrix. As the key and query are multiplied together, the selection matrix can be solved by finding the optimal landmarks as in the **CR decomposition**.
> - These connections and solutions are formalized in **Theorems 1, 2, and 3**.
>
> ---
> ### **Q1: Why does SparseGPT have a fixed compression rate?**
>
> SparseGPT employs a **semi-structured pruning approach** that enforces a specific sparsity pattern known as **2:4 sparsity**, where exactly two out of every four elements are set to zero. Due to this special pattern, the compression rate is **strictly fixed at 50%**.
> Additionally, unlike the fully-structured compression used in our method, this special sparsity pattern requires NVIDIA GPU support for effective real-time acceleration.
>
>
> ---
>
> ### **Q2: Why is the SVD-based method worse than uniform pruning?**
>
> Although neither method accounts for input statistics, uniform pruning has a milder impact on the output scale. For instance, with 30% compression, uniform pruning typically reduces the output scale by 30% on average across the entire model.
>
> In contrast, the SVD-based approach can inadvertently remove parts of the input space corresponding to the largest eigenvalues. This leads to a disproportionate reduction in the output scale, making it more variable and less predictable. Such sensitivity in output scaling reduces the stability of the LLM's overall outputs, ultimately degrading downstream performance.
>
> Furthermore, as shown in Figure 1, SVD produces two matrices during compression. Consequently, for a fixed compression rate, the rank reduction is effectively **doubled**, which further deteriorates the model’s accuracy.
>
> To summarize, two main factors contribute to the inferior performance of vanilla SVD compared to uniform pruning:
> 1. **Instability from input space disruption**: While uniform pruning impacts the model evenly, the SVD-based approach introduces instability by disproportionately altering critical components of the input-output relationship.
> 2. **Double rank reduction**: The rank is reduced twice during compression to accommodate the two matrices produced by SVD, further compromising the model’s accuracy.
>
>
> ---

---

> ### Author Response · Authors · 2024-11-22
>
> ### **Q3: Discussion of different compression rates among modules**
>
> While our work uses nonuniform sparsity across layers, we apply uniform sparsity across modules within the same layer. To investigate the impact of heterogeneous sparsity across modules, we conducted additional experiments by varying the compression rates for MLP and attention blocks while keeping the overall compression rate fixed at 30%. The results are presented in the table below and have also been updated in **Appendix B.10**.
>
> We found that applying larger compression rates to attention blocks improves perplexity but leads to poorer zero-shot task performance. This observation suggests that such a distribution might be more prone to **overfitting during calibration**.
>
> The table also demonstrates that performance is **sensitive to unequal sparsity distribution across modules**, indicating that a more sophisticated allocation strategy might be necessary to outperform the uniform strategy.
>
> ---
>
>   ### **Table: Heterogeneous Sparsity Allocation in Modules**
>
>   | **Compression (MLP, MHA)** | **Perplexity ↓** | **ARC-e (%)** | **ARC-c (%)** | **PIQA (%)** | **WinoGrande (%)** | **HellaSwag (%)** | **Average (%)** |
>   |-----------------------------|------------------|---------------|---------------|--------------|--------------------|-------------------|-----------------|
>   | 30%, 30%               | **7.51**        | **65.49**     | **39.16**     | **73.34**    | **66.22**          | **65.90**         | **62.02**       |
>   | 35%, 20%               | 7.79            | *60.52*       | *38.48*       | 68.82        | *65.98*            | 61.34            | *59.03*         |
>   | 25%, 40%               | 7.14            | 57.03         | 35.15         | *70.89*      | 65.27             | *61.63*          | 57.99          |
>
> ---
> ### **Q4: Why is MoDeGPT more efficient than the baseline at the same compression rate (Figure 3)?**
>
> MoDeGPT achieves superior efficiency in the **accuracy** and **throughput** trade-off, as shown in Figure 3, where its line lies in the bottom-right corner. This is due to the following factors:
>
> 1. **Mathematically Principled Compression**
>    - MoDeGPT’s compression method is principled and mathematically grounded, with all compressions derived using closed-form expressions.
> 2. **Fully-Structured Compression for Speedup**
>    - MoDeGPT leverages fully-structured compression, resulting in descent throughput speedup without the need for specialized GPU support. In contrast, methods like ShortGPT and SLEB rely on coarse compression strategies (e.g., layer pruning), achieving faster speedups but at the cost of accuracy loss.
>
> 3. **Modular Output Optimization**
>    - Unlike decomposition-based approaches such as SliceGPT or SVD, which optimize individual matrix independently, MoDeGPT minimizes the modular outputs by jointly optimizing a pair of matrices. This approach better aligns with the global behavior of the LLM’s output, ensuring superior downstream performance. However, this also introduces greater algorithmic challenges, which MoDeGPT successfully addresses.
>
> ---

---

> > ### Comment · Reviewer_ZuW6 · 2024-11-27
> >
> > Thanks to the authors for addressing all questions. I will keep my score.

---

> > > ### Author Response · Authors · 2024-11-27
> > >
> > > Thank you for your valuable feedback, which has been instrumental in helping us refine our paper. We are pleased to hear that our response addressed your questions. Our preliminary results suggest that naive heterogeneous sparsity allocation across modules does not outperform our current strategy. Nevertheless, we will continue exploring dedicated sparsity allocation methods and will share any additional insights and improvements that arise during the remainder of the rebuttal period!

---

> ### Author Response · Authors · 2024-12-03
> **Improved results on nonuniform module sparsity allocation**
>
> Dear Reviewer ZuW6,
>
> Thank you for your thoughtful feedback and for taking the time to review our paper. We hope you had a wonderful Thanksgiving holiday. Inspired by your invaluable suggestions, we have continued refining our methods, and we are excited to share the **latest improvements** to our methodology, particularly in **inference speed**.
>
> ---
>
> ### **Improved Speed and Accuracy with Nonuniform Module Sparsity**
> Following your insights and drawing inspiration from prior strategies [3], we refined our global sparsity allocation strategy by introducing distinct sparsity levels for the MLP and MHA blocks within each transformer layer. Instead of calculating a single score per layer, we now compute two scores—one for MLP and one for MHA using the same correlation as described in Section 3.3. The updated global sparsity allocation in Equation 10 is as follows:
>
> $$
> \max_{\phi_{1:L}}\sum_{i=1}^L\sum_{j \in \{\text{mlp}, \text{mha}\}} w_j (s^j_i (1-\phi^j_i) + \varepsilon H(\phi_i)) \quad \text{such that} \quad \frac{1}{L(w_{\text{mlp}} + w_{\text{mha}})} \sum_{i=1}^L \sum_{j \in \{\text{mlp}, \text{mha}\}} w_j \phi^j_i = \phi_{\text{avg}}, \quad 0 \leq \phi_i \leq 1,
> $$
>
> where $\phi^j_i$ and $s^j_i$ represent the sparsity and score for the $j$-th block in layer $i$, respectively, and the weights $w_{\text{mlp}}=2, w_{\text{mha}}=1$ are applied to preserve the average sparsity, consistent with the parameter size ratio in transformer blocks. The solution has a similar closed-form solution:
>
> $$
>     \phi = L(w_{\text{mlp}} + w_{\text{mha}})\phi_{\text{avg}}\times\text{Softmax}(-s\odot w/\varepsilon).
> $$
>
> This updated strategy has enhanced both compression accuracy and inference throughput, especially in the inference speed. Notably, in our 30% compression experiments on Llama2-7B (as shown in the table below), we achieved **the fastest throughput among all baselines (even faster than layer pruning strategies!) while maintaining superior accuracy**. Our updated allocation rule is consistent with your insight that our method can benefit from higher sparsity in MHA.
>
> Importantly, these updates come with minimal computational overhead. Although we now calculate two scores per layer (instead of one), the computational cost is negligible as score calculation remains lightweight and does not increase compression time.
>
> We are thrilled to share these findings and will include comprehensive experiments in the revised paper. Thank you for your insightful feedback and the time spent during the rebuttal period, which have greatly enhanced this research.
>
>
>
> | Method                                                 | MLP mean sparsity | MHA mean sparsity | ↑ Throughput  (tokens/s)     | ↑ PIQA | ↑ HellaS. | ↑ WinoG. | ↑ ARC-e | ↑ ARC-c | ↑ Average |
> |--------------------------------------------------------|-------------------|-------------------|---------------------|--------|-----------|----------|---------|---------|-----------|
> | SLEB [1]                                                 | 30%              | 30%              | 2539.39 (1.49x)    | 69.58  | 58.28     | 58.17    | 52.36   | 31.91   | 54.06     |
> | SliceGPT [2]                                              | 30%              | 30%              | 1815.67 (1.07x)    | 68.55  | 48.69     | 59.75    | 56.69   | 34.47   | 53.63     |
> | MoDeGPT                                               | 30%              | 30%              | 2490.15 (1.46x)    | 73.34  | **65.90**     | 66.22    | 65.49   | **39.16**   | 62.02     |
> | MoDeGPT w/ nonuniform module sparsity | 26.80%           | 36.43%           | **2722.98 (1.60x)**    | **73.78**  | 65.14     | **68.03**    | **66.79**   | 38.40   | **62.43**     |
>
>
> ---
>
> ### **References**
>
> [1] Song, Y., et al. "SLEB: Sparsity-aware learning for efficient bandwidth in language models," 2024.
> [2] Ashkboos, S., et al. "SliceGPT: Efficient fine-tuning of large language models by slicing and pruning," 2024.
> [3] Zhang, X., et al. "FINERCUT: Finer-grained Interpretable Layer Pruning for Large Language Models," 2024.

---

### Author Response · Authors · 2024-11-22
**General Response (1/2)**

We would like to thank all reviewers for their encouraging and constructive feedback. Specifically, we sincerely thank **all reviewers** for recognizing the **novelty** of our work.

The insightful comments and suggestions from the reviewers have **significantly enhanced the quality of our submission**. In response, we have conducted additional experiments, provided further clarification on key aspects of our methodology, and strengthened the justification for the MoDeGPT framework.
While we will address each reviewer’s feedback in detail, we summarize the major revisions and additions to the manuscript below:

---

### **Characterization of Modules and Justification of the Proposed Method**

- We have provided more intuitive explanations to justify the use of different decomposition strategies for various modules. Specifically:
  - Each module is characterized by the **number of nonlinear functions** it contains.
  - For each matrix within a nonlinear function, we constrain the compressed matrix to be in the form of a multiplication with a column selection matrix to be optimized. This form is essential for ensuring the **tractable optimization** of our modular decomposition objective.
  - The column selection matrix is sparse, with only one non-zero element per column, and therefore cannot be optimized using traditional SVD, which generally outputs dense matrices.
  - Depending on the number of nonlinear functions, the optimal compression solutions correspond to different types of matrix decompositions. These solutions are formally presented in Theorems 1, 2, and 3, which coincide with various existing matrix decomposition techniques.

---

### **Large Model Experiments (70B)**

- We conducted new experiments on the **Llama2-70B** model, yielding even more promising results than smaller models. Notably, we achieved:
  - **4.5% and 3.2% drops in performance with 30% compression**, using only 128 calibration samples from WikiText-2 and Alpaca, respectively, without recovery fine-tuning.
  - These results outperform decomposition and layer pruning baselines, including SliceGPT [1], ShortGPT [2], and SLEB [3].

| **Method**                       |  **WikitText-2 ↓** | **ARC-e ↑** | **ARC-c ↑** | **PIQA ↑** | **WinoG. ↑** | **HellaS. ↑** | **BoolQ ↑** | **OBQA ↑** | **MathQA ↑** | **MMLU-ml ↑** | **COPA ↑** | **Lamb. ↑** | **Average ↑** |
|-----------------------------------|------------------|-------------|-----------|----------|------------|-------------|-----------|-----------|------------|-------------|----------|----------|---------------|
| Dense Llama-2 70B            | 3.12             | 80.98       | 57.25     | 82.75    | 77.90      | 83.83       | 83.79     | 48.80     | 38.42      | 42.86       | 94.00    | 79.60    | 70.02         |
| SliceGPT                      | 5.76             | 67.05       | 42.06     | 67.52    | 71.11      | 55.57       | 41.56     | 40.20     | 27.87      | 32.14       | 82.00    | 52.03    | 52.65         |
| ShortGPT                      | 66.33            | 60.65       | 34.47     | 72.74    | 64.01      | 63.80       | 66.88     | 34.40     | 23.05      | 31.25       | 75.00    | 27.01    | 48.06         |
| SLEB                          | 5.54             | 71.97       | 44.20     | *77.74*  | 69.38      | *73.54*     | *67.25*   | 41.80     | 27.47      | *32.15*     | *88.00*  | 64.22    | 59.79         |
| MoDeGPT + OWL Sparsity        | **4.67**         | 76.01       | 50.34     | 74.70    | 72.85      | 72.43       | 69.88     | 44.20     | 32.26      | **44.64**   | 87.00    | 69.61    | 63.08         |
| MoDeGPT + Our Sparsity        | *4.89*           | *77.69*     | *50.94*   | 77.53    | *76.87*    | *78.16*     | *74.71*   | *45.60*   | **35.04**  | *42.86*     | *89.00*  | **72.17**| *65.51*       |
| MoDeGPT + Our Sparsity + Alpaca| 5.73            | **78.57**   | **51.54** | **80.85**| **77.19**  | **79.60**   | **82.81** | **46.40** | *32.83*    | 40.18       | **94.00**| *70.72*  | **66.79**     |



---

---

> ### Author Response · Authors · 2024-11-22
> **General Response (2/2)**
>
> ### **Analysis on Global Allocation Strategy**
>
> - We compared our **global sparsity allocation strategy** with the state-of-the-art allocation method (OWL [4]) and uniform allocation for 30% compression using MoDeGPT:
>   - While OWL improves upon uniform allocation, our method demonstrates superior zero-shot task performance for both small (7B, table below) and large (70B, table above) models.
>
>   | **Method**                       | **Sparsity Mean** | **Sparsity Std** | **Perplexity ↓** | **PIQA ↑** | **HellaS. ↑** | **WinoG. ↑** | **ARC-E ↑** | **ARC-C ↑** | **Average ↑** |
>   |-|-|-|-|-|-|-|-|-|-|
>   | Uniform Allocation          | 30%              | 0%               | 9.06             | 65.18      | 55.31         | 63.69        | 52.36      | 30.80      | 53.47        |
>   | Global Sparsity Allocation (Ours) | 30%              | 26.72%           | 7.51             | **71.40**  | **63.26**     | **67.32**    | **63.26**  | **38.73**  | **60.79**     |
>   | OWL [4]                      | 30%              | 4.46%            | **6.9**          | 68.17      | 59.12         | 65.67        | 56.9       | 33.36      | 56.64        |
> ---
> ### **References**
>
> [1] Ashkboos, S., et al. "SliceGPT: Efficient fine-tuning of large language models by slicing and pruning," 2024.
> [2] Men, H., et al. "ShortGPT: Compressed language models for faster inference and reduced memory footprint," 2024.
> [3] Song, Y., et al. "SLEB: Sparsity-aware learning for efficient bandwidth in language models," 2024.
> [4] Yin, et al. "Outlier-aware layer sparsification for efficient neural networks," 2023.

---

### Author Response · Authors · 2024-11-22
**Update of Manuscript**

Dear reviewers,

We sincerely thank you for your encouraging and constructive feedback on our manuscript. In response to your suggestions, we have revised the manuscript and **highlighted all changes in blue** for ease of review. Below is a summary of the key updates:

---

###  **Main Text**
- **Characterization of Modules:**
  **[Section 3.2]** We provided a clear roadmap of our algorithms, detailing the characteristics of each module and justifying the selection of specific matrix decompositions over alternatives.
- **Large Model (70B) Experiments:**
  **[Section 4.3]** We added results from experiments on the large-scale Llama2-70B model, demonstrating even more promising outcomes than smaller models, with only a 3.2% performance drop for 30% compression, achieved without fine-tuning.
- **Enhanced Accuracy-Throughput Trade-Off Plot:**
  **[Section 4.4]** We provided an improved plot to better illustrate the trade-off between throughput and perplexity.

---

### **Appendix**
- **Theory Part:**
  - **[Section A.4]** We refined the proof of Theorem 4, including detailed explanations to improve clarity and rigor.

- **Experiment Part:**
  - **[Section B.3]** Added experiments evaluating the compressed model on a more diverse set of tasks to assess task generalizability.
  - **[Section B.6]** Included baseline comparisons under equal computational cost constraints.
  - **[Section B.9]** Compared our global sparsity allocation approach with the state-of-the-art OWL method and reported layer-wise ranks.
  - **[Section B.10]** Added experiments evaluating the effect of using nonuniform sparsity across modules within the same layer.

---

---

### Author Response · Authors · 2024-12-03
**12/2 Update**

Dear Reviewers,

Thank you for your thoughtful feedback and for taking the time to review our paper. We hope you had a wonderful Thanksgiving holiday. During the rebuttal period, we have worked diligently to refine our methods and are excited to share **the latest improvements** to our methodology, particularly in **inference speed**.

---

### **Improved Speed and Accuracy with Nonuniform Module Sparsity**
Following **Reviewer ZuW6's** suggestion and drawing insights from prior strategies [3], we refined our global sparsity allocation strategy by introducing distinct sparsity levels for the MLP and MHA blocks within each transformer layer. Instead of calculating a single score per layer, we now compute two scores—one for MLP and one for MHA using the same correlation as described in Section 3.3. The updated global sparsity allocation in Equation 10 is as follows:

$$
\max_{\phi_{1:L}}\sum_{i=1}^L\sum_{j \in \{\text{mlp}, \text{mha}\}} w_j (s^j_i (1-\phi^j_i) + \varepsilon H(\phi_i)) \quad \text{such that} \quad \frac{1}{L(w_{\text{mlp}} + w_{\text{mha}})} \sum_{i=1}^L \sum_{j \in \{\text{mlp}, \text{mha}\}} w_j \phi^j_i = \phi_{\text{avg}}, \quad 0 \leq \phi_i \leq 1,
$$

where $\phi^j_i$ and $s^j_i$ represent the sparsity and score for the $j$-th block in layer $i$, respectively, and the weights $w_{\text{mlp}}=2, w_{\text{mha}}=1$ are applied to preserve the average sparsity, consistent with the parameter size ratio in transformer blocks. The solution has a similar closed-form solution:

$$
    \phi = L(w_{\text{mlp}} + w_{\text{mha}})\phi_{\text{avg}}\times\text{Softmax}(-s\odot w/\varepsilon).
$$

This updated strategy has enhanced both compression accuracy and inference throughput, especially in the inference speed. Notably, in our 30% compression experiments on Llama2-7B (as shown in the table below), we achieved **the fastest throughput among all baselines (even faster than layer pruning strategies!) while maintaining superior accuracy**. We are grateful to Reviewer ZuW6 for pointing out this direction and highlighting that our method can benefit from higher sparsity in MHA module.

Importantly, these updates come with minimal computational overhead. Although we now calculate two scores per layer (instead of one), the computational cost is negligible as score calculation remains lightweight and does not increase compression time.

We are thrilled to share these findings and will include comprehensive experiments in the revised paper. Thank you for your insightful feedback and the time spent during the rebuttal period, which have greatly enhanced this research.



| Method                                                 | MLP mean sparsity | MHA mean sparsity | ↑ Throughput  (tokens/s)     | ↑ PIQA | ↑ HellaS. | ↑ WinoG. | ↑ ARC-e | ↑ ARC-c | ↑ Average |
|--------------------------------------------------------|-------------------|-------------------|---------------------|--------|-----------|----------|---------|---------|-----------|
| SLEB [1]                                                 | 30%              | 30%              | 2539.39 (1.49x)    | 69.58  | 58.28     | 58.17    | 52.36   | 31.91   | 54.06     |
| SliceGPT [2]                                             | 30%              | 30%              | 1815.67 (1.07x)    | 68.55  | 48.69     | 59.75    | 56.69   | 34.47   | 53.63     |
| MoDeGPT                                               | 30%              | 30%              | 2490.15 (1.46x)    | 73.34  | **65.90**     | 66.22    | 65.49   | **39.16**   | 62.02     |
| MoDeGPT w/ nonuniform module sparsity | 26.80%           | 36.43%           | **2722.98 (1.60x)**    | **73.78**  | 65.14     | **68.03**    | **66.79**   | 38.40   | **62.43**     |


---

### **References**

[1] Song, Y., et al. "SLEB: Sparsity-aware learning for efficient bandwidth in language models," 2024.
[2] Ashkboos, S., et al. "SliceGPT: Efficient fine-tuning of large language models by slicing and pruning," 2024.
[3] Zhang, X., et al. "FINERCUT: Finer-grained Interpretable Layer Pruning for Large Language Models," 2024.

---

### Meta-Review · Area_Chair_vg7q · 2024-12-23

**Metareview:**

This paper proposes a new approach to compressing Transformer-based models. The idea is to use a set of particular forms of low-rank matrix factorization for the weight matrices. The authors’ strategy is fairly sophisticated, as it seeks to associate various component operations inside a Transformer with different matrix approximation approaches.

There’s a bunch of strengths here: the overall approach is creative, the empirical results are pretty strong. The authors have provided extensive details on the approach.

This is a good paper that is worth accepting.

**Additional Comments On Reviewer Discussion:**

The authors responded in great depth to pretty much all reviewer suggestions; the updated draft is now much stronger.

---

### Decision · Program_Chairs · 2025-01-22

Accept (Oral)